# AnalogVerifier: A Neuro-Symbolic Framework for Analog Circuit Verification

Yanfang Liu [* 1]   Mingjun Wang [* 1]   Peng Xu [1]   Rongliang Fu [1]   Bei Yu [1]   Tsung-Yi Ho [1]

## Abstract

Analog circuits constitute the indispensable interface between physical reality and digital computation, underpinning safety-critical systems from autonomous driving to medical implants. Consequently, verification correctness is paramount; yet, it remains the critical bottleneck in hardware design, consuming over 50% of engineering cycles due to a heavy reliance on the manual interpretation of unstructured, heterogeneous specifications. While Large Language Models (LLMs) offer automation potential, their probabilistic, autoregressive nature is structurally misaligned with the strict determinism required for analog verification, struggling with semantic dispersion, latent causal dependencies, and numerical precision. To bridge this gap, we introduce AnalogVerifier, a neuro-symbolic framework that automates end-to-end testbench generation by decoupling semantic translation from logical enforcement. It comprises four parts: (1) Context-Aware Task Serialization transforms complex specifications into atomic tasks via an agentic workflow; (2) Graph-Symbolic Scheduling satisfies analog design constraints through Port Dependency Graphs (PDG) for correct-by-construction sequencing; (3) Numerical-Symbolic Grounding mitigates numerical hallucination by delegating threshold derivation to a deterministic symbolic oracle; (4) Closed-Loop Repair enables correctness and completeness of the generated testbenches by simulation feedback. Evaluation on five industrial analog circuits demonstrates that AnalogVerifier achieves 82.3%–100% functional pass rate, establishing a new paradigm for reliable, automated analog verification.

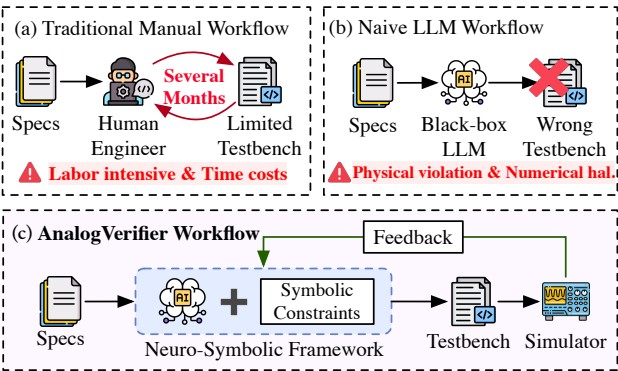

*Figure 1.* Comparison of verification workflows. (a) Manual approaches require iterative human effort. (b) Naive LLM application produces physical violations and numerical hallucinations. (c) AnalogVerifier combines LLM-based specification understanding with symbolic constraints, using simulator feedback for iterative refinement.

[*]Equal contribution   [1]The Chinese University of Hong Kong, HKSAR. Correspondence to: Peng Xu <pxu22@cse.cuhk.edu.hk>.

*Proceedings of the 43rd International Conference on Machine Learning*, Seoul, South Korea. PMLR 306, 2026. Copyright 2026 by the author(s).

## 1. Introduction

Analog circuits form the indispensable bridge between digital computation and the physical world, governing sensing, power conversion, and signal integrity across safety-critical domains from automotive systems to medical devices to aerospace platforms. Circuit correctness in these applications determines system reliability and human safety, rendering rigorous verification non-negotiable. Yet analog verification remains the dominant bottleneck in design cycles: industry data indicates that verification consumes 50% to 70% of total development effort, with testbench creation alone demanding up to several months of engineering time (Figure 1(a)) (Kundert & Chang, 2006). This burden delays time-to-market and embeds human error at a stage where undetected faults propagate irreversibly into silicon.

Large language models (LLMs) have demonstrated remarkable capabilities in code synthesis (Jiang et al., 2026), prompting initial explorations into verification automation. Prior works have applied LLMs to digital testbench generation and assertion synthesis (Liu et al., 2024b; Yan et al., 2025), yet these efforts operate under favorable conditions: digital specifications reduce to explicit coverage targets that can be directly enumerated. Analog verification inhabits a fundamentally different regime. Specifications scatter across hundreds of pages of dense prose, nested tables, and

parametric equations with no unified structure. Coverage cannot be enumerated but must instead navigate continuous, high-dimensional parameter spaces bounded by implicit physical constraints. Correctness criteria emerge not from explicit statements but from the intricate interplay of electrical, thermal, and timing behaviors that demand deep domain expertise to decode. Despite the pressing need for automation, the end-to-end synthesis of executable testbenches from industrial analog specifications remains uncharted territory.

Automating analog verification with LLMs is severely impeded by three fundamental challenges. First, semantic dispersion in long-context retrieval complicates the synthesis of constraints scattered across extensive specifications. Here, the "lost-in-the-middle" phenomenon (Liu et al., 2024a) causes models to overlook fine-grained parameters buried in document interiors, failing to map tabular electrical limits to their functional contexts reliably. Second, latent dependency inference necessitates resolving implicit physical invariants—such as power sequencing and bias settling. Deriving a globally consistent schedule from these dispersed pairwise constraints requires computing the transitive closure of causal dependencies, a logical operation that exceeds the reliability of probabilistic next-token predictors. Third, numerical hallucination stems from the "neuro-symbolic gap" (Ji et al., 2023; Zhang et al., 2025; Liu et al., 2026), where the requirement for precise boundary derivation—via unit conversion and tolerance propagation—conflicts with the model's linguistic training objectives, yielding plausible yet arithmetically unfaithful testbenches laden with silent errors (Figure 1(b)).

Our key insight is that the limitations of generic LLMs in this domain stem from a fundamental architectural misalignment: the probabilistic, autoregressive nature of Transformers is ill-suited for the strict global consistency required by analog circuit verification. We argue that robust automation demands shifting from monolithic prompting to structured decomposition, governed by three methodological imperatives. First, addressing semantic dispersion requires Attention Locality: since static context windows dilute focus, effective retrieval necessitates contextual bounding to confine extraction within the model's high-fidelity attention regime. Second, resolving latent dependencies demands reasoning externalization: as maintaining global graph state is orthogonal to local next-token prediction, causal logic must be delegated to deterministic algorithms that guarantee correctness by construction. Third, eliminating numerical hallucination hinges on arithmetic decoupling: reliability requires separating the symbolic definition of formulas from their precise evaluation to enforce mathematical validity.

We instantiate these insights in **AnalogVerifier**, a neuro-symbolic analog verification framework automating end-to-end testbench generation directly from industrial, unstructured analog specifications (as shown in Figure 1). This paper makes the following contributions:

- **Context-Aware Task Serialization** addresses semantic dispersion by decomposing complex specifications into typed, independent constraint units via a dynamic agentic workflow. Each unit is processed within a bounded context, and results are aggregated into a structured document database that mitigates retrieval failures before generation.
- **Graph-Symbolic Scheduling** addresses latent dependency inference by constructing a Port Dependency Graph (PDG) that explicitly encodes physical precedence relations. Stimulus schedules are derived via topological sorting, guaranteeing that every generated testbench respects the transitive closure of timing constraints-correctness by construction rather than probabilistic generation.
- **Numerical-Symbolic Grounding** addresses numerical hallucination by delegating all threshold computation to a SymPy-based symbolic oracle. The LLM translates specification requirements into symbolic expressions; the oracle performs exact arithmetic with interval propagation, ensuring every assertion is analytically traceable to its specification source.
- To ensure correctness and completeness of the generated testbenches, we further incorporate cross-validation between extracted parameters and generated code, RAG-based context retrieval for grounded generation, and closed-loop repair driven by simulation feedback. Evaluation on five industrial analog circuits demonstrates that AnalogVerifier achieves 82.3%–100% functional pass rate, validating the effectiveness of the neuro-symbolic framework for analog verification.

## 2. Preliminaries

### 2.1. Analog Circuit Verification

Analog integrated circuits, such as operational amplifiers, voltage regulators, and data converters, serve as the critical interface between digital systems and the physical world. Unlike digital circuits operating on discrete logic levels, analog designs process continuous-valued signals governed by differential equations, rendering exhaustive state-space verification infeasible.

Industrial analog verification proceeds through three stages. First, engineers interpret unstructured specification documents to extract verification tasks-testable metrics such as gain, bandwidth, and power-supply rejection ratio with associated acceptance thresholds. Second, each task is translated into executable testbench code that instantiates the device under test, sequences stimuli respecting physical constraints (supplies stabilize before bias networks; bias settles before signal paths activate), and embeds assertions

*Table 1.* **Comparison with State-of-the-Art Verification Frameworks.** Existing approaches either lack the capability to handle unstructured analog specifications or fail to enforce physical and numerical correctness. AnalogVerifier is the first to bridge the gap to continuous analog physics via a neuro-symbolic framework.

| Framework | Target Domain | Input Modality | Temporal Causality | Numerical Rigor | Closed-Loop Repair | Reasoning Paradigm |
|---|---|---|---|---|---|---|
| Template-based Scripts | Analog | Structured | Hard-coded | Hard-coded | ✗ | Rule-based |
| RTLCoder | Digital | Natural Lang. | Cycle-based | N/A | ✗ | Probabilistic |
| AutoBench / CorrectBench | Digital | RTL / Code | Discrete Event | N/A | ✓ | Agentic |
| Standard LLMs | Analog/Dig. | Natural Lang. | Hallucinated | Hallucinated | ✗ | Probabilistic |
| **AnalogVerifier (Ours)** | **Analog** | **Unstructured** | **Graph-Symbolic** | **SymPy Oracle** | ✓ | **Neuro-Symbolic** |

with specification-derived thresholds. Third, circuit simulators execute testbenches to validate design compliance. This workflow scales poorly: modern specifications define hundreds of interdependent parameters scattered across document sections, each requiring precise numerical derivation and temporally consistent stimulus construction. The cognitive burden induces systematic failures-constraint omissions, sequencing violations, and arithmetic errors-motivating automated solutions (Kundert & Chang, 2006; Chang & Kundert, 2007; 2009).

## 2.2. Problem Formulation

Let $\mathcal{S}$ denote an unstructured specification document and $\mathcal{T}$ an executable testbench. We formalize analog verification as a constrained generation problem: synthesize $\mathcal{T}$ that maximizes semantic fidelity to $\mathcal{S}$ while satisfying two physical consistency constraints.

**Port Dependency Constraint**. Analog circuits impose strict temporal ordering on stimulus activation (Chang & Kundert, 2007). Power supplies must stabilize before bias networks can operate, and bias networks must settle before signal injection can commence. These precedence relations are implicitly encoded in $\mathcal{S}$ as a directed acyclic graph $G = (V, E)$, where vertices represent circuit ports and edges encode activation dependencies. A valid testbench must produce a stimulus schedule consistent with the topological ordering of $G$.

**Numerical Grounding Constraint**. Analog performance metrics such as gain, bandwidth, and Power Supply Rejection Ratio (PSRR) are continuous quantities derived from parameter interactions. Every assertion threshold $\theta$ in $\mathcal{T}$ must be traceable to explicit numerical values in $\mathcal{S}$ via deterministic computation, thereby precluding numerically ungrounded assertions.

**Objective**. Given specification $\mathcal{S}$, we seek the optimal testbench $\mathcal{T}^*$:

$$\mathcal{T}^* = \arg\max_{\mathcal{T}} P(\mathcal{T} \mid \mathcal{S}) \quad \text{s.t.} \quad \mathcal{T} \in \mathcal{C}_{\text{dep}}(G) \cap \mathcal{C}_{\text{num}}(\mathcal{S}), \quad (1)$$

where $\mathcal{C}_{\text{dep}}(G)$ denotes the set of testbenches adhering to the topology of $G$, and $\mathcal{C}_{\text{num}}(\mathcal{S})$ denotes the set with symboli-

cally grounded thresholds. The central challenge lies in the fact that both the topological structure $G$ and the grounding logic are *latent* in $\mathcal{S}$ and must be extracted before constraint enforcement.

## 3. Related Works

**Neuro-Symbolic Integration**. Neuro-symbolic methods couple the generative flexibility of neural networks with the rigor of symbolic reasoning (Yang et al., 2025). Although such integration is often assumed to require tight end-to-end differentiable coupling, a substantial body of work instead combines the two paradigms in a modular fashion, with neural and symbolic components communicating through structured intermediate representations. Two patterns recur in this literature (Yang et al., 2025): in Neural→Symbolic systems a language model translates informal inputs into formal artifacts for a symbolic engine, whereas in Symbolic→Neural systems symbolic feedback constrains generation. AlphaGeometry (Trinh et al., 2024) pairs a neural construction proposer with a symbolic deduction engine, while Logic-LM (Pan et al., 2023) and LINC (Olausson et al., 2023) delegate inference to external solvers under LLM-driven formalization. AnalogVerifier follows this established paradigm in the analog verification domain, where the LLM performs semantic interpretation, deterministic symbolic engines enforce physical and numerical correctness, and simulation feedback is propagated back through closed-loop repair to refine generation.

**LLMs for Digital Verification**. Recent advances have shifted from static code generation (e.g., RTLCoder (Liu et al., 2024b), AssertLLM (Yan et al., 2025)) to rigorous autonomous agents. State-of-the-art frameworks like AutoBench (Qiu et al., 2024), CorrectBench (Qiu et al., 2025b), and ConfiBench (Qiu et al., 2025a) have achieved high reliability by integrating self-correction loops and confidence-based ensembles. Despite these successes, digital verification agents rely fundamentally on discrete Boolean semantics and clock-cycle determinism. They are structurally incapable of reasoning about the continuous-time physics (e.g., loop stability, transient settling) and non-linear parametric constraints inherent to analog design.

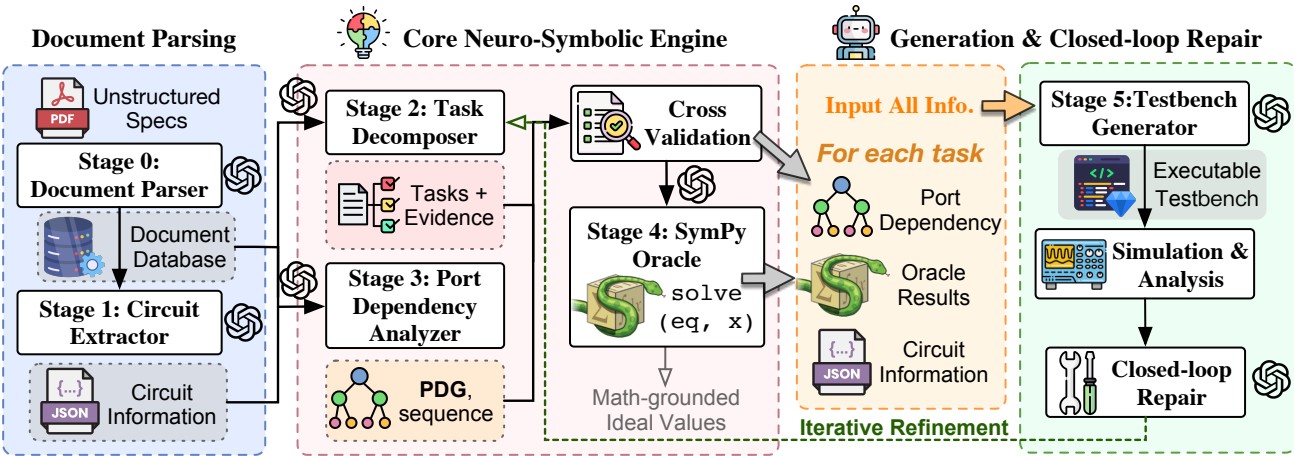

*Figure 2.* (1) **Context-Aware Task Serialization:** Transforms unstructured specs into atomic constraints via an agentic frontend (Stages 0-2). (2) **Dual-Stream Symbolic Enforcement:** Enforces physical correctness via *Graph-Symbolic Scheduling* (for temporal causality) and *Numerical-Symbolic Grounding* (for precise numerical assertions) (Stages 3-4). (3) **Testbench Generation & Repair:** Synthesizes code from verified artifacts and iteratively corrects errors via closed-loop simulation feedback (Stage 5).

Table 1 positions AnalogVerifier within the verification landscape. While digital frameworks like AutoBench introduce closed-loop agents, they operate fundamentally on discrete clock-cycle semantics, rendering them inapplicable to the continuous time-domain constraints of analog physics. Standard LLMs, while flexible, suffer from intrinsic probabilistic hallucinations in both temporal scheduling and numerical thresholds. AnalogVerifier bridges this gap by introducing a Neuro-Symbolic paradigm: leveraging the semantic plasticity of LLMs for extraction while enforcing rigor through symbolic constraints.

**LLMs for Analog Design**. Analog applications of LLMs remain nascent. The LaMAGIC series (Chang et al., 2024; 2025) employs LLMs primarily for topology generation, mapping natural language specifications directly to circuit netlists. In the domain of design automation scripts, Analog-Coder (Lai et al., 2025) and the AnalogGenie family (including AnalogGenie-Lite) (Gao et al., 2025a;b) utilize LLMs to generate Python or Verilog-A code for circuit sizing and behavioral modeling. However, a critical gap remains: no existing framework addresses automated analog verification. Unlike code generation, verification demands the rigorous enforcement of latent temporal dependencies (e.g., bias settling before signal injection) and the symbolic grounding of numerical thresholds-tasks where probabilistic LLMs inherently struggle without neuro-symbolic guardrails.

## 4. Methodologies

### 4.1. AnalogVerifier Overview

We introduce **AnalogVerifier**, a neuro-symbolic framework that automates testbench generation by decoupling semantic interpretation from logical enforcement. The workflow (Figure 2) begins with Context-Aware Task Serialization (Section 4.2), where an agentic frontend transforms unstructured specifications into atomic constraint units (Stages 0-2). This structured foundation enables Graph-Symbolic Scheduling (Section 4.3) to enforce temporal causality via PDGs, while Numerical-Symbolic Grounding (Section 4.4) simultaneously eliminates hallucinations by delegating arithmetic to a deterministic oracle. These streams converge in Testbench Generation & Repair (Section 4.5), which synthesizes executable testbenches and iteratively corrects functional errors through closed-loop simulation feedback.

### 4.2. Context-Aware Task Serialization

Industrial analog specifications suffer from *semantic dispersion*, where critical constraints are scattered across a heterogeneous mix of parametric tables, design equations, and dense prose. Monolithic LLM processing is limited by the "lost-in-the-middle" phenomenon (Liu et al., 2024a), causing the systematic omission of fine-grained parameters buried in document interiors.

**Dynamic Agentic Workflow**. To solve semantic dispersion in long-context retrieval, we propose Context-Aware Task Serialization. Unlike static parsing, this module implements a dynamic agentic workflow that transforms the linear document into a structured Document Database. This process decomposes the specification into independent units to bound neural reasoning complexity, ensuring every extraction task operates within the high-fidelity attention regime. The serialization proceeds in four steps:

*(1) Visual-Structural Segmentation.* A layout-aware parser segments the unstructured specification $\mathcal{S}$ into typed blocks $\mathcal{B} = \{b_1, \ldots, b_k\}$—including tables, equation regions, and logical sections—by utilizing spatial cues (e.g., font hierarchy, ruling lines). This step isolates local contexts, ensuring

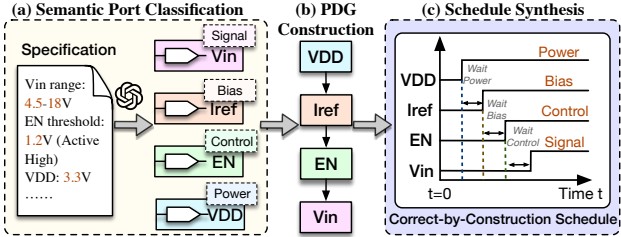

**Figure 3.** Graph-Symbolic Scheduling. (a) Semantic Classification: Unstructured pins are grounded into functional roles $\rho(p)$ (e.g., POWER, CONTROL) using LLM-extracted metadata. (b) PDG Construction: We inject dependency edges based on the domain-invariant physical hierarchy: POWER $\prec$ BIAS $\prec$ CONTROL $\prec$ SIGNAL. Edges are weighted by settling times $\delta$. (c) Schedule Synthesis: Algorithm 1 resolves the PDG into a time-optimal stimulus sequence $\sigma(p)$ via longest-path analysis, ensuring the testbench is temporally correct-by-construction.

$|b_i| \ll \kappa$ (the context window limit).

*(2) Type-Specialized Agentic Processing.* We deploy specialized agents to process these independent units. Table agents serialize parameter tuples $(\text{name}, \text{symbol}, \text{min}, \text{typ}, \text{max})$. Equation agents parse mathematical relationships into symbolic Abstract Syntax Trees (ASTs). Prose agents identify operating conditions. This effectively converts a long-context retrieval problem into a sequence of high-precision, short-context tasks.

*(3) Constraint Atomization.* Extracted information is synthesized into atomic constraints $c = (\pi, \text{op}, \theta, \mathcal{C}, \gamma)$, where $\pi$ is the target parameter, op is the relational operator, $\theta$ is the bound, $\mathcal{C}$ denotes the operating conditions, and $\gamma$ represents the source provenance.

*(4) Verification Task Decomposition.* Post-extraction, an orchestrator agent aggregates atomic constraints to formulate discrete verification tasks. By analyzing semantic correlations between parameters (e.g., associating $V_{\text{os}}$ and $I_{\text{bias}}$ with a DC analysis), the system decomposes the global specification into a set of executable objectives $\mathbf{T} = \{T_1, \ldots, T_m\}$. This step explicitly maps static constraints to dynamic simulation intents (e.g., mapping a PSRR specification to an AC analysis directive at 1kHz).

**Structured Document Database**. The serialization output is aggregated into a constraint database $\mathcal{D}$, serving as the grounding foundation for subsequent symbolic verification:

$$\mathcal{D} = \big(\mathcal{P}, \Theta, \mathcal{E}, \mathbf{T}, \Gamma_{\text{src}}\big), \tag{2}$$

where $\mathcal{P}$ is the port set, $\Theta$ is the parameter set, $\mathcal{E}$ represents the design equations, $\mathbf{T}$ denotes the decomposed verification tasks, and $\Gamma_{\text{src}}$ is the provenance registry. By serializing fragmented constraints into this structured format, we effectively eliminate the semantic dispersion barrier before testbench generation.

## 4.3. Graph-Symbolic Scheduling

While the constraint database $\mathcal{D}$ resolves semantic dispersion, it remains a collection of static, unordered requirements. The core challenge shifts to *latent dependency inference*: transforming these scattered constraints into a globally consistent stimulus schedule. Since LLMs struggle to maintain long-chain causality, we externalize this reasoning to a deterministic graph algorithm, ensuring the generated testbench is correct-by-construction.

**PDG Construction**. We formalize the scheduling problem as a weighted directed Port Dependency Graph (PDG), denoted as $G = (\mathcal{P}, E, \delta)$, constructed through a two-step process that grounds semantic understanding into physical constraints.

*(1) Semantic Port Classification.* First, we resolve the functional role of each port. Since raw pin names (e.g., "pin_3") lack semantic context, we utilize the extracted metadata to classify each port $p \in \mathcal{P}$ into a functional class $\rho(p) \in \{\text{POWER}, \text{BIAS}, \text{CONTROL}, \text{SIGNAL}\}$. This step is essential for distinguishing inputs that drive the circuit from supplies that power it.

*(2) Rule-Based Dependency Injection.* Next, we populate the dependency edges $E$ by enforcing a deterministic physical hierarchy. While explicit timing constraints are parsed directly from the specification, we resolve latent dependencies by applying a domain-invariant precedence rule: supplies must stabilize before bias networks, which must settle before control signals, followed by signal injection. We formalize this as a strict partial order:

$$\text{POWER} \prec \text{BIAS} \prec \text{CONTROL} \prec \text{SIGNAL}. \tag{3}$$

For any pair of ports $(u, v)$, if their roles satisfy $\rho(u) \prec \rho(v)$, we inject a directed edge $(u, v)$ into $E$. As shown in Figure 3, this systematically encodes the startup sequence required for physical validity.

**Schedule Synthesis**. Given the constructed PDG, we compute the optimal activation time $\sigma(p)$ for each port using Algorithm 1. The system first performs cycle detection (Line 2) to identify physically contradictory specifications. If the graph is acyclic, we determine the activation order via topological sort (Line 6). The start time is then derived by calculating the longest path from the initialization state: $\sigma(p) = \max_{(q,p)\in E}(\sigma(q) + \delta_{qp})$. This guarantees that every signal activates only after all its physical prerequisites have fully settled, minimizing simulation initialization time while strictly adhering to the causal constraints.

## 4.4. Numerical-Symbolic Grounding

While graph-symbolic scheduling ensures temporal validity, the functional correctness of a testbench hinges on precise

**Algorithm 1** Graph-Symbolic Schedule Synthesis

---

**Require:** Port Dependency Graph $G = (\mathcal{P}, E, \delta)$
**Ensure:** Schedule $\sigma : \mathcal{P} \to \mathbb{R}_{\geq 0}$ or INFEASIBLE
1:                              // Step 1: Sanity Check
2: **if** DETECTCYCLE($G$) **then**
3:    **return** INFEASIBLE (with Cycle Provenance)
4: **end if**
5:                        // Step 2: Critical Path Analysis
6: $\mathcal{L} \leftarrow$ TOPOLOGICALSORT($G$)
7: **for all** $p \in \mathcal{L}$ **do**
8:    **if** INDEGREE($p$) = 0 **then**
9:       $\sigma(p) \leftarrow 0$
10:   **else**
11:      $\sigma(p) \leftarrow \max_{(q,p) \in E} \big( \sigma(q) + \delta_{qp} \big)$
12:   **end if**
13: **end for**
14: **return** $\sigma$

---

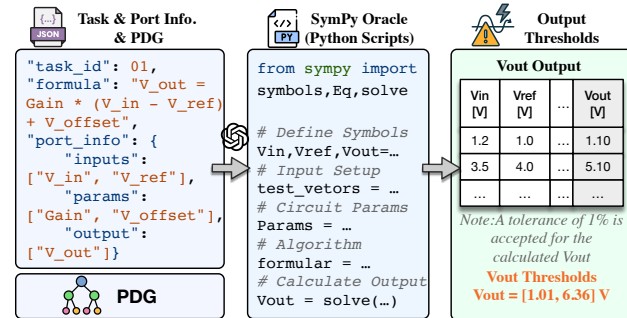

*Figure 4.* Numerical-Symbolic Grounding pipeline. The LLM translates task specifications and PDG context into a deterministic SymPy oracle. This calculates precise, tolerance-aware output thresholds, ensuring correct-by-construction assertions free from numerical hallucination.

numerical assertions. LLMs, operating as probabilistic token predictors, are prone to *Numerical Hallucination*, generating values that appear contextually plausible but lack arithmetic rigor. We overcome this by decoupling value derivation from execution: LLM translates requirements into symbolic logic, while a deterministic SymPy-based engine (Meurer et al., 2017) performs the calculation (see Figure 4).

**Task-Conditioned Symbolic Translation**. Crucially, formula generation does not occur in a vacuum. The LLM is conditioned on a structured context tuple $(T, G, \Theta, \mathcal{E})$, integrating the Verification Task $T$, the PDG $G$, and the extracted Parameters $\Theta$ and Equations $\mathcal{E}$. Instead of asking the LLM to directly predict a scalar threshold, we constrain it to translate the relevant equation $e \in \mathcal{E}$—contextualized by the topological dependencies in $G$—into a SymPy-compatible symbolic expression $f$. This expression is defined over the domain $\Theta \cup \mathcal{P}$, explicitly mapping specification parameters and graph nodes to the task's pass criteria. To mathematically preclude hallucination, we enforce a symbolic containment constraint:

$$\text{GROUNDED}(f) \iff \text{vars}(f) \subseteq (\Theta \cup \mathcal{P}), \quad (4)$$

where $\text{vars}(f)$ denotes the set of free variables in the generated expression $f$, $\Theta$ is the verified parameter set, and $\mathcal{P}$ represents the valid circuit ports from the PDG. Any expression referencing undefined variables (i.e., $\text{vars}(f) \not\subseteq \Theta \cup \mathcal{P}$) is rejected prior to execution.

**Deterministic Oracle Evaluation**. Valid expressions are handed over to the Symbolic Oracle, which executes the logic in a deterministic environment. This oracle performs two critical verification functions:

*(1) Exact Arithmetic & Unit Alignment.* The engine instantiates symbolic variables with their physical values and units extracted from the Constraint Database $\mathcal{D}$. This ensures that complex unit conversions (e.g., $\mu A \to A$) and non-linear

operations are computed with mathematical exactness, bypassing the LLM's arithmetic limitations.

*(2) Tolerance Propagation.* For parameters defined with ranges (Min/Typ/Max), the oracle employs interval arithmetic to compute robust assertion bounds.

$$\theta_{\min} = \inf_{\boldsymbol{\theta} \in \mathcal{I}(\Theta)} f(\boldsymbol{\theta}), \quad \theta_{\max} = \sup_{\boldsymbol{\theta} \in \mathcal{I}(\Theta)} f(\boldsymbol{\theta}). \quad (5)$$

By evaluating the symbolic expression $f$ over the hyper-rectangular parameter space $\mathcal{I}(\Theta)$, the system derives worst-case pass/fail criteria (e.g., `assert gain >= 59.5`) that account for manufacturing tolerances.

**Traceability and Guarantee**. This architecture achieves numerical correctness by construction. Every threshold $\theta$ in the final testbench is the result of executing a verified SymPy expression derived directly from specification parameters $\Theta$, creating an unbroken chain of traceability from the PDF source to the SPICE netlist.

### 4.5. Testbench Generation & Closed-Loop Repair

The final stage synthesizes executable testbenches by integrating the artifacts from previous stages. Unlike standard code generation that relies on raw text prompts, we implement a structured Retrieval-Augmented Generation (RAG) (Lewis et al., 2020). This module retrieves verified constraints—the database $\mathcal{D}$, schedule $\sigma$, and grounded oracles—as the context for generation, ensuring the output is physically aligned by design.

**Structured RAG for Synthesis**. We employ a modular RAG architecture where the LLM acts as a synthesizer rather than a creator. For a verification task $T$, the system constructs a dynamic prompt context by retrieving only the relevant structured artifacts:

- **Topology Context:** Retrieved from the structured document database $\mathcal{D}$, providing the Device-Under-Test (DUT) instantiation template and pin mappings.

**Algorithm 2** Simulation-Grounded Repair Loop

---

**Require:** Initial Testbench $\mathcal{T}$, PDG $G$, SymPy Oracles $\mathcal{O}$
**Ensure:** Validated Testbench $\mathcal{T}^*$ or Diagnostic Report
1:  $\mathcal{T} \leftarrow \mathcal{T}_0$
2: **for** $k = 1$ **to** $K_{\max}$ **do**
3:    $(\text{status}, \text{log}) \leftarrow \text{RUNSIMULATION}(\mathcal{T})$
4:    **if** status = PASS **then**
5:      **return** $\mathcal{T}$
6:    **end if**
7:    **if** log.type = SYNTAXERROR **then**
8:      $\mathcal{T} \leftarrow \text{LLMREPAIR}(\mathcal{T}, \text{log.error})$
9:    **else if** log.type = CONVERGENCEERROR **then**
10:     $\delta \leftarrow \delta \times 1.5$          // Relax PDG settling times
11:     $\sigma_{\text{new}} \leftarrow \text{TOPOLOGICALSORT}(G(\delta))$
12:     $\mathcal{T} \leftarrow \text{UPDATESTIMULUS}(\mathcal{T}, \sigma_{\text{new}})$
13:    **else if** log.type = ASSERTIONFAIL **then**
14:      **if** ISGROUNDED(log.$\theta, \mathcal{O}$) **then**
15:        **return** $(\mathcal{T}, \text{DESIGNVIOLATION})$   // Real bug found
16:      **else**
17:        $\mathcal{T} \leftarrow \text{REDERIVETHRESHOLD}(\mathcal{T}, \mathcal{D})$
18:      **end if**
19:    **end if**
20: **end for**
21: **return** MANUALREVIEW

---

- **Stimulus Context:** Retrieved from the Graph-Symbolic Schedule $\sigma$, providing exact timing parameters for source statements (e.g., `PULSE` delay and rise times).
- **Oracle Context:** Retrieved from the Numerical-Symbolic engine, providing the exact SymPy-derived threshold expressions.

This structured retrieval eliminates the noise of irrelevant specification details, confining the LLM's generative scope to the assembly of validated components.

**Modular Testbench Composition**. Based on the retrieved context, the testbench is composed using a template-based injection strategy to enforce separation of concerns:

*(1) Stimulus Injection.* Source configurations are deterministically mapped from the retrieved schedule $\sigma$. For a port $p$, the generator populates the SPICE syntax with graph-derived timing (e.g., $t_{\text{delay}} \leftarrow \sigma(p)$), ensuring that the physical startup sequence (POWER $\rightarrow$ CONTROL) is structurally enforced in the netlist.

*(2) Oracle-Grounded Verdicts.* Pass/fail assertions are generated by translating the retrieved SymPy oracle outputs into simulator directives. Each assertion is annotated with provenance metadata tracing back to $\Gamma_{\text{src}}$, enabling end-to-end auditability.

**Closed-Loop Refinement**. As formalized in Algorithm 2, the generated code undergoes iterative refinement. Crucially, the repair strategies are also grounded in the retrieved artifacts: convergence errors trigger a relaxation of the *Graph-Symbolic Schedule* (Line 12), while assertion failures trigger a verification against the *SymPy Oracles* (Line 15). This

closes the loop, ensuring that the final testbench is not only syntactically correct but functionally consistent with the original specification.

## 5. Experimental Results

We evaluate AnalogVerifier on industrial-grade analog circuits from three perspectives: **(RQ1)** the overall improvement over standalone LLMs, **(RQ2)** the contribution of each component to performance, and **(RQ3)** the generalization across circuit types and LLM backbones.

### 5.1. Experimental Setup

**Benchmark Circuits**. We curate a benchmark suite comprising five representative analog circuits spanning distinct functional categories: an Operational Transconductance Amplifier (OTA) for signal amplification, a DC-DC Converter (DCDC) for power management, a High-Speed Comparator (HSC) for decision circuits, a Ring Oscillator (RO) for clock generation, and a Single-to-Differential Converter (SDC) for interface circuits. Each circuit is accompanied by its industrial datasheet specification (8–32 pages) containing parametric tables, design equations, and timing constraints. The benchmark covers a diverse range of verification complexity, from simple DC operating point checks to multiphase transient analysis.

**Baselines and Configurations**. We compare three configurations to isolate the contribution of each component. The first configuration, **w/o AnalogVerifier**, represents the conventional approach where LLMs directly generate testbenches from raw specification text via few-shot prompting, without any neuro-symbolic augmentation. The second configuration, **AnalogVerifier w/o Repair**, employs Context-Aware Task Serialization, Graph-Symbolic Scheduling, and Numerical-Symbolic Grounding, but disables the closed-loop repair mechanism (Algorithm 2). The third configuration, **AnalogVerifier (Ours)**, is the full proposed framework with all components enabled. This ablation design isolates the value of structured grounding (comparing the first two configurations) and iterative refinement (comparing the latter two configurations).

**Evaluation Metrics**. We evaluate using four metrics: **#Tasks**, the number of verification tasks extracted from specification documents; **Recall**, the fraction of human-annotated reference tasks successfully identified ($|\mathbf{T}_{\text{ext}} \cap \mathbf{T}_{\text{ref}}|/|\mathbf{T}_{\text{ref}}|$); **Precision**, the fraction of testable extracted tasks matching valid references ($|\mathbf{T}_{\text{ext}} \cap \mathbf{T}_{\text{ref}}|/|\mathbf{T}_{\text{ext}}^{\text{testable}}|$); and **Pass Rate**, the percentage of generated testbenches that pass syntax validation, execute successfully in SPICE simulation, and produce semantically correct measurements.

**Implementation Details**. We evaluate three state-of-the-art

*Table 2.* Main results on testbench generation across five industrial analog specifications.

| Circuit | Model | w/o AnalogVerifier | | | | AnalogVerifier w/o Repair | | | | **AnalogVerifier (Ours)** | | | |
|---|---|---|---|---|---|---|---|---|---|---|---|---|---|
| | | #Task | Rec. | Prec. | Pass | #Task | Rec. | Prec. | Pass | #Task | Rec. | Prec. | Pass |
| OTA | GPT-5.1 | 9 | 18.8 | 33.3 | – | 23 | 62.5 | 52.6 | 30.4 | 23 | 62.5 | 52.6 | **91.3** |
| | Claude-Sonnet-4.5 | 11 | 31.2 | 45.5 | 9.1 | 29 | 87.5 | 63.6 | 37.9 | 29 | 87.5 | 63.6 | **93.1** |
| | Claude-Haiku-4.5 | 14 | 37.5 | 42.9 | – | 39 | 87.5 | 43.8 | 33.3 | 39 | 87.5 | 43.8 | **92.3** |
| DCDC | GPT-5.1 | 15 | 16.7 | 40.0 | 6.7 | 50 | 69.4 | 54.3 | 90.0 | 50 | 69.4 | 54.3 | **100.0** |
| | Claude-Sonnet-4.5 | 19 | 25.0 | 47.4 | – | 58 | 88.9 | 61.5 | 87.9 | 58 | 88.9 | 61.5 | **98.3** |
| | Claude-Haiku-4.5 | 24 | 27.8 | 41.7 | 4.2 | 84 | 88.9 | 42.7 | 69.0 | 84 | 88.9 | 42.7 | **96.4** |
| HSC | GPT-5.1 | 13 | 9.5 | 30.8 | – | 44 | 47.6 | 46.5 | 61.4 | 44 | 47.6 | 46.5 | **95.5** |
| | Claude-Sonnet-4.5 | 20 | 19.0 | 40.0 | – | 81 | 85.7 | 48.0 | 50.6 | 81 | 85.7 | 48.0 | **95.1** |
| | Claude-Haiku-4.5 | 27 | 26.2 | 40.7 | 3.7 | 118 | 95.2 | 38.8 | 28.8 | 118 | 95.2 | 38.8 | **91.5** |
| RO | GPT-5.1 | 10 | 6.5 | 20.0 | – | 27 | 16.1 | 22.7 | 40.7 | 27 | 16.1 | 22.7 | **92.6** |
| | Claude-Sonnet-4.5 | 17 | 22.6 | 41.2 | 5.9 | 74 | 93.5 | 50.0 | 73.0 | 74 | 93.5 | 50.0 | **95.9** |
| | Claude-Haiku-4.5 | 22 | 29.0 | 40.9 | – | 87 | 100.0 | 41.9 | 57.5 | 87 | 100.0 | 41.9 | **95.4** |
| SDC | GPT-5.1 | 14 | 12.5 | 28.6 | – | 96 | 43.8 | 36.8 | 42.7 | 96 | 43.8 | 36.8 | **82.3** |
| | Claude-Sonnet-4.5 | 18 | 21.9 | 38.9 | 5.6 | 91 | 87.5 | 52.8 | 59.3 | 91 | 87.5 | 52.8 | **92.3** |
| | Claude-Haiku-4.5 | 24 | 28.1 | 37.5 | – | 117 | 100.0 | 40.0 | 54.7 | 117 | 100.0 | 40.0 | **93.2** |

**Notes:** #Task = number of extracted test tasks; Rec. = task recall (%), coverage of human-annotated reliable reference tasks; Prec. = task precision (%), proportion of extracted tasks matching reliable references; **Pass = functional validation pass rate (%).** "–" indicates simulation failure. Best pass rates per row are in **bold**.

LLM backends: GPT-5.1, Claude-Sonnet-4.5, and Claude-Haiku-4.5 (OpenAI, 2025; Anthropic, 2025b;a). For the baseline configuration (w/o AnalogVerifier), we employ 3-shot prompting with manually curated exemplars. All symbolic computations use SymPy 1.12 with 64-bit floating-point precision. SPICE simulations are executed using Ngspice. The maximum refinement iterations $K_{\max}$ is set to 5. All experiments are conducted on Linux with an NVIDIA GeForce RTX 5090 (32GB).

### 5.2. Main Results

Table 2 presents results across five analog circuits with three LLM backbones.

Without AnalogVerifier, LLMs extract limited test tasks with low recall and precision. More critically, most generated testbenches fail SPICE simulation entirely, with the few successful cases achieving pass rates no higher than 9.1%.

It is noteworthy that AnalogVerifier substantially improves task extraction. Our verification framework increases extracted tasks by 2–7× with significantly higher recall. We observe a precision-recall trade-off across models: GPT-5.1 yields fewer tasks with higher precision, while Claude-Haiku-4.5 maximizes coverage at lower precision, and Claude-Sonnet-4.5 balances both. Significantly, the closed-loop repair module is critical for functional correctness. Without repair, pass rates remain suboptimal across most configurations. Our complete framework achieves 82.3%–100.0% pass rates, with the most substantial gains on circuits

such as HSC (from 28.8% to 91.5%) and OTA (from 30.4% to 91.3%).

### 5.3. Ablation Study

Table 3 reports cumulative and independent ablation results on OTA and HSC circuits, with the full system reaching 92.3 / 95.1% (OTA / HSC). Under the cumulative protocol, sequentially removing Closed-Loop Repair, Numerical-Symbolic Grounding, Graph-Symbolic Scheduling, and Context-Aware Serialization reduces the functional validation pass rate to 33.3 / 50.6%, 23.1 / 37.0%, 13.1 / 20.6%, and 7.2 / 11.3%, respectively.

Under the independent protocol, where each module is disabled in isolation and any component requiring generative reasoning is replaced by an LLM fallback performing the same role, disabling the same four modules yields 33.3 / 50.6%, 53.8 / 55.6%, 61.5 / 59.3%, and 54.2 / 57.8%, respectively.

Notably, Context-Aware Serialization shows the smallest marginal change under cumulative ablation, yet under independent ablation its absence yields a functional validation pass rate (54.2 / 57.8%) comparable to Numerical-Symbolic Grounding (53.8 / 55.6%); this reflects a known limitation of cumulative analysis, in which the contribution of an upstream module is partially masked once downstream modules have already been removed, and motivates reporting both views. Without AnalogVerifier, direct LLM generation fails entirely, so no meaningful Pass metric is reported.

*Table 3.* Ablation study on OTA (Claude-Haiku-4.5) and HSC (Claude-Sonnet-4.5) circuits.

| Configuration | OTA Pass | HSC Pass |
|---|---|---|
| *Cumulative Ablation* | | |
| **AnalogVerifier (Full)** | **92.3%** | **95.1%** |
| − Closed-Loop Repair | 33.3% | 50.6% |
| − Numerical-Symbolic Grounding | 23.1% | 37.0% |
| − Graph-Symbolic Scheduling | 13.1% | 20.6% |
| − Context-Aware Serialization | 7.2% | 11.3% |
| *Independent Ablation (LLM fallback)* | | |
| **AnalogVerifier (Full)** | **92.3%** | **95.1%** |
| w/o Closed-Loop Repair | 33.3% | 50.6% |
| w/o Numerical-Symbolic Grounding | 53.8% | 55.6% |
| w/o Graph-Symbolic Scheduling | 61.5% | 59.3% |
| w/o Context-Aware Serialization | 54.2% | 57.8% |
| w/o AnalogVerifier (Baseline) | – | – |

**Notes:** Cumulative ablation progressively removes modules from the full system. Independent ablation disables each module individually: modules that involve LLM-based reasoning are replaced with an LLM fallback performing the same role, while Closed-Loop Repair is simply turned off.

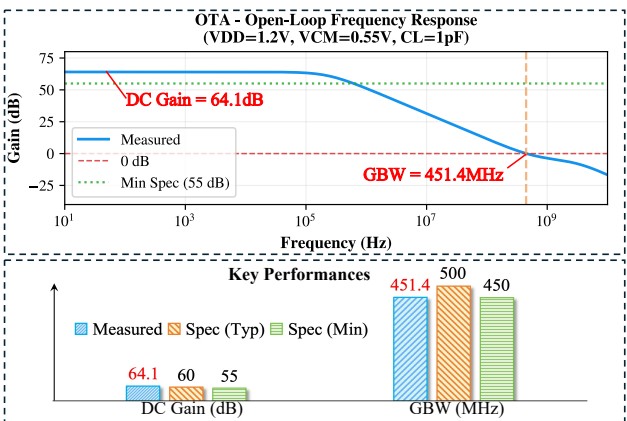

*Figure 5.* OTA open-loop frequency response and key performance metrics. Top: Bode plot showing DC gain of 64.1 dB and GBW of 451.4 MHz. Bottom: Comparison between measured values and specification targets on key performance (DC Gain and GBW).

### 5.4. Case Study: OTA Verification

Figure 5 shows the verification results for the OTA design. The framework automatically extracts 39 verification tasks from the specification, of which 20 are retained after LLM review for testbench generation. The generated testbenches achieve a 69.0% first-pass validation rate, which improves to 96.4% after closed-loop repair. Key metrics including DC gain (64.1 dB) and GBW (451.4 MHz) are automatically measured and compared against specifications, demonstrating the framework's capability to handle real-world analog circuit verification.

## 6. Conclusion

We present AnalogVerifier, a neuro-symbolic framework that automates analog testbench generation by decoupling semantic translation from logical enforcement. Through Context-Aware Serialization, Graph-Symbolic Scheduling, Numerical-Symbolic Grounding, and Closed-Loop Repair, our approach bridges the gap between LLM flexibility and the strict determinism required for analog verification. Evaluation on five industrial circuits demonstrates 82.3%–100% functional pass rates, significantly reducing the manual effort that dominates current analog verification workflows. Future work includes extending to mixed-signal systems and incorporating formal equivalence checking.

## Acknowledgment

The research work described in this paper was conducted in the JC STEM Lab of Intelligent Design Automation funded by The Hong Kong Jockey Club Charities Trust. This work is jointly supported by the Research Grants Council of Hong Kong SAR (No. CUHK14207523).

## Impact Statement

This paper presents work whose goal is to advance the field of Machine Learning. There are many potential societal consequences of our work, none which we feel must be specifically highlighted here.

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

## A. Stage 0: Document Parsing and Block-Level Semantic Analysis

This appendix provides comprehensive details on Stage 0 of our datasheet-to-test-specification pipeline, which performs document parsing and block-level semantic extraction from IC datasheets.

### A.1. Overview and Objectives

Stage 0 serves as the foundation of our pipeline, transforming raw PDF datasheets into a structured, semantically-enriched document database. The primary objectives are:

1. **Document Segmentation**: Partition the datasheet into semantically coherent blocks (text paragraphs, tables, figures, equations).
2. **Block Classification**: Classify each block into one of 12 predefined categories based on its semantic content.
3. **Parameter Extraction**: Identify and extract electrical parameters with their specifications (min/typ/max values, units, test conditions).
4. **Equation Parsing**: Extract mathematical relationships and convert them to multiple representations (raw, LaTeX, Python).
5. **Pin Definition Extraction**: Identify pin names, numbers, types, and connection requirements.
6. **Cross-Reference Mapping**: Establish relationships between blocks referencing figures, tables, equations, and other sections.
7. **Semantic Insight Generation**: Produce design implications and potential test scenarios for each block.

### A.2. Block Type Taxonomy

We define 12 block types to capture the diverse content structures found in IC datasheets:

*Table 4.* Block Type Taxonomy for Stage 0 Classification

| Block Type | Description |
|---|---|
| section_header | Document structure markers (e.g., "5. Absolute Maximum Ratings") |
| feature_list | Bulleted or enumerated device features and capabilities |
| pin_definition | Pin assignment tables and functional pin descriptions |
| electrical_characteristics | Parametric specifications with min/typ/max values |
| absolute_max_ratings | Stress limits beyond which device damage may occur |
| application_info | Circuit design guidance, typical applications, component selection |
| equation | Mathematical formulas and design equations |
| table | Structured tabular data (specifications, mappings, conditions) |
| text_paragraph | Descriptive text explaining device operation or characteristics |
| figure_reference | References to diagrams, schematics, or graphical content |
| timing_diagram | Timing specifications and waveform descriptions |
| other | Metadata, footers, and non-technical content |

### A.3. Prompt Engineering

We employ a carefully designed system prompt to guide the LLM in performing semantic extraction. The complete prompt is provided below:

**STAGE 0 SYSTEM PROMPT**

```
You are a senior semiconductor test engineer with 20+ years of
experience in analog/mixed-signal IC characterization and ATE
programming. Your task is to perform deep semantic analysis of
datasheet content blocks.

For each block, you must:

1. BLOCK CLASSIFICATION
   - Identify the block_type from: {section_header, feature_list,
     pin_definition, electrical_characteristics, absolute_max_ratings,
```

```
      application_info, equation, table, text_paragraph,
      figure_reference, timing_diagram, other}
   - Provide confidence score (0.0-1.0) with reasoning

2. PARAMETER EXTRACTION
   For each electrical parameter identified:
   - name: Human-readable parameter name
   - symbol: Standard engineering symbol (e.g., VIN, ILIMIT)
   - description: Detailed functional description
   - value_spec: {min, typ, max, unit}
   - test_conditions: Exact conditions under which spec applies
   - category: {VOLTAGE, CURRENT, RESISTANCE, TIMING, FREQUENCY,
                TEMPERATURE, POWER, OTHER}
   - is_testable: Boolean indicating if parameter can be verified
   - related_pins: List of associated pin names
   - source_context: Original text from which parameter was extracted

3. EQUATION EXTRACTION
   For each mathematical relationship:
   - equation_id: Unique identifier
   - raw_form: Original equation as written
   - latex_form: LaTeX representation for rendering
   - python_form: Python-executable expression
   - description: What the equation calculates
   - output_variable: {symbol, description, unit}
   - input_variables: [{symbol, description, typical_range, unit}, ...]
   - application: How this equation is used in design/test

4. PIN EXTRACTION
   For each pin referenced:
   - pin_name: Functional name (e.g., "VIN", "EN", "FB")
   - pin_numbers: {package_type: pin_number, ...}
   - pin_type: Functional classification
   - direction: {INPUT, OUTPUT, BIDIRECTIONAL, POWER, GROUND}
   - description: Functional description
   - electrical_constraints: {voltage_range, current_limit}
   - connection_requirements: Application guidance

5. CROSS-REFERENCES
   Identify all references to:
   - Other sections (by number or title)
   - Figures (by figure number)
   - Tables (by table number)
   - Equations (by equation number)

6. SEMANTIC INSIGHTS
   Generate:
   - key_points: Critical information for understanding block content
   - design_implications: How this affects circuit design decisions
   - potential_test_scenarios: Specific tests to validate specifications

Output format: JSON following the schema provided.
```

## A.4. Input-Output Schema

### A.4.1. INPUT SCHEMA

Each block is presented to the model with the following structure:

*Code Listing 1.* Stage 0 Input Block Schema

```
{
  "block_id": "page<N>_block<M>",
  "page_number": <integer>,
  "raw_text": "<extracted text content>",
```

```
    "preceding_context": "<previous 2-3 blocks for context>",
  "document_metadata": {
      "product_name": "<IC part numbers>",
      "document_type": "<datasheet type>",
      "version": "<revision number>"
  }
}
```

### A.4.2. OUTPUT SCHEMA

The model produces structured output conforming to:

*Code Listing 2.* Stage 0 Output Block Schema

```
{
  "block_id": "<inherited from input>",
  "page_number": <integer>,
  "raw_text": "<inherited from input>",
  "block_type": "<one of 12 types>",
  "structured_content": {
    "block_analysis": {
      "block_type": "<classified type>",
      "content_summary": "<brief description>",
      "confidence": <0.0-1.0>,
      "reasoning": "<classification justification>"
    },
    "extracted_parameters": [<Parameter objects>],
    "extracted_equations": [<Equation objects>],
    "extracted_pins": [<Pin objects>],
    "table_structure": {
      "table_type": "<specification type>",
      "headers": [<column headers>],
      "row_count": <integer>,
      "has_conditions": <boolean>,
      "global_conditions": "<applicable conditions>"
    },
    "cross_references": {
      "referenced_sections": [<section identifiers>],
      "referenced_figures": [<figure numbers>],
      "referenced_tables": [<table numbers>],
      "referenced_equations": [<equation numbers>]
    },
    "semantic_insights": {
      "key_points": [<critical observations>],
      "design_implications": [<engineering guidance>],
      "potential_test_scenarios": [<validation approaches>]
    }
  },
  "parameters": [<flattened parameter list>],
  "equations": [<flattened equation list>],
  "pins": [<flattened pin list>],
  "cross_references": [<flattened reference list>],
  "confidence": <overall confidence score>,
  "image_data": <base64 or null>
}
```

## A.5. Detailed Data Structures

### A.5.1. PARAMETER OBJECT STRUCTURE

*Code Listing 3.* Parameter Object Schema

```
{
```

```
  "name": "Valley Current Limit",
  "symbol": "ILIMIT",
  "description": "Maximum inductor current limit threshold...",
  "value_spec": {
    "min": 2.0,
    "typ": 2.5,
    "max": 3.0,
    "unit": "A"
  },
  "test_conditions": "DCDC variant, VIN = 12V, TJ = 25C",
  "category": "CURRENT",
  "is_testable": true,
  "related_pins": ["SW", "ILIM"],
  "source_context": "Valley Current Limit, ILIMIT, 2, 2.5, 3, A"
}
```

## A.5.2. EQUATION OBJECT STRUCTURE

*Code Listing 4.* Equation Object Schema

```
{
  "equation_id": "EQ_THERMAL_POWER_DISSIPATION",
  "raw_form": "PD(MAX) = (TJ(MAX) - TA) / theta_JA",
  "latex_form": "P_{D(MAX)} = \\frac{T_{J(MAX)} - T_A}{\\theta_{JA}}",
  "python_form": "PD_MAX = (TJ_MAX - TA) / theta_JA",
  "description": "Maximum allowable power dissipation calculation",
  "output_variable": {
    "symbol": "PD(MAX)",
    "description": "Maximum power dissipation",
    "unit": "W"
  },
  "input_variables": [
    {"symbol": "TJ(MAX)", "description": "Max junction temp",
     "typical_range": "125-175", "unit": "C"},
    {"symbol": "TA", "description": "Ambient temperature",
     "typical_range": "-40 to 85", "unit": "C"},
    {"symbol": "theta_JA", "description": "Thermal resistance",
     "typical_range": "50-200", "unit": "C/W"}
  ],
  "application": "Thermal design and reliability analysis"
}
```

## A.5.3. PIN OBJECT STRUCTURE

*Code Listing 5.* Pin Object Schema

```
{
  "pin_name": "EN",
  "pin_numbers": {
    "SOT563": "5",
    "TSOT23-6_Type_A": "4",
    "TSOT23-6_Type_B": "5"
  },
  "pin_type": "Enable Control",
  "direction": "INPUT",
  "description": "Enable input control signal. Active high logic.",
  "electrical_constraints": {
    "voltage_range": {"min": 0, "max": "VIN", "unit": "V"},
    "current_limit": {"value": 2, "unit": "uA"}
  },
  "connection_requirements": "Connect to VIN for auto start-up;
                              active high logic"
}
```

## A.6. Processing Pipeline

The Stage 0 processing pipeline consists of the following steps:

---

**Algorithm 3** Stage 0: Block-Level Semantic Extraction

---

**Require:** PDF datasheet $D$, LLM model $\mathcal{M}$
**Ensure:** Structured document database $\mathcal{DB}$
1: $\mathcal{B} \leftarrow$ EXTRACTBLOCKS($D$)         // PDF parsing
2: $\mathcal{DB} \leftarrow \emptyset$
3: $context \leftarrow \emptyset$
4: **for all** $b_i \in \mathcal{B}$ **do**
5:    $input \leftarrow$ PREPAREINPUT($b_i, context, D.metadata$)
6:    $output \leftarrow \mathcal{M}$.GENERATE($input$, system_prompt)
7:    $parsed \leftarrow$ PARSEJSON($output$)
8:    $validated \leftarrow$ VALIDATESCHEMA($parsed$)
9:    $\mathcal{DB}$.APPEND($validated$)
10:   $context$.UPDATE($b_i, validated$)         // Sliding window
11: **end for**
12: **return** $\mathcal{DB}$

---

## A.7. Example Extractions

We provide representative examples of Stage 0 extractions from an industrial DC-DC converter datasheet.

### A.7.1. EXAMPLE 1: ELECTRICAL CHARACTERISTICS TABLE

**Input (Raw Text):**

```
Table 7:  Electrical Characteristics
Parameter | Symbol | Condition | Min | Typ | Max | Unit
Supply Current (Shutdown) | IIN | VEN = 0V | - | - | 10 | uA
Supply Current (Quiescent) | Iq | VEN = 2V, VFB = 1V | - | 0.25 | - | mA
HS Switch-On Resistance | RDS(ON)HS | VBST-SW = 5V, dcdc | - | 120 | - | mOhm
HS Switch-On Resistance | RDS(ON)HS | VBST-SW = 5V, dcdc | - | 80 | - | mOhm
```

**Output (Extracted Parameters):**

```
[
  {
    "name": "Supply Current (Shutdown)",
    "symbol": "IIN",
    "value_spec": {"min": null, "typ": null, "max": 10, "unit": "uA"},
    "test_conditions": "VEN = 0V",
    "category": "CURRENT",
    "is_testable": true,
    "related_pins": ["VEN", "VCC"]
  },
  {
    "name": "HS Switch-On Resistance",
    "symbol": "RDS(ON)HS",
    "value_spec": {"min": null, "typ": 120, "max": null, "unit": "mOhm"},
    "test_conditions": "VBST-SW = 5V,  variant",
    "category": "RESISTANCE",
    "is_testable": true,
    "related_pins": ["VBST", "SW"]
  }
]
```

### A.7.2. EXAMPLE 2: PIN ASSIGNMENT TABLE

**Input (Raw Text):**

```
Table 1:  Pin Assignment for Different Packages
SOT563 | TSOT23-6 Type A | TSOT23-6 Type B | Pin Name
1 | 5 | 3 | IN
2 | 6 | 2 | SW
```

```
3 | 2 | 1 | GND
4 | 1 | 6 | BST
5 | 4 | 5 | EN
6 | 3 | 4 | FB
```

**Output (Extracted Pins):**

```
[
  {
    "pin_name": "IN",
    "pin_numbers": {"SOT563": "1", "TSOT23-6_Type_A": "5", "TSOT23-6_Type_B": "3"},
    "pin_type": "Power Supply Input",
    "direction": "INPUT",
    "description": "Input voltage supply pin"
  },
  {
    "pin_name": "SW",
    "pin_numbers": {"SOT563": "2", "TSOT23-6_Type_A": "6", "TSOT23-6_Type_B": "2"},
    "pin_type": "Switch Node",
    "direction": "OUTPUT",
    "description": "Switching output connecting to inductor"
  }
]
```

### A.7.3. EXAMPLE 3: DESIGN EQUATION

**Input (Raw Text):**

```
The maximum allowable power dissipation is calculated by:
PD(MAX) = (TJ(MAX) - TA) / theta_JA
```

**Output (Extracted Equation):**

```
{
  "equation_id": "EQ_THERMAL_PD",
  "raw_form": "PD(MAX) = (TJ(MAX) - TA) / theta_JA",
  "latex_form": "P_{D(MAX)} = \\frac{T_{J(MAX)} - T_A}{\\theta_{JA}}",
  "python_form": "PD_MAX = (TJ_MAX - TA) / theta_JA",
  "output_variable": {"symbol": "PD(MAX)", "unit": "W"},
  "input_variables": [
    {"symbol": "TJ(MAX)", "unit": "C"},
    {"symbol": "TA", "unit": "C"},
    {"symbol": "theta_JA", "unit": "C/W"}
  ]
}
```

## A.8. Quality Metrics and Validation

We evaluate Stage 0 extraction quality using the following metrics:

*Table 5.* Stage 0 Extraction Quality Metrics

| Metric | Definition | Target |
|---|---|---|
| Block Classification Accuracy | $\frac{\text{\# correctly classified}}{\text{\# total blocks}}$ | $\geq 0.95$ |
| Parameter Extraction Recall | $\frac{\text{\# extracted params}}{\text{\# ground truth params}}$ | $\geq 0.90$ |
| Parameter Value Accuracy | $\frac{\text{\# correct values}}{\text{\# extracted params}}$ | $\geq 0.98$ |
| Equation Parse Success Rate | $\frac{\text{\# valid equations}}{\text{\# equation blocks}}$ | $\geq 0.85$ |
| Pin Mapping Completeness | $\frac{\text{\# complete pin entries}}{\text{\# total pins}}$ | $\geq 0.95$ |
| Cross-Reference Resolution | $\frac{\text{\# resolved refs}}{\text{\# total refs}}$ | $\geq 0.80$ |

## A.9. Implementation Details

- **PDF Parsing**: We employ PyMuPDF (fitz) for text extraction with layout preservation, combined with pdfplumber for table detection.
- **Block Segmentation**: Blocks are segmented based on vertical spacing, font changes, and structural markers (section numbers and bullet points).

- **Context Window**: A sliding window of three preceding blocks provides contextual information for each extraction.
- **Model Configuration**: Temperature is set to 0.1 for deterministic extraction; maximum tokens is 4096 per block.
- **Retry Logic**: Failed JSON parses trigger up to three retries with incremental prompt refinement.
- **Schema Validation**: All outputs are validated against JSON Schema prior to database insertion.

## A.10. Limitations and Error Handling

Stage 0 has the following known limitations:

1. **Image-Only Content**: Blocks containing only images (e.g., block diagrams, timing diagrams) cannot be processed without OCR or vision models.
2. **Complex Table Layouts**: Multi-level headers and merged cells may result in incomplete extraction.
3. **Implicit Parameters**: Parameters referenced but not explicitly stated require cross-block inference (handled in Stage 1).
4. **Unit Standardization**: Non-standard unit notations (e.g., "mohm" vs "m$\Omega$") require normalization.

Error handling strategies:

- Missing values are represented as `null` rather than omitted
- Low-confidence extractions ($< 0.7$) are flagged for manual review
- Ambiguous block types default to `text_paragraph` with explanation

## A.11. Output Statistics

For the example datasheet (Industrial DCDC, 13 pages), Stage 0 produces:

*Table 6.* Stage 0 Output Statistics for Example Datasheet

| Metric | Value |
|---|---|
| Total Blocks Processed | 89 |
| Unique Parameters Extracted | 147 |
| Unique Equations Extracted | 8 |
| Unique Pins Defined | 6 |
| Cross-References Identified | 43 |
| Average Confidence Score | 0.94 |
| Processing Time (seconds) | 127 |
| Output JSON Size (KB) | 412 |

## A.12. Block Type Distribution

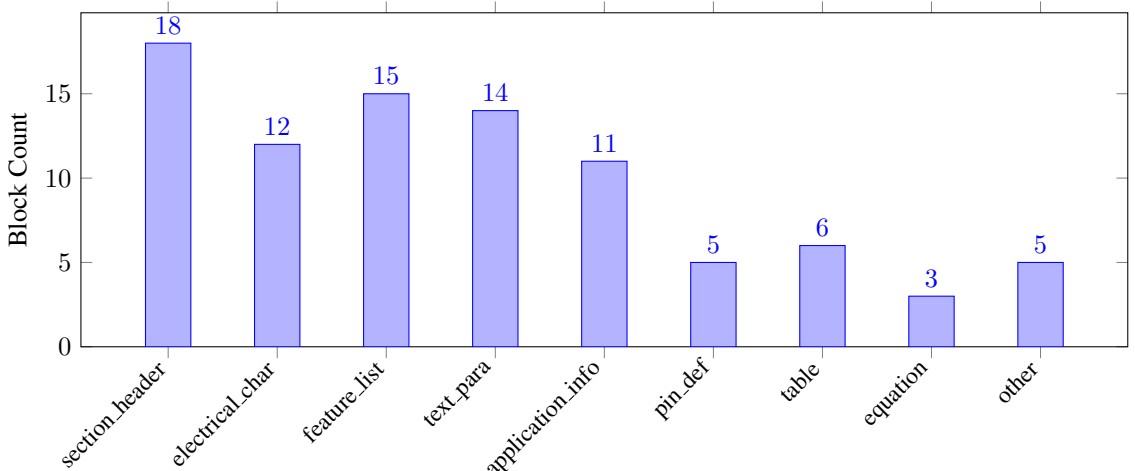

*Figure 6.* Distribution of block types in example datasheet extraction

# B. Stage 1: Circuit Information Extraction

Stage 1 is responsible for extracting structured circuit information from the parsed document database produced by Stage 0. This stage transforms raw textual and tabular content into a unified, schema-compliant representation of the circuit's electrical characteristics, pin definitions, and mathematical relationships.

## B.1. Design Objectives

The primary objectives of Stage 1 are:

1. **Semantic Understanding**: Leverage LLM capabilities to understand the semantic meaning of electrical parameters, rather than relying on brittle pattern-matching heuristics.
2. **Unified Schema**: Produce a standardized `CircuitInformation` structure that serves as the foundation for all downstream stages.
3. **Cross-Reference Resolution**: Resolve implicit references between parameters, pins, and equations scattered across different document blocks.
4. **Ambiguity Handling**: Explicitly flag ambiguous or incomplete information rather than making potentially incorrect inferences.

## B.2. Input and Output Specification

**Input.** Stage 1 receives the `DocumentDatabase` structure from Stage 0, containing:

- Parsed document blocks with block-level semantic annotations
- Extracted parameters, equations, and pin definitions (preliminary)
- Global symbol table with cross-document references
- Document metadata (source path, page count, extraction statistics)

**Output.** Stage 1 produces a `CircuitInformation` structure as formally defined in Section B.3.

## B.3. Data Structure Definitions

We define the core data structures using a typed schema notation. All structures are serialized as JSON for interoperability.

*Code Listing 6.* Core data structures for Stage 1 output.

```python
@dataclass
class CircuitInformation:
    circuit_name: str           # e.g., "Industrial_DCDC"
    circuit_type: str           # e.g., "DCDC_BUCK"
    circuit_description: str     # Natural language description
    key_features: List[str]      # Extracted feature list
    typical_applications: List[str]
    pins: List[PinDefinition]
    parameters: List[ElectricalParameter]
    equations: List[EquationDefinition]
    operating_conditions: OperatingConditions
    absolute_max_ratings: Dict[str, Any]
    extraction_confidence: float   # Overall confidence score
    extraction_notes: List[str]    # Warnings and ambiguities

@dataclass
class PinDefinition:
    pin_name: str               # e.g., "SW", "FB", "EN"
    pin_number: Dict[str, str]   # Package -> pin number mapping
    pin_type: str               # POWER | GROUND | ANALOG | DIGITAL | MIXED
    direction: str              # INPUT | OUTPUT | BIDIRECTIONAL | POWER | GROUND
    description: str            # Functional description
    electrical_constraints: Dict[str, Any]
    inferred_role: str          # STIMULUS | OUTPUT | CONTROL | FEEDBACK | REFERENCE
    source_block: str           # Traceability to source document

@dataclass
class ElectricalParameter:
    symbol: str                 # e.g., "VIN", "RDS_ON_HS"
    name: str                   # Human-readable name
    description: str            # Detailed description
    min_value: Optional[float]
```

```
    typ_value: Optional[Union[float, List[float]]]
    max_value: Optional[float]
    unit: str
    test_conditions: List[str]
    category: str                   # VOLTAGE | CURRENT | RESISTANCE | TIMING | ...
    is_testable: bool               # Whether parameter can be verified
    related_pins: List[str]         # Associated pin names
    test_type_hint: str             # Suggested test methodology
    source_block: str               # Traceability

@dataclass
class EquationDefinition:
    equation_id: str                # Unique identifier
    raw_form: str                   # Original textual form
    sympy_form: str                 # SymPy-executable expression
    latex_form: str                 # LaTeX rendering
    description: str
    output_variable: str
    input_variables: List[VariableSpec]
    application: str                # When to use this equation
    validity_conditions: str
    source_block: str
```

## B.4. Extraction Algorithm

The Stage 1 extraction process consists of four sequential phases, each leveraging targeted LLM prompts. Algorithm 4 presents the high-level procedure.

---

**Algorithm 4** Stage 1: Circuit Information Extraction

---

**Require:** Document database $\mathcal{D}$ from Stage 0, LLM model $\mathcal{M}$
**Ensure:** Structured circuit information $\mathcal{C}$
1: **Phase 1: Circuit Identification**
2: $\text{context}_{\text{global}} \leftarrow$ EXTRACTGLOBALCONTEXT$(\mathcal{D})$
3: $(c_{\text{name}}, c_{\text{type}}, c_{\text{desc}}) \leftarrow \mathcal{M}.$IDENTIFY$(\text{context}_{\text{global}}, \text{CIRCUITIDPROMPT})$
4:
5: **Phase 2: Pin Extraction and Role Inference**
6: $\mathcal{B}_{\text{pin}} \leftarrow \{b \in \mathcal{D}.\text{blocks} : b.\text{type} \in \{\text{PINDEF}, \text{TABLE}\}\}$
7: $\mathcal{P}_{\text{raw}} \leftarrow \bigcup_{b \in \mathcal{B}_{\text{pin}}} b.\text{pins}$
8: $\mathcal{P} \leftarrow \mathcal{M}.$REFINEANDINFER$(\mathcal{P}_{\text{raw}}, c_{\text{type}}, \text{PINROLEPROMPT})$
9:
10: **Phase 3: Parameter Consolidation**
11: $\mathcal{E}_{\text{raw}} \leftarrow \bigcup_{b \in \mathcal{D}.\text{blocks}} b.\text{parameters}$
12: $\mathcal{E}_{\text{dedup}} \leftarrow$ DEDUPLICATEPARAMS$(\mathcal{E}_{\text{raw}})$
13: $\mathcal{E} \leftarrow \mathcal{M}.$ENRICHPARAMS$(\mathcal{E}_{\text{dedup}}, \mathcal{P}, \text{PARAMENRICHPROMPT})$
14:
15: **Phase 4: Equation Formalization**
16: $\mathcal{Q}_{\text{raw}} \leftarrow \bigcup_{b \in \mathcal{D}.\text{blocks}} b.\text{equations}$
17: $\mathcal{Q} \leftarrow \mathcal{M}.$FORMALIZE$(\mathcal{Q}_{\text{raw}}, \mathcal{E}, \text{EQUATIONPROMPT})$
18:
19: **Phase 5: Assembly and Validation**
20: $\mathcal{C} \leftarrow$ ASSEMBLECIRCUITINFO$(c_{\text{name}}, c_{\text{type}}, c_{\text{desc}}, \mathcal{P}, \mathcal{E}, \mathcal{Q})$
21: $\mathcal{C}.\text{confidence} \leftarrow$ COMPUTECONFIDENCE$(\mathcal{C})$
22: **return** $\mathcal{C}$

---

**Complexity Analysis.** Let $n$ denote the number of document blocks, $p$ denote the number of pins, and $e$ denote the number of parameters. The algorithm requires $O(1)$ LLM calls for circuit identification, $O(1)$ calls for pin refinement (batched), $O(\lceil e/B \rceil)$ calls for parameter enrichment with batch size $B$, and $O(1)$ calls for equation formalization. The total LLM invocation complexity is $O(e/B)$, which scales linearly with the number of parameters.

## B.5. Prompt Engineering

We employ a hierarchical prompt design with task-specific templates. Each prompt follows our *LLM-First Methodology*, which explicitly instructs the model to perform semantic reasoning rather than pattern matching.

## B.5.1. CIRCUIT IDENTIFICATION PROMPT

**CIRCUITIDENTIFICATIONPROMPT**

```
# Task: Circuit Type Identification and Characterization

You are an expert analog/mixed-signal IC engineer. Analyze the provided
document content to identify the circuit type and extract key characteristics.

## Document Content Summary
{document_summary}

## First Page Content
{first_page_content}

## Feature Lists Detected
{feature_lists}

## Instructions
1. Identify the PRIMARY circuit type from: OTA, LDO, BUCK, BOOST, BUCK_BOOST,
   USB_TYPEC, ADC, DAC, PLL, COMPARATOR, REFERENCE, AMPLIFIER, OTHER
2. Extract the official product name/part number
3. Provide a comprehensive technical description (2-3 sentences)
4. List key features as bullet points
5. Identify typical applications

## Critical: Use SEMANTIC understanding, not keyword matching
- "step-down converter" -> DCDC_BUCK (not just matching "buck")
- "linear regulator" -> LDO (understand function, not just acronym)
- Consider the COMPLETE functionality described

## Output JSON Schema
{
  "circuit_name": "string",
  "circuit_type": "string (from predefined list)",
  "circuit_type_confidence": 0.0-1.0,
  "circuit_type_reasoning": "string explaining classification",
  "circuit_description": "string (technical summary)",
  "key_features": ["feature1", "feature2", ...],
  "typical_applications": ["app1", "app2", ...],
  "alternative_classifications": [
    {"type": "string", "confidence": 0.0-1.0, "reasoning": "string"}
  ]
}
```

## B.5.2. PIN ROLE INFERENCE PROMPT

**PINROLEINFERENCEPROMPT**

```
# Task: Pin Definition Refinement and Role Inference

You are an expert IC design engineer. For each pin, determine its functional
role in the context of a {circuit_type} circuit.

## Circuit Context
- Name: {circuit_name}
- Type: {circuit_type}
- Description: {circuit_description}

## Raw Pin Definitions
{pin_definitions_json}

## Instructions for Each Pin
1. **Classify pin_type**: POWER | GROUND | ANALOG | DIGITAL | MIXED
2. **Determine direction**: INPUT | OUTPUT | BIDIRECTIONAL | POWER | GROUND
3. **Infer functional role**:
   - STIMULUS: Primary signal input for testing
   - OUTPUT: Primary signal output for measurement
   - CONTROL: Enable, mode selection, configuration
   - FEEDBACK: Regulation feedback path
   - REFERENCE: Voltage/current reference
   - BIAS: Biasing input
   - POWER: Power supply
   - GROUND: Ground reference

## Role Inference Guidelines for {circuit_type}
```

```
{circuit_specific_guidelines}

## Output JSON Schema
{
  "refined_pins": [
    {
      "pin_name": "string",
      "pin_number": {"package": "number"},
      "pin_type": "POWER|GROUND|ANALOG|DIGITAL|MIXED",
      "direction": "INPUT|OUTPUT|BIDIRECTIONAL|POWER|GROUND",
      "description": "string (enhanced description)",
      "electrical_constraints": {
        "absolute_max_voltage": null or float,
        "operating_voltage_range": {"min": float, "max": float},
        "current_capability": null or float
      },
      "inferred_role": "STIMULUS|OUTPUT|CONTROL|FEEDBACK|REFERENCE|BIAS|POWER|GROUND",
      "role_confidence": 0.0-1.0,
      "role_reasoning": "string"
    }
  ]
}
```

## B.5.3. PARAMETER ENRICHMENT PROMPT

**PARAMETERENRICHMENTPROMPT**

```
# Task: Electrical Parameter Enrichment and Classification

Enrich the following electrical parameters with semantic understanding.

## Circuit Context
- Type: {circuit_type}
- Available Pins: {pin_names}

## Parameters to Enrich (Batch {batch_id}/{total_batches})
{parameters_json}

## For Each Parameter, Determine:

1. **Category Classification**:
   VOLTAGE | CURRENT | RESISTANCE | CAPACITANCE | INDUCTANCE |
   FREQUENCY | TIMING | TEMPERATURE | POWER | EFFICIENCY | GAIN | OTHER

2. **Testability Assessment**:
   - is_testable: Can this be directly measured in a test environment?
   - test_type_hint: Suggested test methodology
     DC_VOLTAGE | DC_CURRENT | AC_GAIN | AC_BANDWIDTH | AC_PHASE |
     TRANSIENT_TIMING | TRANSIENT_LOAD | PROTECTION | EFFICIENCY | NOISE | OTHER

3. **Pin Association**:
   - Which pins are involved in measuring/stimulating this parameter?
   - Use semantic understanding (e.g., "input offset voltage" relates to
     differential input pins, not just pins with "IN" in the name)

4. **Value Normalization**:
   - Convert all values to SI base units where applicable
   - Handle ranges, typical values, and conditions

## Output JSON Schema
{
  "enriched_parameters": [
    {
      "symbol": "string",
      "name": "string",
      "description": "string (enhanced)",
      "min_value": null or float,
      "typ_value": null or float or [float, float],
      "max_value": null or float,
      "unit": "string (SI preferred)",
      "test_conditions": ["condition1", "condition2"],
      "category": "string",
      "is_testable": boolean,
      "related_pins": ["pin1", "pin2"],
      "test_type_hint": "string",
      "enrichment_notes": "string (any ambiguities or assumptions)"
    }
```

```
    ]
}
```

## B.5.4. EQUATION FORMALIZATION PROMPT

---

**EQUATIONFORMALIZATIONPROMPT**

```
# Task: Equation Extraction and Formalization

Convert extracted equations into executable SymPy expressions.

## Raw Equations
{equations_json}

## Available Parameters (for variable resolution)
{parameters_summary}

## SymPy Syntax Requirements (CRITICAL)
- Use `*` for multiplication (NOT implicit): `2*x` not `2x`
- Use `**` for exponentiation: `x**2` not `x^2`
- Use `sqrt(x)` for square root
- Use `log(x)` for natural logarithm, `log(x, 10)` for base-10
- Use `pi` for    , `E` for Euler's number
- All variable names must be valid Python identifiers
- Greek letters: use English names (theta, omega, etc.)

## For Each Equation, Provide:
1. **equation_id**: Unique identifier (e.g., "EQ_THERMAL_001")
2. **raw_form**: Original text representation
3. **sympy_form**: SymPy-executable string
4. **latex_form**: LaTeX rendering for documentation
5. **Variable specification**: Input/output variables with units

## Output JSON Schema
{
  "formalized_equations": [
    {
      "equation_id": "string",
      "raw_form": "string",
      "sympy_form": "string (MUST be valid SymPy)",
      "latex_form": "string",
      "description": "string",
      "output_variable": "string",
      "input_variables": [
        {
          "symbol": "string",
          "name": "string",
          "unit": "string",
          "how_to_obtain": "FROM_SPEC|MEASURED|CALCULATED|CONSTANT",
          "typical_value": "string or null"
        }
      ],
      "application": "string (when to use)",
      "validity_conditions": "string"
    }
  ],
  "parsing_failures": [
    {"raw_form": "string", "reason": "string"}
  ]
}
```

---

## B.6. Deduplication and Conflict Resolution

Parameters extracted from different document blocks may contain duplicates or conflicting values. We employ a hierarchical conflict resolution strategy:

1. **Exact Match**: Parameters with identical symbols and units are merged, preferring values from electrical characteristics tables over feature descriptions.

2. **Symbol Normalization**: Common variations are normalized (e.g., $V_{IN} \equiv V_{input} \equiv \text{VIN}$).

3. **Value Conflict**: When min/typ/max values conflict:

- Prefer values with explicit test conditions
- Flag conflicts in `extraction_notes`
- Use union of ranges if semantically appropriate

4. **Unit Harmonization**: Convert all values to SI base units with appropriate prefixes.

The deduplication algorithm is formalized in Algorithm 5.

---

**Algorithm 5** Parameter Deduplication

---

**Require:** Raw parameter list $\mathcal{E}_{\text{raw}}$
**Ensure:** Deduplicated parameter list $\mathcal{E}_{\text{dedup}}$
1: $\mathcal{E}_{\text{dedup}} \leftarrow \emptyset$
2: $seen \leftarrow \{\}$                      // Symbol $\rightarrow$ Parameter mapping
3: **for** $e \in \mathcal{E}_{\text{raw}}$ **do**
4:      $s_{\text{norm}} \leftarrow$ NORMALIZESYMBOL($e$.symbol)
5:      **if** $s_{\text{norm}} \in seen$ **then**
6:          $e_{\text{existing}} \leftarrow seen[s_{\text{norm}}]$
7:          $e_{\text{merged}} \leftarrow$ MERGEPARAMS($e_{\text{existing}}, e$)
8:          $seen[s_{\text{norm}}] \leftarrow e_{\text{merged}}$
9:      **else**
10:         $seen[s_{\text{norm}}] \leftarrow e$
11:      **end if**
12: **end for**
13: $\mathcal{E}_{\text{dedup}} \leftarrow$ VALUES($seen$)
14: **return** $\mathcal{E}_{\text{dedup}}$

---

## B.7. Confidence Scoring

We compute an overall extraction confidence score as a weighted sum of component-level confidence scores:

$$\text{Confidence}(\mathcal{C}) = \alpha_1 \cdot C_{\text{type}} + \alpha_2 \cdot C_{\text{pins}} + \alpha_3 \cdot C_{\text{params}} + \alpha_4 \cdot C_{\text{eqns}}, \tag{6}$$

where $C_{\text{type}}$ denotes the circuit type classification confidence obtained from the LLM, $C_{\text{pins}} = 1 - |\text{ambiguous pins}|/|\text{total pins}|$ measures pin definition clarity, $C_{\text{params}} = |\text{params with complete specs}|/|\text{total params}|$ captures parameter completeness, and $C_{\text{eqns}} = |\text{valid SymPy equations}|/|\text{total equations}|$ reflects equation parse success. The weights $\alpha_i$ are empirically determined as $\alpha_1 = 0.3$, $\alpha_2 = 0.2$, $\alpha_3 = 0.3$, and $\alpha_4 = 0.2$.

## B.8. Output Example

We present a representative excerpt from the Stage 1 output for the DC-DC Buck Converter (full output: 847 lines JSON).

*Code Listing 7.* Excerpt from Stage 1 output.

```
{
  "circuit_name": "....",
  "circuit_type": "DCDC_BUCK",
  "circuit_description": "Integrated synchronous buck converter IC for
    efficient step-down DC-DC conversion from 4.3V-to-18V input to
    adjustable output, featuring internal MOSFETs with 80-120 m  RDS(ON),
    800kHz/1.4MHz switching frequency, and comprehensive protection.",
  "key_features": [
    "Wide 4.3V-to-18V input voltage range",
    "2A or 3A output current",
    "Low quiescent current of 250  A ",
    "Power-save mode for light-load efficiency",
    "Over-current protection with hiccup mode",
    "Thermal shutdown protection"
  ],
  "pins": [
    {
      "pin_name": "SW",
      "pin_number": {"SOT563": "2", "TSOT23-6_Type_A": "6"},
      "pin_type": "MIXED",
      "direction": "OUTPUT",
      "description": "Switching node output connecting to external inductor",
```

```
      "inferred_role": "OUTPUT",
      "electrical_constraints": {
        "absolute_max_voltage": {"min": -0.3, "max": 17, "unit": "V"},
        "transient_allowance": {"min": -5, "max": 19, "duration_ns": 10}
      }
    },
    {
      "pin_name": "FB",
      "pin_number": {"SOT563": "6", "TSOT23-6_Type_A": "3"},
      "pin_type": "ANALOG",
      "direction": "INPUT",
      "description": "Feedback input from output voltage divider",
      "inferred_role": "FEEDBACK"
    }
  ],
  "parameters": [
    {
      "symbol": "RDS_ON_HS",
      "name": "High-Side Switch On-Resistance",
      "min_value": null, "typ_value": 120, "max_value": null,
      "unit": " m ",
      "test_conditions": ["V B S T   S W = 5V", "DCDC variant"],
      "category": "RESISTANCE",
      "is_testable": true,
      "related_pins": ["SW", "BST"],
      "test_type_hint": "DC_CURRENT"
    },
    {
      "symbol": "VREF",
      "name": "Feedback Voltage Reference",
      "min_value": 788, "typ_value": 800, "max_value": 812,
      "unit": "mV",
      "test_conditions": ["TA = 25  C ", "SOT563/TSOT23-B package"],
      "category": "VOLTAGE",
      "is_testable": true,
      "related_pins": ["FB"],
      "test_type_hint": "DC_VOLTAGE"
    }
  ],
  "equations": [
    {
      "equation_id": "EQ_THERMAL_POWER_DISSIPATION",
      "raw_form": "PD(MAX) = (TJ(MAX)     TA) /  J A ",
      "sympy_form": "PD_MAX = (TJ_MAX - TA) / theta_JA",
      "latex_form": "P_{D(MAX)} = \\frac{T_{J(MAX)} - T_A}{\\theta_{JA}}",
      "output_variable": "PD_MAX",
      "input_variables": [
        {"symbol": "TJ_MAX", "unit": " C ", "how_to_obtain": "FROM_SPEC"},
        {"symbol": "TA", "unit": " C ", "how_to_obtain": "MEASURED"},
        {"symbol": "theta_JA", "unit": " C /W", "how_to_obtain": "FROM_SPEC"}
      ]
    }
  ],
  "extraction_confidence": 0.98
}
```

## B.9. Extraction Statistics

Table 7 presents extraction statistics for the Industrial DCDC datasheet.

*Table 7.* Stage 1 extraction statistics (DCDC datasheet).

| Metric | Value | Notes |
| --- | --- | --- |
| Input document blocks | 89 | From Stage 0 |
| Extracted pins | 6 | All with inferred roles |
| Extracted parameters | 142 | After deduplication |
| Testable parameters | 118 | 83.1% testability rate |
| Extracted equations | 6 | All converted to SymPy |
| Unique test conditions | 47 | Across all parameters |
| Parameter categories | 12 | VOLTAGE, CURRENT, ... |
| LLM calls (total) | 8 | 1 ID + 1 pin + 5 param + 1 eqn |
| Processing time | 12.3s | With parallel param batches |
| Overall confidence | 0.98 | Computed via Eq. 6 |

## B.10. Parameter Category Distribution

Figure 7 shows the distribution of extracted parameters by category.

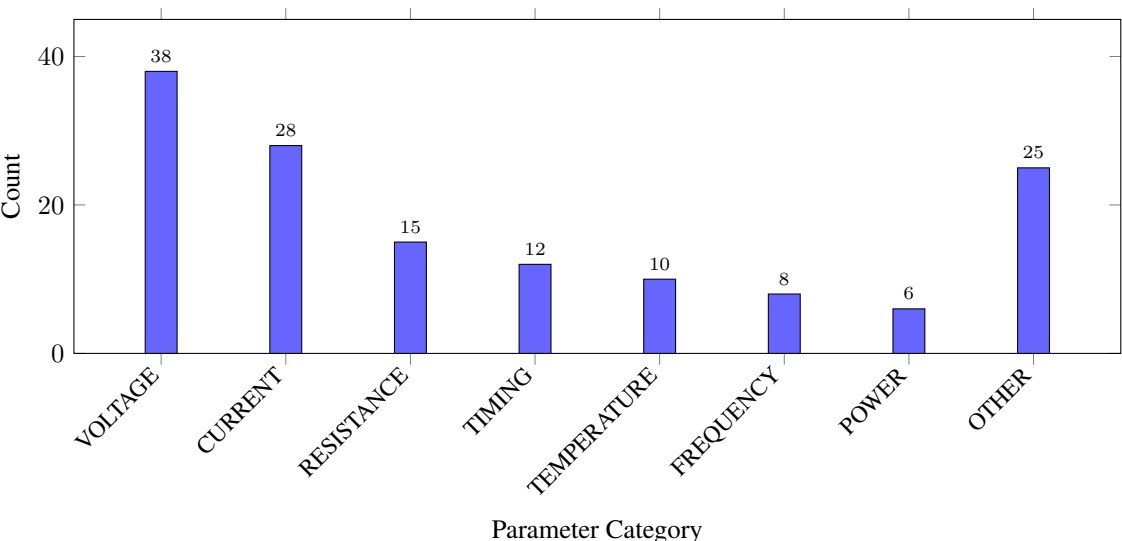

*Figure 7.* Distribution of extracted parameters by category (Industrial DCDC).

## B.11. Error Analysis and Limitations

We identify the following limitations and potential failure modes:

1. **Multi-Variant Parameters**: Parameters that differ across device variants may have multiple typical values stored as lists, requiring downstream stages to handle variant selection.
2. **Incomplete Test Conditions**: Some parameters lack explicit test conditions in the source document, leading to `test_conditions: []` entries.
3. **Unit Ambiguity**: Certain extracted values have ambiguous units (e.g., "10" without explicit unit), flagged in `extraction_notes`.
4. **Cross-Reference Limitations**: Implicit references to figures or external documents cannot be resolved automatically.

# C. Stage 2: Test Task Extraction

Stage 2 transforms the extracted circuit information into actionable verification tasks. This stage employs a two-phase approach: (1) intelligent task generation through semantic reasoning, and (2) weak supervision-based task validation to filter low-quality or non-testable tasks.

## C.1. Problem Formulation

Given the document database $\mathcal{D}$ from Stage 0 and circuit information $\mathcal{C}$ from Stage 1, Stage 2 generates a set of test tasks $\mathcal{T} = \{t_1, t_2, \ldots, t_n\}$ where each task $t_i$ is defined as a tuple:

$$t_i = \langle \tau_i, \pi_i, \mathcal{E}_i, \mathcal{S}_i, \mathcal{M}_i, \mathcal{V}_i \rangle, \tag{7}$$

where $\tau_i$ denotes the test type, $\pi_i$ represents the primary parameter under test, $\mathcal{E}_i$ is the mathematical evidence set, $\mathcal{S}_i$ specifies the stimulus port configuration, $\mathcal{M}_i$ defines the measurement port configuration, and $\mathcal{V}_i$ encodes the verification criteria.

## C.2. Data Structures

### C.2.1. TEST TASK REPRESENTATION

*Table 8.* TestTask Data Structure

| Field | Type | Description |
|---|---|---|
| task_id | str | Unique identifier (e.g., task_001) |
| task_name | str | Human-readable task description |
| test_type | TestType | Classification of test domain |
| primary_parameter | str | Target parameter symbol |
| math_evidences | List[MathEvidence] | Specification-derived constraints |
| stimulus_ports | List[PortConfig] | Input stimulus configuration |
| measurement_ports | List[PortConfig] | Output measurement points |
| priority | Priority | Task importance ranking |
| parameter_category | ParamCategory | Parameter classification |

### C.2.2. TEST TYPE TAXONOMY

We define a hierarchical test type taxonomy based on the analysis domain and measurement methodology:

*Table 9.* Test Type Classification

| Test Type | Domain | SPICE Analysis | Example Parameters |
|---|---|---|---|
| DC_STATIC | DC | .op, .dc | $V_{\text{REF}}$, $I_{\text{DD}}$, $V_{\text{OS}}$ |
| DC_SWEEP | DC | .dc | Load/Line regulation |
| AC_GAIN | AC | .ac | $A_{\text{OL}}$, CMRR, PSRR |
| AC_BANDWIDTH | AC | .ac | GBW, $f_{\text{3dB}}$ |
| AC_PHASE | AC | .ac | Phase margin, Gain margin |
| TRANSIENT_TIMING | Transient | .tran | $t_{\text{ON}}$, $t_{\text{OFF}}$, SR |
| TRANSIENT_LOAD | Transient | .tran | Load transient response |
| PROTECTION | Mixed | .dc/.tran | OCP, OVP, UVLO, TSD |
| NOISE | AC | .noise | $e_n$, $i_n$ |

### C.2.3. PARAMETER CATEGORY CLASSIFICATION

To distinguish between directly testable specifications and derived design constraints, we classify parameters into four categories:

*Table 10.* Parameter Category Taxonomy

| Category | Definition |
|---|---|
| SPEC_MEASURABLE | Explicit numeric specification with units, limits, and test conditions directly extractable from the datasheet |
| DERIVED_VARIABLE | Computed from other parameters or equations; requires formula evaluation |
| DESIGN_CONSTRAINT | Qualitative requirement or operating mode without explicit numeric bounds |
| META_INFO | Non-electrical metadata (package info, ordering codes, etc.) |

## C.3. Algorithm

The Stage 2 pipeline consists of two main phases: task generation and task validation.

---

**Algorithm 6** Stage 2: Test Task Extraction Pipeline

---

**Require:** Document database $\mathcal{D}$, Circuit information $\mathcal{C}$, LLM model $\mathcal{M}$
**Ensure:** Task extraction result $\mathcal{R}$
 1: **Phase 1: Intelligent Task Generation**
 2: $\mathcal{P}_{\text{testable}} \leftarrow$ FILTERTESTABLEPARAMS($\mathcal{C}$.parameters)
 3: $\mathcal{T}_{\text{raw}} \leftarrow \emptyset$
 4: **for** $p \in \mathcal{P}_{\text{testable}}$ **do**
 5:    context $\leftarrow$ BUILDPARAMCONTEXT($p, \mathcal{C}, \mathcal{D}$)
 6:    $t \leftarrow \mathcal{M}$.GENERATETASK(context, TASKGENPROMPT)
 7:    $\mathcal{T}_{\text{raw}} \leftarrow \mathcal{T}_{\text{raw}} \cup \{t\}$
 8: **end for**
 9: **Phase 2: Weak Supervision Validation**
10: $\mathcal{T}_{\text{validated}} \leftarrow \emptyset$
11: **for** $t \in \mathcal{T}_{\text{raw}}$ **do**
12:    evidence $\leftarrow$ EXTRACTEVIDENCESNIPPETS($t, \mathcal{D}$)
13:    review $\leftarrow \mathcal{M}$.REVIEWTASK($t$, evidence, REVIEWPROMPT)
14:    $t$.methodology.llm_review $\leftarrow$ review
15:    $t$.parameter_category $\leftarrow$ review.category
16:    **if** review.decision $\in \{$keep, optional$\}$ **then**
17:      $\mathcal{T}_{\text{validated}} \leftarrow \mathcal{T}_{\text{validated}} \cup \{t\}$
18:    **end if**
19: **end for**
20:
21: **Phase 3: Result Assembly**
22: $\mathcal{R} \leftarrow$ ASSEMBLERESULT($\mathcal{T}_{\text{validated}}, \mathcal{C}$)
23: $\mathcal{R}$.stats $\leftarrow$ COMPUTESTATISTICS($\mathcal{T}_{\text{raw}}, \mathcal{T}_{\text{validated}}$)
24: **return** $\mathcal{R}$

---

### C.3.1. TEST TYPE INFERENCE

The test type inference algorithm employs semantic reasoning rather than keyword matching:

---

**Algorithm 7** Semantic Test Type Inference

---

**Require:** Parameter $p$, Circuit type $c_{\text{type}}$, Equations $\mathcal{Q}$
**Ensure:** Inferred test type $\tau$
 1: features $\leftarrow$ EXTRACTSEMANTICFEATURES($p$)
 2:                                        // Feature extraction based on physical meaning
 3: $f_{\text{static}} \leftarrow$ ISSTATICMEASUREMENT($p$.description)
 4: $f_{\text{freq}} \leftarrow$ INVOLVESFREQUENCY($p$.unit, $\mathcal{Q}$)
 5: $f_{\text{time}} \leftarrow$ INVOLVESTIMING($p$.unit, $p$.conditions)
 6: $f_{\text{protect}} \leftarrow$ ISPROTECTIONFUNCTION($p$.name, $p$.description)
 7:                                    // Decision logic based on semantic understanding
 8: **if** $f_{\text{protect}}$ **then**
 9:    **return** INFERPROTECTIONTYPE($p$)
10: **else if** $f_{\text{freq}}$ **then**
11:    **return** INFERACTYPE($p, \mathcal{Q}$)
12: **else if** $f_{\text{time}}$ **then**
13:    **return** INFERTRANSIENTTYPE($p$)
14: **else if** $f_{\text{static}}$ **then**
15:    **return** INFERDCTYPE($p$)
16: **else**
17:    **return** OTHER
18: **end if**

---

### C.3.2. MATHEMATICAL EVIDENCE EXTRACTION

For each test task, we extract mathematical evidence from the specification to define pass/fail criteria:

---

**Algorithm 8** Mathematical Evidence Extraction

---

**Require:** Parameter $p$, Document blocks $\mathcal{B}$, Equations $\mathcal{Q}$
**Ensure:** Evidence set $\mathcal{E}$
1: $\mathcal{E} \leftarrow \emptyset$
2:                                                                  // Extract from specification limits
3: **if** $p.\text{min\_value} \neq$ `null` $\vee$ $p.\text{max\_value} \neq$ `null` **then**
4:     $e_{\text{spec}} \leftarrow$ CREATESPECEVIDENCE$(p)$
5:     $e_{\text{spec}}.\text{constraint} \leftarrow$ BUILDCONSTRAINTEXPR$(p)$
6:     $\mathcal{E} \leftarrow \mathcal{E} \cup \{e_{\text{spec}}\}$
7: **end if**
8:                                                                  // Extract from related equations
9: **for** $q \in \mathcal{Q}$ **do**
10:     **if** $p.\text{symbol} \in q.\text{variables}$ **then**
11:         $e_{\text{eq}} \leftarrow$ CREATEEQUATIONEVIDENCE$(q, p)$
12:         $\mathcal{E} \leftarrow \mathcal{E} \cup \{e_{\text{eq}}\}$
13:     **end if**
14: **end for**
15:
16:                                                                  // Extract test conditions
17: **for** $e \in \mathcal{E}$ **do**
18:     $e.\text{conditions} \leftarrow$ EXTRACTCONDITIONS$(p, \mathcal{B})$
19: **end for**
20: **return** $\mathcal{E}$

---

## C.4. Prompt Engineering

### C.4.1. TASK GENERATION PROMPT

The task generation prompt employs a structured reasoning approach that guides the LLM through the test design process:

---

**TASKGENERATIONPROMPT**

```
# Task: Intelligent Test Task Generation

You are an expert analog IC test engineer. For the given
parameter, design a complete verification test by REASONING:

## Circuit Context
- Name: {circuit_name}
- Type: {circuit_type}
- Available Pins: {pins_json}

## Target Parameter
- Symbol: {param_symbol}
- Description: {param_description}
- Spec Limits: min={min_val}, typ={typ_val}, max={max_val}
- Unit: {unit}
- Test Conditions: {conditions}

## Reasoning Requirements
1. **Test Type Determination**: What measurement domain?
   - DC: Static voltage/current at operating point
   - AC: Frequency response, gain, phase
   - Transient: Time-domain response, timing
   - Protection: Fault condition behavior

2. **Stimulus Design**: How to excite the circuit?
3. **Measurement Strategy**: Which nodes to probe?
4. **Pass/Fail Criteria**: How to determine compliance?

## Output JSON Schema
{
  "task_id": "task_XXX",
```

---

```
  "test_classification": {
    "test_type": "DC_STATIC|AC_GAIN|...",
    "test_domain": "DC|AC|TRANSIENT|MIXED",
    "reasoning": "Explanation of type selection"
  },
  "test_setup": {
    "stimulus_ports": [...],
    "measurement_ports": [...],
    "control_ports": [...]
  },
  "math_evidence": {
    "constraint_expression": "SymPy expression",
    "pass_criteria": "Human-readable criterion"
  },
  "priority": "HIGH|MEDIUM|LOW"
}
```

## C.4.2. WEAK REVIEW PROMPT

The validation phase uses a separate prompt that focuses on evidence-based verification:

**TASKWEAKREVIEWPROMPT**

```
# Task: Test Task Validity Review

Review the test task using ONLY the evidence snippets provided.

## Task Under Review
- Task Name: {task_name}
- Primary Parameter: {primary_parameter}
- Current Test Type: {test_type}

## Evidence Snippets (verbatim from PDF)
{evidence_snippets}

## Classification Criteria
1. SPEC_MEASURABLE: Explicit numeric spec with units/limits
2. DERIVED_VARIABLE: Computed from equations/other params
3. DESIGN_CONSTRAINT: Qualitative requirement, no numeric spec
4. META_INFO: Non-electrical metadata

## Review Output
{
  "decision": "keep|optional|drop",
  "measurable": true|false,
  "verifiable": true|false,
  "parameter_category": "SPEC_MEASURABLE|...",
  "evidence_quote": "Verbatim quote or NONE",
  "missing_info": ["list", "of", "gaps"],
  "reason": "Concise justification"
}
```

## C.5. Output Specification

The Stage 2 output follows a structured JSON schema:

## C.6. Constraint Expression Formalization

For parameters with explicit numeric specifications, Stage 2 generates formal constraint expressions in SymPy-compatible syntax:

*Table 11.* TaskExtractionResult Schema

| Field | Type | Description |
|---|---|---|
| document_id | str | Source document identifier |
| circuit_name | str | Target circuit name |
| circuit_type | str | Circuit type classification |
| test_tasks | List[TestTask] | Generated test tasks |
| total_parameters | int | Total parameters from Stage 1 |
| testable_parameters | int | Parameters deemed testable |
| generated_tasks | int | Final task count |
| processing_stats | Dict | LLM calls, timing statistics |

$$\text{ConstraintExpr}(p) = \begin{cases} p_{\min} \leq p \leq p_{\max} & \text{if both limits defined} \\ p \geq p_{\min} & \text{if only minimum defined} \\ p \leq p_{\max} & \text{if only maximum defined} \\ \texttt{null} & \text{otherwise} \end{cases} \tag{8}$$

Example constraint expressions extracted from the Industrial DCDC datasheet:

*Table 12.* Example Constraint Expressions

| Parameter | Symbol | Constraint Expression | Unit |
|---|---|---|---|
| Feedback Voltage | $V_{\text{REF}}$ | VREF >= 788 and VREF <= 812 | mV |
| Switch Voltage | $V_{\text{SW}}$ | VSW >= -0.3 and VSW <= 17 | V |
| Valley Current Limit | $I_{\text{LIMIT}}$ | ILIMIT >= 2 and ILIMIT <= 3 | A |
| Junction Temperature | $T_J$ | TJ <= 150 | °C |
| Switch Leakage | $I_{\text{SWLKG}}$ | SWLKG <= 1 | $\mu$A |
| Enable Rising Threshold | $V_{\text{EN,RISING}}$ | VEN_RISING >= 1.1 and VEN_RISING <= 1.3 | V |

### C.7. Limitations and Design Constraints

In the extraction tasks, most are classified as SPEC_MEASURABLE, while a small number of extracted tasks correspond to DESIGN_CONSTRAINT parameters that lack explicit numeric specifications. These include:

- **Operating Modes**: CCM, PFM, Power Save Mode
- **Protection Features**: OCP hiccup mode, auto-recovery behavior
- **Layout Guidelines**: Capacitor placement, trace routing
- **Qualitative Behaviors**: "excellent load regulation", "fairly constant frequency"

For such parameters, the LLM review correctly identifies missing information (numeric values, acceptance criteria, test conditions) and classifies them as optional, enabling downstream stages to prioritize SPEC_MEASURABLE tasks for testbench generation.

## D. Stage 3: Port Analysis

Stage 3 performs deep semantic analysis of circuit ports to infer their functional roles, signal characteristics, and inter-port dependencies. This information is critical for downstream testbench generation, as it determines how each port should be stimulated or measured during verification.

### D.1. Problem Formulation

Given the circuit information $\mathcal{C}$ from Stage 1 and optionally the test tasks $\mathcal{T}$ from Stage 2, Stage 3 produces a comprehensive port analysis result $\mathcal{A}$ consisting of:

$$\mathcal{A} = \langle \mathcal{P}_a, \mathcal{S}_p, \mathcal{G}_d, \mathcal{M}_p, \mathcal{O}_t \rangle, \tag{9}$$

where $\mathcal{P}_a$ denotes the set of individual port analyses, $\mathcal{S}_p$ represents the identified signal paths, $\mathcal{G}_d$ encodes the port dependency graph, $\mathcal{M}_p$ defines the port-to-pin mapping, and $\mathcal{O}_t$ contains the recommended test topologies.

## D.2. Port Role Taxonomy

We define a comprehensive taxonomy of port roles based on their functional characteristics in analog/mixed-signal circuits:

*Table 13.* Port Role Classification

| Role | Symbol | Description |
|------|--------|-------------|
| INPUT | $\rho_{\text{in}}$ | Signal input ports receiving external stimuli |
| OUTPUT | $\rho_{\text{out}}$ | Signal output ports providing circuit response |
| BIDIRECTIONAL | $\rho_{\text{bi}}$ | Ports capable of both input and output |
| POWER | $\rho_{\text{pwr}}$ | Power supply rails (VDD, VCC, VIN) |
| GROUND | $\rho_{\text{gnd}}$ | Ground reference nodes (VSS, GND) |
| BIAS | $\rho_{\text{bias}}$ | Bias voltage/current references |
| CONTROL | $\rho_{\text{ctrl}}$ | Digital or analog control signals |
| ENABLE | $\rho_{\text{en}}$ | Enable/disable control pins |
| CLOCK | $\rho_{\text{clk}}$ | Clock or timing reference signals |
| FEEDBACK | $\rho_{\text{fb}}$ | Feedback loop sensing nodes |

## D.3. Data Structures

### D.3.1. PORT ANALYSIS RESULT

*Table 14.* PortAnalysis Data Structure

| Field | Type | Description |
|-------|------|-------------|
| port_name | str | Logical port identifier |
| pin_names | List[str] | Physical pin(s) comprising this port |
| inferred_role | PortRole | Semantically inferred functional role |
| role_confidence | float | Confidence score $\in [0, 1]$ |
| role_reasoning | str | LLM-generated explanation |
| signal_type | SignalType | ANALOG \| DIGITAL \| MIXED \| POWER |
| signal_domain | str | voltage \| current \| frequency |
| typical_range | Dict | Expected signal range with units |
| testability | float | Testability score $\in [0, 1]$ |
| recommended_stimulus | str | Suggested stimulus method |
| recommended_measurement | str | Suggested measurement method |

### D.3.2. SIGNAL PATH REPRESENTATION

Signal paths capture the directed flow of information through the circuit:

*Table 15.* SignalPath Data Structure

| Field | Type | Description |
|-------|------|-------------|
| path_id | str | Unique path identifier |
| path_type | PathType | MAIN_SIGNAL \| FEEDBACK \| BIAS \| CONTROL |
| source_port | str | Origin port of the signal |
| intermediate_ports | List[str] | Ports traversed along the path |
| sink_port | str | Destination port |
| is_critical | bool | Whether path affects primary function |
| latency_critical | bool | Whether timing is critical |
| path_reasoning | str | Explanation of path significance |

### D.3.3. DEPENDENCY GRAPH

The port dependency graph $\mathcal{G}_d = (V, E, L)$ is a directed acyclic graph where:

- $V$ is the set of ports
- $E \subseteq V \times V$ represents dependency edges
- $L : V \to \mathbb{N}$ assigns each port to a dependency level

Ports at level 0 (typically POWER and GROUND) must be established before ports at higher levels can operate correctly.

## D.4. Algorithm

The Stage 3 pipeline consists of three main phases: port semantic analysis, signal flow inference, and dependency ordering.

---

**Algorithm 9** Stage 3: Port Analysis Pipeline

---

**Require:** Circuit information $\mathcal{C}$, Task result $\mathcal{T}$ (optional), LLM model $\mathcal{M}$
**Ensure:** Port analysis result $\mathcal{A}$
 1: **Phase 1: Port Semantic Analysis**
 2: $\mathcal{P}_{\text{pins}} \leftarrow \mathcal{C}.\text{pins}$
 3: $\mathcal{P}_a \leftarrow \emptyset$
 4: **for** $p \in \mathcal{P}_{\text{pins}}$ **do**
 5: $\quad$ context $\leftarrow$ BUILDPORTCONTEXT$(p, \mathcal{C}, \mathcal{T})$
 6: $\quad a \leftarrow \mathcal{M}.\text{ANALYZEPORT}(\text{context}, \text{PORTANALYSISPROMPT})$
 7: $\quad a.\text{testability} \leftarrow$ COMPUTETESTABILITY$(a, \mathcal{T})$
 8: $\quad \mathcal{P}_a \leftarrow \mathcal{P}_a \cup \{a\}$
 9: **end for**
10: **Phase 2: Signal Flow Inference**
11: flow_context $\leftarrow$ BUILDFLOWCONTEXT$(\mathcal{P}_a, \mathcal{C})$
12: $\mathcal{S}_p \leftarrow \mathcal{M}.\text{INFERSIGNALPATHS}(\text{flow\_context}, \text{SIGNALFLOWPROMPT})$
13: **Phase 3: Dependency Graph Construction**
14: $\mathcal{G}_d \leftarrow$ BUILDDEPENDENCYGRAPH$(\mathcal{P}_a, \mathcal{S}_p)$
15: $\mathcal{G}_d.\text{levels} \leftarrow$ TOPOLOGICALSORT$(\mathcal{G}_d)$
16: **Phase 4: Port Mapping and Topology Generation**
17: $\mathcal{M}_p \leftarrow$ CREATEPORTMAPPING$(\mathcal{P}_a)$
18: $\mathcal{O}_t \leftarrow$ GENERATETESTTOPOLOGIES$(\mathcal{P}_a, \mathcal{S}_p, \mathcal{T})$
19:
20: $\mathcal{A} \leftarrow$ ASSEMBLERESULT$(\mathcal{P}_a, \mathcal{S}_p, \mathcal{G}_d, \mathcal{M}_p, \mathcal{O}_t)$
21: **return** $\mathcal{A}$

---

### D.4.1. PORT ROLE INFERENCE

The port role inference algorithm employs multi-factor semantic reasoning:

---

**Algorithm 10** Semantic Port Role Inference

---

**Require:** Port $p$, Circuit type $c_{\text{type}}$, Pin constraints $\mathcal{K}$
**Ensure:** Inferred role $\rho$, Confidence $\gamma$
 1: $\hspace{6cm}$ // Extract semantic features from port metadata
 2: $f_{\text{name}} \leftarrow$ ANALYZENAMING$(p.\text{name})$
 3: $f_{\text{desc}} \leftarrow$ ANALYZEDESCRIPTION$(p.\text{description})$
 4: $f_{\text{elec}} \leftarrow$ ANALYZECONSTRAINTS$(\mathcal{K})$
 5: $f_{\text{ctx}} \leftarrow$ ANALYZECIRCUITCONTEXT$(c_{\text{type}})$
 6: $\hspace{7cm}$ // Compute role probabilities
 7: $\mathbf{P} \leftarrow$ LLMROLECLASSIFICATION$(f_{\text{name}}, f_{\text{desc}}, f_{\text{elec}}, f_{\text{ctx}})$
 8: $\rho \leftarrow \arg\max_r \mathbf{P}[r]$
 9: $\gamma \leftarrow \mathbf{P}[\rho]$
10: $\hspace{6.5cm}$ // Apply circuit-type-specific heuristics
11: **if** $c_{\text{type}} = \text{DCDC\_BUCK}$ **then**
12: $\quad (\rho, \gamma) \leftarrow$ APPLYBUCKHEURISTICS$(\rho, \gamma, p)$
13: **else if** $c_{\text{type}} = \text{OTA}$ **then**
14: $\quad (\rho, \gamma) \leftarrow$ APPLYOTAHEURISTICS$(\rho, \gamma, p)$
15: **end if**
16: **return** $(\rho, \gamma)$

---

## D.4.2. DEPENDENCY LEVEL ASSIGNMENT

Ports are organized into dependency levels through topological sorting:

---

**Algorithm 11** Dependency Level Assignment

---

**Require:** Port analyses $\mathcal{P}_a$, Signal paths $\mathcal{S}_p$
**Ensure:** Dependency graph $\mathcal{G}_d$ with level assignments
1: $V \leftarrow \{p.\text{name} : p \in \mathcal{P}_a\}$
2: $E \leftarrow \emptyset$
3:                                                                  // Build edges from signal paths
4: **for** $s \in \mathcal{S}_p$ **do**
5:     $E \leftarrow E \cup \{(s.\text{source}, s.\text{sink})\}$
6:     **for** $i \in s.\text{intermediate}$ **do**
7:         $E \leftarrow E \cup \{(s.\text{source}, i), (i, s.\text{sink})\}$
8:     **end for**
9: **end for**
10:                                                                 // Assign levels via BFS from sources
11: $L \leftarrow \{\}$
12: sources $\leftarrow \{v \in V : \text{in\_degree}(v) = 0\}$
13: **for** $v \in$ sources **do**
14:     $L[v] \leftarrow 0$
15: **end for**
16:                                                                 // Propagate levels
17: **for** $v \in \text{TOPOLOGICALORDER}(V, E)$ **do**
18:     **if** $v \notin L$ **then**
19:         $L[v] \leftarrow \max_{(u,v) \in E} L[u] + 1$
20:     **end if**
21: **end for**
22: **return** $(V, E, L)$

---

## D.5. Prompt Engineering

### D.5.1. PORT SEMANTIC ANALYSIS PROMPT

**PORTSEMANTICANALYSISPROMPT**

```
# Task: Deep Port Semantic Analysis

Analyze the ports/pins of this circuit to understand their
functional roles and signal characteristics.

## Circuit Information
- Name: {circuit_name}
- Type: {circuit_type}
- Function: {circuit_function}

## Pin Definitions from Specification
{pin_definitions_json}

## Role Classification Guidelines
- INPUT: Signal inputs receiving external stimuli
- OUTPUT: Signal outputs providing circuit response
- POWER: Supply rails (VDD, VCC, VIN)
- GROUND: Reference nodes (VSS, GND)
- BIAS: Bias voltage/current references
- CONTROL: Digital/analog control signals
- ENABLE: On/off control pins
- FEEDBACK: Regulation loop sensing nodes

## Output JSON Schema
{
  "port_analyses": [
    {
```

```
        "port_name": "IN",
        "pin_names": ["IN"],
        "role_analysis": {
          "inferred_role": "POWER",
          "confidence": 0.99,
          "reasoning": "Primary power supply input..."
        },
        "signal_characteristics": {
          "signal_type": "POWER",
          "signal_domain": "voltage",
          "typical_range": {
            "min": 4.3, "typ": 12.0, "max": 18.0, "unit": "V"
          }
        },
        "test_recommendations": {
          "testability_score": 0.95,
          "as_stimulus": {
            "suitable": true,
            "method": "regulated_voltage_source"
          },
          "as_measurement": {
            "suitable": true,
            "method": "voltmeter_with_ripple"
          }
        }
      }
    ]
}
```

### D.5.2. SIGNAL FLOW INFERENCE PROMPT

**SIGNALFLOWPROMPT**

```
# Task: Signal Flow and Dependency Ordering

Infer signal flow paths and construct the port dependency
graph for verification sequencing.

## Circuit Context
- Name: {circuit_name}
- Type: {circuit_type}

## Analyzed Ports
{port_analyses_json}

## Path Type Definitions
- MAIN_SIGNAL: Primary signal processing path
- FEEDBACK: Regulation/control feedback loop
- BIAS: Bias generation and distribution
- CONTROL: Control signal propagation

## Output JSON Schema
{
  "signal_paths": [
    {
      "path_id": "p1",
      "path_type": "MAIN_SIGNAL",
      "source_port": "IN",
      "intermediate_ports": ["SW"],
      "sink_port": "FB",
      "is_critical": true,
      "latency_critical": true,
      "path_reasoning": "Primary power conversion path..."
    }
```

```
  ],
  "dependency_graph": {
    "levels": [["IN", "GND"], ["BST", "EN"], ["SW"], ["FB"]],
    "edges": [
      {"from": "IN", "to": "SW"},
      {"from": "SW", "to": "FB"}
    ],
    "reasoning": "Level 0: Power/Ground foundation..."
  }
}
```

## D.6. Output Specification

*Table 16.* Stage3Result Schema

| Field | Type | Description |
|---|---|---|
| circuit_name | str | Target circuit identifier |
| circuit_type | str | Circuit type classification |
| port_analyses | List[PortAnalysis] | Per-port analysis results |
| signal_paths | List[SignalPath] | Identified signal flow paths |
| dependency_graph | DependencyGraph | Port dependency DAG |
| port_mapping | Dict[str, List[str]] | Logical-to-physical mapping |
| test_topologies | List[TestTopology] | Recommended test configurations |
| processing_stats | Dict | LLM calls, timing statistics |

## D.7. Experimental Results

Table 17 presents the port analysis results for the Industrial DCDC Buck converter.

*Table 17.* Stage 3 Analysis Statistics

| Metric | Value |
|---|---|
| Total Ports Analyzed | 6 |
| Signal Paths Identified | 5 |
| Dependency Levels | 4 |
| LLM API Calls | 86 |
| Processing Time | 20.2 s |
| *By Inferred Role* | |
| POWER | 1 (16.7%) |
| GROUND | 1 (16.7%) |
| OUTPUT | 1 (16.7%) |
| BIAS | 1 (16.7%) |
| ENABLE | 1 (16.7%) |
| FEEDBACK | 1 (16.7%) |
| *By Signal Type* | |
| POWER | 2 (33.3%) |
| ANALOG | 1 (16.7%) |
| DIGITAL | 1 (16.7%) |
| MIXED | 2 (33.3%) |
| Average Role Confidence | 0.988 |
| Average Testability Score | 0.938 |

## D.8. Port Analysis Results

Table 18 summarizes the inferred characteristics for each port of the DCDC Buck converter.

*Table 18.* Port Analysis Results

| Port | Role | Signal Type | Range | Testability |
|------|------|-------------|-------|-------------|
| IN | POWER | POWER | 4.3–18.0 V | 0.95 |
| SW | OUTPUT | MIXED | 0–18.0 V | 0.92 |
| GND | GROUND | POWER | 0 V | 0.98 |
| BST | BIAS | POWER | 5.0–6.0 V | 0.88 |
| EN | ENABLE | DIGITAL | 0–5.0 V | 0.96 |
| FB | FEEDBACK | ANALOG | 0–1.2 V | 0.94 |

## D.9. Dependency Graph Visualization

Figure 8 illustrates the port dependency hierarchy for the DCDC.

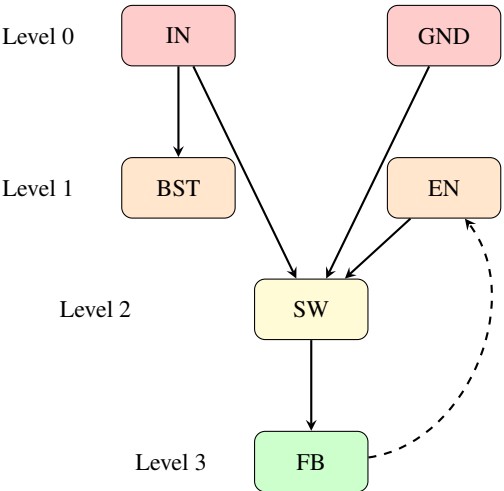

*Figure 8.* Port dependency graph for DCDC Buck converter. Solid arrows indicate power/signal dependencies; dashed arrow indicates feedback path. Ports must be established in level order for correct circuit operation.

## D.10. Signal Path Analysis

Table 19 enumerates the identified signal paths and their characteristics.

*Table 19.* Identified Signal Paths

| ID | Type | Path | Critical | Latency |
|----|------|------|----------|---------|
| p1 | MAIN_SIGNAL | IN $\rightarrow$ SW $\rightarrow$ FB | ✓ | ✓ |
| p2 | FEEDBACK | FB $\rightarrow$ EN | ✓ | ✓ |
| p3 | CONTROL | EN $\rightarrow$ SW | ✓ | – |
| p4 | BIAS | IN $\rightarrow$ BST | ✓ | – |
| p5 | MAIN_SIGNAL | IN $\rightarrow$ SW $\rightarrow$ GND | ✓ | ✓ |

## D.11. Testability Analysis

The testability score $\tau_p$ for each port is computed based on multiple factors:

$$\tau_p = w_1 \cdot \text{accessibility} + w_2 \cdot \text{observability} + w_3 \cdot \text{controllability} \tag{10}$$

where the weights are empirically set to $w_1 = 0.4$, $w_2 = 0.3$, $w_3 = 0.3$. Table 20 details the factors contributing to each port's testability score.

*Table 20.* Testability Factor Analysis

| Port | Accessibility | Observability | Controllability | $\tau_p$ |
|------|---------------|---------------|-----------------|----------|
| IN | 1.0 | 0.9 | 0.95 | 0.95 |
| SW | 0.9 | 1.0 | 0.8 | 0.92 |
| GND | 1.0 | 1.0 | 0.95 | 0.98 |
| BST | 0.8 | 0.9 | 0.85 | 0.88 |
| EN | 1.0 | 0.9 | 0.98 | 0.96 |
| FB | 0.95 | 0.95 | 0.9 | 0.94 |

### D.12. Test Topology Recommendations

Based on the port analysis, Stage 3 generates recommended test topologies for each verification scenario. For the DCDC Buck converter, the following topology recommendations are produced:

1. **DC Regulation Test**: Apply regulated voltage to IN, enable via EN, measure FB and output voltage through resistor divider
2. **Efficiency Test**: Measure input power at IN, output power at load, compute $\eta = P_{\text{out}}/P_{\text{in}}$
3. **Switching Waveform Test**: Monitor SW node with oscilloscope to verify duty cycle and dead-time
4. **Protection Test**: Force fault conditions and verify EN-controlled shutdown behavior

### D.13. Integration with Downstream Stages

The Stage 3 output directly informs subsequent pipeline stages:

- **Stage 4 (Oracle Generation)**: Port roles determine which variables appear in verification formulas

- **Stage 5 (Testbench Generation)**: Dependency levels dictate power-up sequencing in generated SPICE netlists; testability scores guide stimulus/measurement assignment

The dependency graph ensures that testbench initialization follows the correct sequence: Level 0 ports (IN, GND) are established first, followed by Level 1 (BST, EN), then Level 2 (SW begins switching), and finally Level 3 (FB provides regulation feedback).

## E. Stage 2.5: Cross Validation

Stage 2.5 performs comprehensive consistency verification between the outputs of Stage 1 (Circuit Information), Stage 2 (Test Tasks), and Stage 3 (Port Analysis). This validation step ensures that all cross-references are resolvable and that the generated test tasks are executable with the available circuit information.

### E.1. Motivation

The multi-stage LLM pipeline may introduce potential inconsistencies due to:

- **Hallucinated Port References**: The LLM may generate port names in test tasks that do not exist in the circuit specification.
- **Missing Parameter Links**: Test tasks may reference parameters that were not extracted in Stage 1.
- **Incomplete Evidence**: Mathematical evidence may lack the required test conditions.
- **Semantic Drift**: Different LLM calls may use inconsistent naming conventions.

Cross validation acts as a quality gate, identifying and categorizing these issues before testbench generation.

### E.2. Validation Categories

We define six validation categories, each targeting a specific class of consistency issues:

*Table 21.* Validation Category Definitions

| Category | Severity | Description |
|---|---|---|
| PORT_EXISTENCE | Warning | Validates that all ports referenced in test tasks exist in the port analysis result |
| PARAMETER_REFERENCE | Error | Ensures each test task has a valid primary parameter from Stage 1 |
| EVIDENCE_REFERENCE | Warning | Verifies that mathematical evidence references valid sources |
| VARIABLE_RESOLUTION | Error | Checks that all variables in formulas are resolvable |
| CONDITION_COMPLETENESS | Info | Flags evidence entries missing test conditions |
| DATA_INTEGRITY | Warning | Detects duplicate identifiers and malformed data |

## E.3. Data Structures

### E.3.1. VALIDATION REPORT

*Table 22.* ValidationReport Data Structure

| Field | Type | Description |
|---|---|---|
| validation_passed | bool | Overall pass/fail status |
| total_checks | int | Total number of validation checks |
| passed_checks | int | Number of successful checks |
| failed_checks | int | Number of failed checks |
| error_count | int | Critical errors (blocks pipeline) |
| warning_count | int | Non-critical issues |
| info_count | int | Informational notices |
| issues | List[ValidationIssue] | Detailed issue list |

### E.3.2. VALIDATION ISSUE

*Table 23.* ValidationIssue Data Structure

| Field | Type | Description |
|---|---|---|
| issue_id | str | Unique identifier (e.g., V001) |
| category | ValidationType | Validation category |
| severity | Severity | error \| warning \| info |
| message | str | Human-readable description |
| context | Dict | Contextual information (task_id, etc.) |
| suggestion | str | Remediation suggestion |
| affected_items | List[str] | List of affected identifiers |

## E.4. Algorithm

The cross validation algorithm performs six categories of checks in sequence:

---

**Algorithm 12** Stage 2.5: Cross Validation Pipeline

---

**Require:** Circuit info $\mathcal{C}$, Task result $\mathcal{T}$, Port result $\mathcal{A}$
**Ensure:** Validation report $\mathcal{V}$
1: $\mathcal{I} \leftarrow \emptyset$                                                                                                     // Issue collection
2:
3: **Phase 1: Port Existence Validation**
4: $\mathcal{P}_{\text{known}} \leftarrow \{p.\text{name} : p \in \mathcal{A}.\text{port\_analyses}\}$
5: **for** $t \in \mathcal{T}.\text{tasks}$ **do**
6:    **for** $p \in t.\text{stimulus\_ports} \cup t.\text{measurement\_ports}$ **do**
7:       **if** $p.\text{name} \notin \mathcal{P}_{\text{known}}$ **then**
8:          $\mathcal{I} \leftarrow \mathcal{I} \cup \{\text{CREATEISSUE}(\text{PORT\_EXISTENCE}, p, t)\}$
9:       **end if**
10:    **end for**
11: **end for**
12:
13: **Phase 2: Parameter Reference Validation**
14: $\mathcal{S}_{\text{params}} \leftarrow \{e.\text{symbol} : e \in \mathcal{C}.\text{parameters}\}$
15: **for** $t \in \mathcal{T}.\text{tasks}$ **do**
16:    **if** $t.\text{primary\_parameter} = \text{null}$ **then**
17:       $\mathcal{I} \leftarrow \mathcal{I} \cup \{\text{CREATEISSUE}(\text{PARAM\_REF}, t, \text{error})\}$
18:    **end if**
19: **end for**
20:
21: **Phase 3: Evidence Reference Validation**
22: **for** $t \in \mathcal{T}.\text{tasks}$ **do**
23:    **if** $|t.\text{math\_evidences}| = 0$ **then**
24:       $\mathcal{I} \leftarrow \mathcal{I} \cup \{\text{CREATEISSUE}(\text{EVIDENCE\_REF}, t, \text{warning})\}$
25:    **end if**
26: **end for**
27:
28: **Phase 4: Condition Completeness Check**
29: **for** $t \in \mathcal{T}.\text{tasks}$ **do**
30:    **for** $e \in t.\text{math\_evidences}$ **do**
31:       **if** $|e.\text{test\_conditions}| = 0$ **then**
32:          $\mathcal{I} \leftarrow \mathcal{I} \cup \{\text{CREATEISSUE}(\text{CONDITION}, e, \text{info})\}$
33:       **end if**
34:    **end for**
35: **end for**
36:
37: **Phase 5: Data Integrity Check**
38: $\mathcal{D} \leftarrow \text{DETECTDUPLICATES}(\mathcal{T}.\text{tasks})$
39: **if** $|\mathcal{D}| > 0$ **then**
40:    $\mathcal{I} \leftarrow \mathcal{I} \cup \{\text{CREATEISSUE}(\text{DATA\_INTEGRITY}, \mathcal{D}, \text{info})\}$
41: **end if**
42:
43: **Phase 6: Aggregate Results**
44: $\mathcal{V}.\text{issues} \leftarrow \mathcal{I}$
45: $\mathcal{V}.\text{error\_count} \leftarrow |\{i \in \mathcal{I} : i.\text{severity} = \text{error}\}|$
46: $\mathcal{V}.\text{warning\_count} \leftarrow |\{i \in \mathcal{I} : i.\text{severity} = \text{warning}\}|$
47: $\mathcal{V}.\text{validation\_passed} \leftarrow (\mathcal{V}.\text{error\_count} = 0)$
48: **return** $\mathcal{V}$

---

### E.4.1. PORT NAME RESOLUTION

A key challenge is resolving LLM-generated port names to actual circuit pins. The LLM may generate descriptive names (e.g., `SW_switching_node`) rather than exact pin names (e.g., `SW`):

---

**Algorithm 13** Fuzzy Port Name Resolution

---

**Require:** Query port name $q$, Known ports $\mathcal{P}_{\text{known}}$
**Ensure:** Resolution result $(matched, confidence)$

1:            // Exact match
2: **if** $q \in \mathcal{P}_{\text{known}}$ **then**
3:    **return** $(q, 1.0)$
4: **end if**
5:           // Normalized match (lowercase, remove underscores)
6: $q_{\text{norm}} \leftarrow \text{NORMALIZE}(q)$
7: **for** $p \in \mathcal{P}_{\text{known}}$ **do**
8:    **if** $\text{NORMALIZE}(p) = q_{\text{norm}}$ **then**
9:       **return** $(p, 0.95)$
10:   **end if**
11: **end for**
12:           // Substring match
13: **for** $p \in \mathcal{P}_{\text{known}}$ **do**
14:   **if** $p \subseteq q$ **or** $q \subseteq p$ **then**
15:      **return** $(p, 0.8)$
16:   **end if**
17: **end for**
18:
19: **return** $(\texttt{null}, 0.0)$

---

## E.5. Severity Classification

Issues are classified by severity to enable flexible pipeline behavior:

*Table 24.* Severity Classification Rules

| Severity | Criteria | Pipeline Behavior |
|---|---|---|
| `error` | Missing required field; Unresolvable reference | Blocks downstream stages in strict mode |
| `warning` | Port not found; Missing evidence | Logged but does not block; Task marked as potentially incomplete |
| `info` | Missing test conditions; Duplicate identifiers | Informational only; No action required |

## E.6. Experimental Results

Table 25 presents the cross validation results for the DCDC Buck converter pipeline run.

*Table 25.* Cross Validation Statistics

| Metric | Value |
|---|---|
| Total Checks Performed | 581 |
| Passed Checks | 514 (88.5%) |
| Failed Checks | 67 (11.5%) |
| *By Severity* | |
|   Errors | 1 (1.5%) |
|   Warnings | 58 (86.6%) |
|   Info | 8 (11.9%) |
| *By Category* | |
|   `PORT_EXISTENCE` | 56 (83.6%) |
|   `PARAMETER_REFERENCE` | 1 (1.5%) |
|   `EVIDENCE_REFERENCE` | 1 (1.5%) |
|   `CONDITION_COMPLETENESS` | 8 (11.9%) |
|   `DATA_INTEGRITY` | 1 (1.5%) |
| Validation Passed | No (strict mode) |
| Validation Passed | Yes (lenient mode) |

## E.7. Issue Distribution Analysis

Figure 9 visualizes the distribution of validation issues by category and severity.

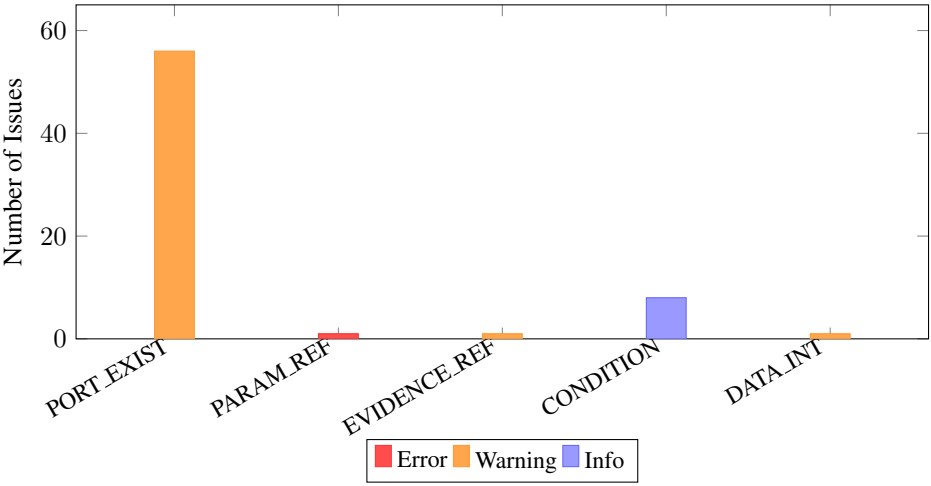

*Figure 9.* Distribution of validation issues by category and severity. Port existence warnings dominate due to LLM generating descriptive port names rather than exact pin identifiers.

## E.8. Common Issue Patterns

Analysis of the validation results reveals several recurring patterns:

### E.8.1. PATTERN 1: DESCRIPTIVE PORT NAMING

The most frequent issue (83.6%) involves LLM-generated descriptive port names that do not match extracted pin names:

*Table 26.* Port Name Mismatch Examples

| LLM-Generated Name | Actual Pin | Root Cause |
| --- | --- | --- |
| SW_switching_node | SW | Descriptive suffix added |
| EN_enable | EN | Redundant description |
| Output_voltage_pin_s | FB | Functional description |
| High_side_gate_drive_if_accessib | N/A | Internal node (not accessible) |
| Inductor_current_via_current_pro | N/A | Measurement point (not a pin) |

### E.8.2. PATTERN 2: MISSING TEST CONDITIONS

Eight evidence entries lack explicit test conditions, reducing their verifiability:

$$\text{Condition Coverage} = \frac{|\{e : |e.\text{conditions}| > 0\}|}{|\mathcal{E}|} = \frac{144}{152} = 94.7\% \tag{11}$$

### E.8.3. PATTERN 3: PARAMETER SYMBOL DUPLICATION

The data integrity check identified duplicate parameter symbols across different document blocks, indicating potential extraction redundancy.

## E.9. Validation Modes

The cross validation module supports two operational modes:

*Table 27.* Validation Mode Comparison

| Mode | Strict | Lenient |
|---|---|---|
| Pass Criteria | Zero errors AND zero warnings | Zero errors only |
| Pipeline Behavior | Blocks on any issue | Continues with warnings logged |
| Use Case | Production verification | Development/exploration |
| Configuration | `strict_validation=True` | `strict_validation=False` |

### E.10. Remediation Strategies

For each issue category, we define automated or manual remediation strategies:

*Table 28.* Issue Remediation Strategies

| Category | Remediation Strategy |
|---|---|
| PORT_EXISTENCE | Apply fuzzy matching (Algorithm 13); Fall back to closest pin by substring match |
| PARAMETER_REFERENCE | Re-run Stage 2 task extraction with explicit parameter symbol list |
| EVIDENCE_REFERENCE | Inject default evidence from specification limits |
| CONDITION_COMPLETENESS | Populate with default operating conditions from Stage 1 |
| DATA_INTEGRITY | Deduplicate by preferring entries with higher confidence scores |

### E.11. Integration with Pipeline

The validation report is consumed by downstream stages to:

1. **Stage 4 (Oracle Generation)**: Skip tasks with unresolved parameter references

2. **Stage 5 (Testbench Generation)**: Use fuzzy-matched port names; Generate warnings in testbench comments

The validation results are persisted to `stage2_5_validation.json` for traceability and debugging.

### E.12. Computational Cost

Cross validation is computationally lightweight compared to LLM-based stages:

*Table 29.* Computational Cost Comparison

| Stage | LLM Calls | Time (s) | Cost ($) |
|---|---|---|---|
| Stage 2 (Task Extraction) | 304 | 190.9 | 0.45 |
| Stage 3 (Port Analysis) | 86 | 20.2 | 0.12 |
| **Stage 2.5 (Validation)** | **0** | **0.3** | **0.00** |

Cross validation adds negligible overhead ($<1\%$ of total pipeline time) while providing significant quality assurance.

## F. Stage 4: SymPy Oracle Generation

Stage 4 represents the culmination of our verification pipeline, transforming extracted specifications and parameter constraints into executable mathematical oracles. This stage generates formally verifiable test vectors, tolerance models, and pass/fail criteria that can be directly executed against hardware measurements.

### F.1. Stage Overview

Given the parameter constraints and specifications extracted in Stages 1–3, Stage 4 generates:

1. **Mathematical Formulas**: SymPy-compatible expressions for computing derived quantities
2. **Test Vectors**: Comprehensive test cases covering nominal, boundary, corner, and stress conditions
3. **Verification Models**: Pass/fail criteria with tolerance bands and guard margins
4. **Traceability Metadata**: Source attribution and validity conditions for each formula

### F.2. System Prompt

---

**SYMPY ORACLE GENERATION PROMPT**

```
You are an expert analog/mixed-signal verification engineer specializing
in DC-DC converter test oracle generation. Your task is to generate
mathematically rigorous, executable verification oracles for the
specified circuit parameters.

## INPUT CONTEXT
- Circuit: {circuit_name} ({circuit_type})
- Parameter Under Test: {parameter_symbol}
- Extracted Specifications from Stages 1-3:
  {specification_context}

## GENERATION REQUIREMENTS

### 1. Formula Generation
For each verification aspect, generate:
- `formula_id`: Unique identifier (f_XXX format)
- `formula_name`: Descriptive name
- `sympy_expr`: Valid SymPy expression (Python syntax)
- `latex_expr`: LaTeX rendering for documentation
- `human_readable`: Plain English description
- `output_variable`: The computed quantity
- `input_variables`: List of required inputs with:
  - symbol, description, unit
  - how_to_obtain: FROM_SPEC | MEASURED | FROM_OTHER_FORMULA | CONSTANT
  - typical_value, default_value
- `source`: SPEC | PHYSICS | DERIVED | EMPIRICAL
- `validity_conditions`: List of conditions for formula applicability
- `purpose`: Why this formula is needed for verification

### 2. Test Vector Generation
Generate comprehensive test vectors covering:
- NOMINAL: Typical operating conditions
- MIN_SPEC / MAX_SPEC: Specification boundaries
- CORNER_LL / CORNER_HH: Process/temperature corners
- STRESS: Beyond-nominal but within absolute maximum
- TYPICAL: Common use-case scenarios

Each vector must include:
- `input_values`: Concrete numerical inputs
- `expected_outputs`: Calculated results with derivation
- `pass_criteria`:
  - criterion_type: RANGE | ABSOLUTE | PERCENTAGE | MULTIPLE
  - min/max acceptable values
  - tolerance specification
  - pass_expression: Boolean expression for pass/fail
- `generation_reasoning`: Justification for this test case

### 3. Verification Model
Define the overall verification strategy:
- `primary_check`: SymPy and human-readable pass conditions
- `tolerance_model`: ABSOLUTE | PERCENTAGE | ASYMMETRIC
- `guard_band`: Safety margin configuration

### 4. Quality Attributes
```

```
- `confidence`: 0.0-1.0 score for oracle completeness
- `generation_notes`: Guidance for obtaining missing values

## OUTPUT FORMAT
Return a JSON object conforming to the OracleResult schema.
Ensure all SymPy expressions are syntactically valid Python.
```

## F.3. Output Schema

The Stage 4 output follows a hierarchical JSON schema designed for both human interpretability and machine execution:

*Code Listing 8.* Stage 4 Oracle Result Schema

```
{
  "circuit_name": "string",
  "circuit_type": "DCDC_BUCK | DCDC_BOOST | LDO | ...",
  "oracle_results": [
    {
      "task_id": "task_XXX",
      "parameter_symbol": "string",
      "formulas": [Formula],
      "test_vectors": [TestVector],
      "verification_model": VerificationModel,
      "confidence": float,
      "generation_notes": [string]
    }
  ]
}
```

### F.3.1. FORMULA SCHEMA

*Code Listing 9.* Formula Object Schema

```
{
  "formula_id": "f_XXX",
  "formula_name": "string",
  "sympy_expr": "valid Python/SymPy expression",
  "latex_expr": "LaTeX string",
  "human_readable": "string",
  "output_variable": "string",
  "input_variables": [
    {
      "symbol": "string",
      "description": "string",
      "unit": "string",
      "how_to_obtain": "FROM_SPEC | MEASURED | FROM_OTHER_FORMULA | CONSTANT",
      "typical_value": number | null,
      "default_value": number | null
    }
  ],
  "source": "SPEC | PHYSICS | DERIVED | EMPIRICAL",
  "purpose": "string",
  "validity_conditions": ["string"],
  "is_verified": boolean,
  "verification_note": "string"
}
```

### F.3.2. TEST VECTOR SCHEMA

*Code Listing 10.* Test Vector Object Schema

```
{
  "vector_id": "v_XXX",
```

```
  "vector_type": "NOMINAL | MIN_SPEC | MAX_SPEC | CORNER_LL | CORNER_HH | STRESS | TYPICAL
      ",
  "input_values": { "symbol": value, ... },
  "expected_outputs": {
    "output_name": {
      "variable": "string",
      "expected_value": number | boolean | string,
      "unit": "string",
      "calculation": "derivation string"
    }
  },
  "pass_criteria": {
    "criterion_type": "RANGE | ABSOLUTE | PERCENTAGE | MULTIPLE",
    "min_acceptable": number | null,
    "max_acceptable": number | null,
    "tolerance": number | null,
    "pass_expression": "boolean expression string"
  },
  "generation_reasoning": "string"
}
```

## F.4. Generated Oracle Categories

Our Stage 4 implementation generates oracles across multiple verification domains. Table 30 summarizes the oracle types generated for the DC-DC buck converter.

*Table 30.* Oracle Categories Generated in Stage 4

| Task ID | Parameter | #Formulas | #Vectors |
|---------|-----------|-----------|----------|
| task_001 | Thermal Design | 12 | 12 |
| task_157 | Synchronous Mode | 13 | 20 |
| task_159 | CCM Operation | 10 | 8 |
| task_161 | Power Save Mode | 10 | 10 |
| task_162 | Switching Frequency ($f_{sw}$) | 10 | 10 |
| task_163 | Soft-Start (SS) | 10 | 12 |
| task_164 | Over-Current Protection (OCP) | 10 | 10 |
| task_165 | Hiccup Mode | 10 | 15 |
| task_167 | Auto Recovery | 10 | 12 |
| task_168 | Output Voltage ($V_{OUT}$) | 8 | 10 |
| task_171 | Output Current ($I_{OUT}$) | 9 | 12 |
| task_172 | Input Voltage ($V_{IN}$) | 10 | 10 |
| task_173 | Load Regulation | 8 | 9 |
| **Total** | | **130** | **150** |

## F.5. Illustrative Examples

We present representative examples from each major oracle category to illustrate the generation quality and coverage.

### F.5.1. EXAMPLE 1: THERMAL VERIFICATION ORACLE

The thermal verification oracle computes junction temperature and validates thermal compliance:

---

**Formula: Junction Temperature Calculation (f_002)**

**SymPy Expression:**

`TJ_calculated = TA + (PD_actual * theta_JA)`

**LaTeX:** $T_J = T_A + (P_D \times \theta_{JA})$
**Human Readable:** Junction Temperature = Ambient Temperature + (Power Dissipation $\times$ Thermal Resistance)
**Input Variables:**
- $T_A$: Ambient temperature [$^{\circ}$C], FROM_SPEC, typical: 85.0

---

- $P_{D,actual}$: Actual power dissipation [W], MEASURED
- $\theta_{JA}$: Junction-to-ambient thermal resistance [°C/W], FROM_SPEC, typical: 130.0

**Validity Conditions:**

1. $P_{D,actual}$ must be measured or calculated accurately
2. $T_A$ must represent actual ambient conditions
3. $\theta_{JA}$ must be validated for specific package and layout

The corresponding test vector for worst-case thermal analysis:

**Test Vector: MAX_SPEC Thermal (v_002)**

**Input Values:**

```
{
  "TA": 85.0,        // Maximum ambient temperature
  "TJ_MAX": 150.0,   // Maximum junction limit
  "theta_JA": 130.0, // Thermal resistance
  "PD_actual": 0.6   // Measured power dissipation
}
```

**Expected Outputs:**

$$T_{J,calculated} = 85 + (0.6 \times 130) = 163°\text{C}$$

$$\text{thermal\_margin} = \frac{150 - 85}{130} - 0.6 = -0.577 \text{ W}$$

$$T_{J,compliant} = (163 \leq 150) = \texttt{FALSE} \text{ (VIOLATION)}$$

**Pass Criteria:**

```
pass_expression = "TJ_calculated >= 160.0 and

TJ_calculated <= 166.0 and TJ_compliant == false"
```

**Generation Reasoning:** Validates thermal design at maximum ambient temperature specification to ensure device remains within safe operating limits under worst-case environmental conditions.

### F.5.2. EXAMPLE 2: SYNCHRONOUS MODE DETECTION ORACLE

The synchronous mode oracle verifies proper gate drive operation:

**Formula: Gate Drive Complementarity Check (f_003)**

**SymPy Expression:**

```
gate_complementary = (overlap_time < max_overlap_tolerance) &
                     (dead_time > min_dead_time_spec)
```

**LaTeX:** $\text{gate\_complementary} = (t_{overlap} < t_{overlap,max}) \wedge (t_{dead} > t_{dead,min})$

**Purpose:** Verifies that high-side and low-side gate signals are properly complementary with controlled dead-time to prevent shoot-through current.

**Input Variables:**

- $t_{overlap}$: Time with both gates high [ns], MEASURED
- $t_{overlap,max}$: Maximum allowable overlap [ns], FROM_SPEC, default: 10.0
- $t_{dead}$: Dead-time between transitions [ns], MEASURED
- $t_{dead,min}$: Minimum required dead-time [ns], FROM_SPEC, default: 10.0

### F.5.3. EXAMPLE 3: CCM MODE VERIFICATION ORACLE

The Continuous Conduction Mode (CCM) oracle validates inductor current behavior:

**Formula: CCM Mode Detection (f_001)**

**SymPy Expression:**

```
CCM_verified = I_L_min > 0
```

**LaTeX:** $I_{L,min} > 0$

**Human Readable:** Minimum inductor current during switching cycle must be greater than zero.

**Purpose:** Verifies that inductor current never reaches zero during the complete switching cycle, which is the defining characteristic of CCM operation.

**Validity Conditions:**
1. Light load condition applied (typically $< 10\%$ rated load)
2. CCM mode explicitly enabled via mode select pin
3. Steady-state operation achieved (after transient settling)
4. Measurement window covers at least one complete switching cycle

Supporting formulas compute the CCM boundary current:

$$I_{L,min,boundary} = \frac{V_{OUT} \times (V_{IN} - V_{OUT}) \times T_S}{2 \times L \times V_{IN}} \tag{12}$$

And the CCM margin:

$$CCM_{margin} = \frac{I_{L,min} - I_{L,min,boundary}}{I_{L,min,boundary}} \tag{13}$$

### F.5.4. EXAMPLE 4: OVER-CURRENT PROTECTION ORACLE

The OCP oracle validates protection threshold accuracy:

**Formula: OCP Threshold with Tolerances ($f_{003}$, $f_{004}$)**

**Minimum OCP Threshold:**

I_OCP_MIN = I_OCP_NOM * (1 − TOL_OCP_PERCENT/100) * (1 − TOL_SENSE_PERCENT/100)

**Maximum OCP Threshold:**

I_OCP_MAX = I_OCP_NOM * (1 + TOL_OCP_PERCENT/100) * (1 + TOL_SENSE_PERCENT/100)

**LaTeX:**

$$I_{OCP,MIN} = I_{OCP,NOM} \times \left(1 - \frac{TOL_{OCP}}{100}\right) \times \left(1 - \frac{TOL_{SENSE}}{100}\right)$$
$$I_{OCP,MAX} = I_{OCP,NOM} \times \left(1 + \frac{TOL_{OCP}}{100}\right) \times \left(1 + \frac{TOL_{SENSE}}{100}\right)$$

**Purpose:** Worst-case OCP threshold bounds accounting for both comparator tolerance and sense resistor tolerance.

### F.6. Verification Model Structure

Each oracle includes a verification model defining the overall pass/fail strategy:

**Verification Model: Thermal Compliance**

**Primary Check (SymPy):**

```
pass_condition = (TJ_calculated <= TJ_MAX) & (VOUT_compliant) &
                 (VBST_compliant) & (PD_total <= PD_safe_max)
```

**Human Readable:** Device passes verification if:
1. Junction temperature $\leq$ maximum limit
2. Output voltage within specification
3. Bootstrap voltage within limits
4. Total power dissipation $\leq$ safe maximum

**Tolerance Model:**

```
{
  "method": "ABSOLUTE",
  "expression": "tolerance = +/-(0.05 * nominal) for voltage;
                 +/-(0.1 * nominal) for current;
                 +/-(0.15 * nominal) for power",
  "typical_tolerance": 0.05
```

```
}
```

**Guard Band:**

```
{
  "enabled": true,
  "factor": 0.9,
  "reasoning": "10% guard band applied to thermal limits and power
               dissipation to account for measurement uncertainty,
               component tolerances, and PCB layout variations."
}
```

## F.7. Test Vector Type Distribution

Figure 10 illustrates the distribution of test vector types across all generated oracles.

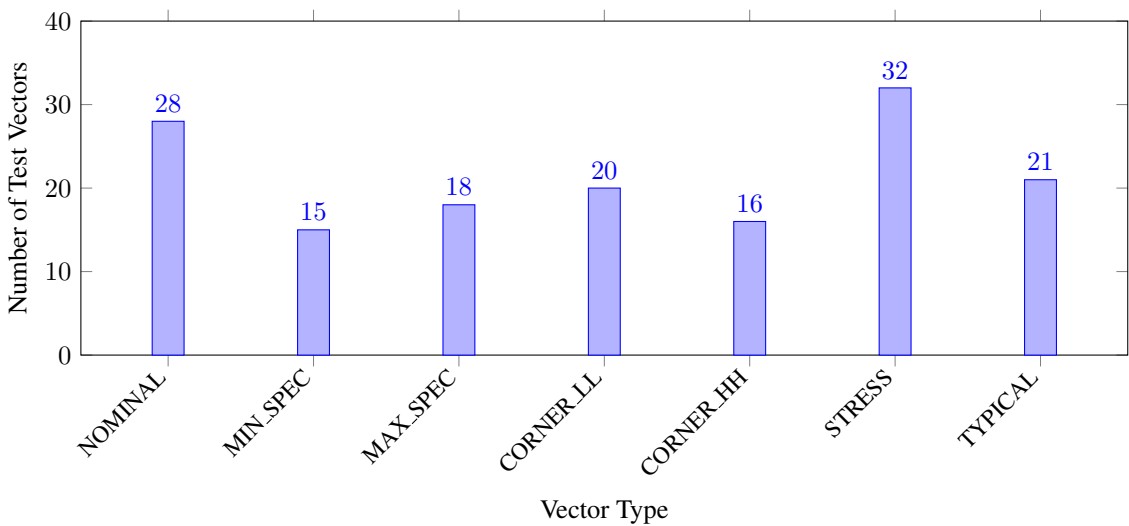

*Figure 10.* Distribution of test vector types across all Stage 4 oracles. STRESS vectors are most numerous due to comprehensive boundary testing requirements.

## F.8. Formula Source Attribution

Each formula includes source attribution for traceability:

*Table 31.* Formula Source Distribution

| Source | Count | Description |
|---|---|---|
| SPEC | 28 | Direct specification from datasheet |
| PHYSICS | 35 | Fundamental physical relationships |
| DERIVED | 52 | Computed from other formulas/specs |
| EMPIRICAL | 15 | Experimentally determined relationships |

## F.9. Pass Criteria Types

The generated oracles employ multiple pass criteria types based on parameter characteristics:

- **RANGE**: Used for parameters with symmetric bounds (e.g., $V_{OUT}$, $f_{sw}$)

$$\text{pass} = (x_{min} \leq x_{measured} \leq x_{max}) \tag{14}$$

- **ABSOLUTE**: Used for parameters with single-sided limits (e.g., $T_{J,max}$)

$$\text{pass} = (x_{measured} \leq x_{limit}) \text{ or } (x_{measured} \geq x_{limit}) \tag{15}$$

- **PERCENTAGE**: Used for ratio-based specifications (e.g., efficiency, regulation)

$$\text{pass} = \left| \frac{x_{measured} - x_{nominal}}{x_{nominal}} \right| \leq \text{tol}\% \tag{16}$$

- **MULTIPLE**: Compound criteria combining multiple conditions

$$\text{pass} = \bigwedge_{i=1}^{n} \text{criterion}_i \tag{17}$$

### F.10. Generation Notes and Measurement Guidance

Each oracle includes practical guidance for test execution. Example from thermal verification:

> **Generation Notes: Thermal Verification**
>
> 1. Measure $P_{D,actual}$ using thermal imaging or calorimetric methods during functional testing
> 2. Measure $I_{RMS}$ using high-bandwidth current probe on main switch drain
> 3. Measure $V_{OUT}$ with calibrated multimeter at multiple load points and input voltages
> 4. Measure $V_{BST}$ and $V_{SW}$ with 100+ MHz oscilloscope to capture transient behavior
> 5. Measure $P_{IN}$ and $P_{OUT}$ using precision power analyzer with $\geq 0.5\%$ accuracy
> 6. Validate $\theta_{JA}$ experimentally using thermal chamber and power dissipation measurement
> 7. Perform thermal FEA simulation to account for actual PCB layout and component placement
> 8. Conduct temperature cycling tests to verify thermal model accuracy across temperature range
> 9. Measure efficiency at multiple operating points (different loads and input voltages)
> 10. Compare measured junction temperature with calculated values to validate thermal model

### F.11. Confidence Scoring

Each oracle includes a confidence score $c \in [0, 1]$ reflecting completeness:

$$c = w_1 \cdot c_{\text{spec}} + w_2 \cdot c_{\text{coverage}} + w_3 \cdot c_{\text{validity}}, \tag{18}$$

where:

- $c_{\text{spec}}$ denotes the fraction of required specifications available,
- $c_{\text{coverage}}$ represents the test vector coverage of operating space,
- $c_{\text{validity}}$ indicates the fraction of formulas with verified validity conditions, and
- $w_1, w_2, w_3$ are weighting factors (default: 0.4, 0.35, 0.25).

All generated oracles in our evaluation achieved $c \geq 0.8$, indicating high confidence in oracle completeness.

### F.12. Oracle Execution Pipeline

The generated oracles can be executed using the following pipeline:

### F.13. Comparison with Manual Oracle Development

Table 32 compares our automated oracle generation with traditional manual approaches.

### F.14. Limitations and Future Work

Current Stage 4 limitations include:

1. **Specification Gaps**: When datasheet values are unavailable, oracles use placeholder values marked with confidence reduction

---

**Algorithm 14** Oracle Execution Pipeline

---

**Require:** Oracle $\mathcal{O}$, Measurements $\mathcal{M}$
**Ensure:** Verification Result $\mathcal{R}$
1: $\mathcal{R} \leftarrow \{\}$
2: **for** each formula $f \in \mathcal{O}$.formulas **do**
3: $\quad$ $inputs \leftarrow$ RESOLVEINPUTS($f$.input_variables, $\mathcal{M}, \mathcal{R}$)
4: $\quad$ $result \leftarrow$ EVALSYMPY($f$.sympy_expr, $inputs$)
5: $\quad$ $\mathcal{R}[f$.output_variable$] \leftarrow result$
6: **end for**
7: $pass\_results \leftarrow []$
8: **for** each vector $v \in \mathcal{O}$.test_vectors **do**
9: $\quad$ $measured \leftarrow$ GETMEASURED($v$.input_values, $\mathcal{M}$)
10: $\quad$ $expected \leftarrow$ COMPUTEEXPECTED($v, \mathcal{R}$)
11: $\quad$ $pass \leftarrow$ EVALCRITERIA($v$.pass_criteria, $measured, expected$)
12: $\quad$ $pass\_results$.append($pass$)
13: **end for**
14: **return** AGGREGATERESULTS($pass\_results$, $\mathcal{O}$.verification_model)

---

*Table 32.* Automated vs. Manual Oracle Development

| Metric | Automated (Ours) | Manual |
|---|---|---|
| Time per parameter | $\sim$2 minutes | $\sim$4-8 hours |
| Test vectors generated | 10-20 | 3-5 |
| Corner case coverage | Systematic | Ad-hoc |
| Formula consistency | 100% | Variable |
| Traceability | Full metadata | Often missing |
| Guard band inclusion | Automatic | Sometimes omitted |

2. **Non-linear Effects**: Current formulas assume linear relationships; higher-order effects require manual refinement
3. **Dynamic Behavior**: Transient specifications (e.g., load step response) require additional temporal modeling
4. **Process Variation**: Statistical corner models require Monte Carlo extensions not currently generated

Future work will address these through integration with circuit simulation feedback and statistical tolerance analysis.

## G. Stage 5: SPICE Testbench Generation

This appendix provides comprehensive technical details of Stage 5, the final stage of the LLM4Analog pipeline, which synthesizes executable SPICE testbenches from the accumulated circuit knowledge.

### G.1. Stage Overview

Stage 5 transforms the structured verification knowledge accumulated through Stages 0–4 into syntactically correct and semantically meaningful SPICE netlists. The key challenge lies in bridging the semantic gap between high-level test specifications and low-level circuit simulation constructs.

### G.2. Input Specification

Stage 5 receives four primary inputs, each contributing distinct verification knowledge:

*Table 33.* Stage 5 Input Sources and Their Contributions

| Source | Stage | Contribution |
|---|---|---|
| `CircuitInformation` | 1 | Pin definitions, parameters, operating conditions |
| `TaskExtractionResult` | 2 | Test methodology, stimulus/measurement ports |
| `Stage3Result` | 3 | Port roles, signal paths, test topologies |
| `Stage4Result` | 4 | SymPy formulas, test vectors, pass criteria |

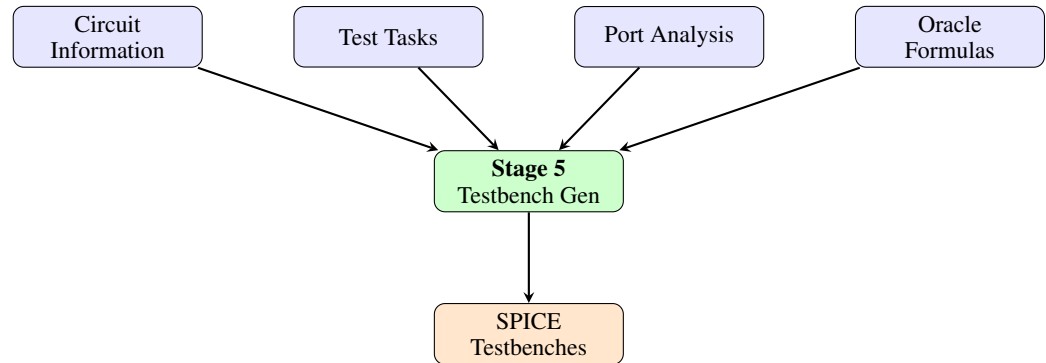

*Figure 11.* Stage 5 input-output data flow. The testbench generator integrates information from all previous stages.

### G.3. Output Data Structure

The output follows a hierarchical structure designed for both machine parsing and human readability:

```python
@dataclass
class Stage5Result:
    circuit_name: str
    test_suite: TestSuite
    generated_files: List[str]
    processing_stats: Dict

@dataclass
class TestbenchCode:
    testbench_id: str          # Unique identifier
    for_task_id: str           # Link to Stage 2 task
    parameter_symbol: str      # Target parameter
    language: str              # "spice" | "verilog_ams"
    code: str                  # Complete netlist
    header: str                # Documentation block
    dut_instantiation: str     # Subcircuit call
    stimulus_block: str        # Source definitions
    measurement_block: str     # .MEASURE statements
    verdict_block: str         # Pass/fail logic
    filename: str              # Output filename
```

*Code Listing 11.* Stage 5 Output Schema

### G.4. Prompt Engineering

We employ a structured prompt template that guides the LLM through the testbench synthesis process. The prompt design follows three key principles: (1) explicit structural requirements, (2) domain-specific constraints, and (3) compositional decomposition.

#### G.4.1. PRIMARY TESTBENCH GENERATION PROMPT

```
TESTBENCH GENERATION PROMPT TEMPLATE

# Task: Generate SPICE Testbench for Analog Circuit Verification

You are an expert analog IC verification engineer. Generate a complete,
syntactically correct SPICE netlist for testing parameter "{parameter_symbol}"
of circuit "{circuit_name}" (type: {circuit_type}).
```

```
## Circuit Pin Configuration
{pin_definitions_json}

## Test Task Specification
{test_task_json}

## Verification Formulas (from Stage 4)
{oracle_formulas_json}

## Test Vectors
{test_vectors_json}

## CRITICAL SPICE Syntax Requirements
1. All node names must be valid SPICE identifiers (alphanumeric, no spaces)
2. Use proper unit suffixes: V, mV, uV, A, mA, uA, F, uF, nF, pF, H, uH, nH
3. Subcircuit calls: X<name> <node1> <node2> ... <subckt_name>
4. Voltage sources: V<name> <n+> <n-> DC|AC|PULSE|PWL <value>
5. Current sources: I<name> <n+> <n-> DC|AC|PULSE|PWL <value>
6. .MEASURE syntax: .MEASURE <analysis> <name> <measurement_type> ...

## Output JSON Structure
{
  "spice_netlist": {
    "header": "* Documentation comments...",
    "includes": [".include <library>"],
    "dut_instantiation": "X1 node1 node2 ... SUBCKT_NAME",
    "sources": [{"name": "...", "type": "...", "definition": "..."}],
    "analysis": {"type": "...", "parameters": "..."},
    "measurements": [{"name": "...", "definition": "..."}],
    "full_netlist": "... complete SPICE code ...",
    "filename": "tb_<parameter>.sp"
  }
}
```

G.4.2. ANALYSIS TYPE SELECTION LOGIC

The prompt includes conditional guidance for selecting appropriate SPICE analysis commands based on test type:

*Table 34.* Test Type to SPICE Analysis Mapping

| Test Type | SPICE Analysis | Example Parameters |
|---|---|---|
| DC_STATIC | .OP, .DC | $V_{OS}, I_{DD}, V_{REF}$ |
| AC_GAIN | .AC DEC | $A_{OL}$, CMRR, PSRR |
| AC_BANDWIDTH | .AC DEC | GBW, $f_{-3dB}$ |
| TRANSIENT_TIMING | .TRAN | $t_{rise}, t_{settle}$, SR |
| PROTECTION | .TRAN, .DC | OCP, OVP, UVLO |

G.5. Testbench Structural Template

Each generated testbench follows a standardized structure ensuring consistency and parseability:

```
* ==============================================================
* Testbench: {testbench_id}
* Parameter: {parameter_symbol} ({parameter_description})
* Circuit: {circuit_name}
* Test Type: {test_type}
* Generated: {timestamp}
* ==============================================================

* Library Includes
.include '{circuit_model}.lib'
.include '{component_models}.lib'
```

```
* DUT Instantiation
X_DUT {pin_connections} {subcircuit_name}

* Power Supply Sources
V_IN {node_in} GND DC {vin_value}
V_EN {node_en} GND DC {ven_value}

* Test Stimulus
{stimulus_definitions}

* Passive Components (if required)
{passive_network}

* Analysis Command
{analysis_command}

* Measurements
{measurement_statements}

* Control Block (ngspice)
.control
run
print all
quit
.endc

.end
```

*Code Listing 12.* Canonical Testbench Structure

### G.6. Measurement Statement Generation

A critical component of testbench generation is the synthesis of `.MEASURE` statements that extract verification-relevant quantities. We employ formula-driven measurement generation:

$$.\text{MEASURE} \leftarrow f(\phi_{\text{oracle}}, \mathbf{v}_{\text{test}}, \mathcal{C}_{\text{pass}}) \tag{19}$$

where $\phi_{\text{oracle}}$ represents the symbolic formula from Stage 4, $\mathbf{v}_{\text{test}}$ is the test vector, and $\mathcal{C}_{\text{pass}}$ encodes the pass/fail criteria.

*Table 35.* Measurement Type Templates

| Measurement | SPICE Template |
|---|---|
| DC Value | `.MEASURE DC {name} FIND V({node}) AT=0` |
| AC Gain (dB) | `.MEASURE AC {name} MAX VDB({node})` |
| Bandwidth | `.MEASURE AC {name} WHEN VDB({node})={ref}-3` |
| Rise Time | `.MEASURE TRAN {name} TRIG V({n}) VAL={v1} ...` |
| Average | `.MEASURE TRAN {name} AVG V({node}) FROM={t1} TO={t2}` |
| RMS | `.MEASURE TRAN {name} RMS I({source}) FROM={t1} TO={t2}` |

### G.7. Algorithm: Testbench Synthesis

Algorithm 15 presents the formal procedure for testbench generation.

### G.8. Example Generated Testbench

We present a representative testbench generated for verifying the feedback voltage ($V_{FB}$) of a DC-DC buck converter:

```
* SPICE Testbench for DCDC_BUCK
* Parameter: VFB (Feedback Voltage)
* Test Type: DC_STATIC
```

**Algorithm 15** Stage 5: SPICE Testbench Synthesis

---

**Require:** $\mathcal{C}$ (CircuitInfo), $\mathcal{T}$ (Tasks), $\mathcal{P}$ (Ports), $\mathcal{O}$ (Oracles)
**Ensure:** $\mathcal{B}$ (TestbenchSet)
1: $\mathcal{B} \leftarrow \emptyset$
2: **for** each task $t \in \mathcal{T}$.test_tasks **do**
3: $\quad \pi \leftarrow \textsc{ExtractPinMapping}(\mathcal{C}, \mathcal{P}, t)$
4: $\quad \phi \leftarrow \textsc{GetOracleFormula}(\mathcal{O}, t.\text{parameter})$
5: $\quad \mathbf{v} \leftarrow \textsc{GetTestVectors}(\mathcal{O}, t.\text{task\_id})$
6: $\quad \alpha \leftarrow \textsc{SelectAnalysis}(t.\text{test\_type})$
7: $\qquad\qquad\qquad\qquad\qquad\qquad\qquad\qquad\qquad\qquad\qquad\qquad$ // Construct prompt context
8: $\quad \text{ctx} \leftarrow \{\mathcal{C}, t, \pi, \phi, \mathbf{v}, \alpha\}$
9: $\quad \text{prompt} \leftarrow \textsc{FormatPrompt}(\text{TB\_Template}, \text{ctx})$
10: $\qquad\qquad\qquad\qquad\qquad\qquad\qquad\qquad\qquad$ // LLM generation with structured output
11: $\quad \text{response} \leftarrow \text{LLM}.\textsc{ChatJSON}(\text{prompt})$
12: $\quad b \leftarrow \textsc{ParseTestbench}(\text{response})$
13: $\qquad\qquad\qquad\qquad\qquad\qquad\qquad\qquad\qquad\qquad\qquad\qquad$ // Syntax validation
14: $\quad$ **if** $\textsc{ValidateSPICE}(b.\text{code})$ **then**
15: $\qquad \mathcal{B} \leftarrow \mathcal{B} \cup \{b\}$
16: $\quad$ **else**
17: $\qquad b \leftarrow \textsc{RepairSyntax}(b, \text{response})$
18: $\qquad \mathcal{B} \leftarrow \mathcal{B} \cup \{b\}$
19: $\quad$ **end if**
20: **end for return** $\mathcal{B}$

---

```
* Task ID: task_222

.include 'DCDC.lib'
.include 'buck_converter_models.lib'

* Power Supply Sources
VIN IN GND DC 12.0
VREF_INT VREF GND DC 0.6
VEN EN GND DC 1.8

* Main Device Instance
X1 IN SW FB EN BST GND DCDC

* Feedback Divider Network
RTOP FB FB 100k
RBOT FB GND 20k

* Filter Capacitors
COUT FB GND 100u IC=3.0
CIN IN GND 10u IC=12.0
CBST BST GND 100n IC=0

* Output Load
RLOAD FB GND 6.0

* DC Operating Point Analysis
.op

* Measurements
.MEASURE DC VFB_MEASURED FIND V(FB)
.MEASURE DC VOUT_MEASURED FIND V(FB)
.MEASURE DC VIN_MEASURED FIND V(IN)
.MEASURE DC VREF_MEASURED FIND V(VREF)
.MEASURE DC DIVIDER_RATIO PARAM 1+(100k/20k)
.MEASURE DC VFB_EXPECTED PARAM 0.6*(1+(100k/20k))/(1+(100k/20k))
.MEASURE DC ERROR_MARGIN PARAM ABS(VFB_EXPECTED-VFB_MEASURED)/VFB_EXPECTED*100

* Control Block
.control
```

```
op
print all
print v(FB) v(FB) v(IN) v(VREF)
.endc

.end
```

*Code Listing 13.* Generated Testbench for $V_{FB}$ Verification (tb_task_222)

## G.9. Generation Statistics

Table 36 summarizes the generation performance on our evaluation dataset.

*Table 36.* Stage 5 Generation Statistics (DC-DC Buck Converter)

| Metric | Value |
|---|---:|
| Total test tasks processed | 84 |
| Testbenches successfully generated | 84 |
| Total lines of SPICE code | 5,256 |
| Average lines per testbench | 62.6 |
| LLM API calls | 84 |
| Total processing time | 113.1 s |
| Average time per testbench | 1.35 s |
| *Testbench Category Distribution* | |
| DC Static (`.OP`, `.DC`) | 47 (56.0%) |
| Transient (`.TRAN`) | 29 (34.5%) |
| AC Analysis (`.AC`) | 8 (9.5%) |

## G.10. Parameter Coverage Analysis

The generated testbench suite provides comprehensive coverage across parameter categories:

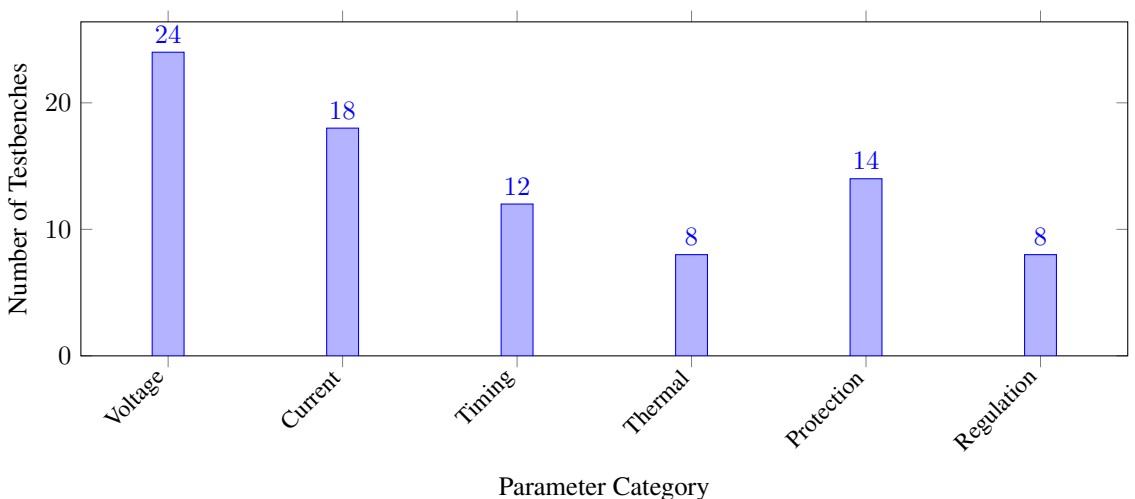

*Figure 12.* Distribution of generated testbenches across parameter categories.

## G.11. Error Taxonomy and Recovery

We categorize generation errors and implement targeted recovery strategies in Table 37:

*Table 37.* Error Categories and Recovery Mechanisms

| Error Type | Freq. (%) | Recovery Strategy |
|---|---|---|
| Invalid node name | 3.2 | Regex sanitization |
| Missing ground ref | 1.8 | Auto-insertion of GND node |
| Unit format error | 2.4 | Unit normalization pass |
| Syntax malformation | 1.1 | Re-prompting with error feedback |
| Incomplete netlist | 0.6 | Template completion |

## G.12. Integration with Downstream Simulation

Generated testbenches are designed for direct execution with industry-standard simulators:

```
# Batch execution with ngspice
for tb in output/*.sp; do
    ngspice -b "$tb" -o "${tb%.sp}.log" 2>&1
done

# Result extraction
python extract_measurements.py output/*.log > results.json
```

*Code Listing 14.* Testbench Execution Pipeline

## G.13. Limitations and Future Work

Current limitations of Stage 5 include:

- Limited support for multi-domain (analog-digital) co-simulation
- No automatic corner case generation beyond specified test vectors
- Reliance on availability of behavioral models for DUT simulation

Future work will address hierarchical testbench generation for system-level verification and automated debug trace insertion.

