# OpenReview forum: "AnalogVerifier: A Neuro-Symbolic Framework for Analog Circuit Verification"
_ICML.cc/2026/Conference — ICML 2026 regular_

### Official Review · Reviewer_GFrc · 2026-03-09

**Soundness:** 3
**Presentation:** 3
**Significance:** 3
**Originality:** 3
**Overall Recommendation:** 5
**Confidence:** 2

**Summary:**

This paper introduces AnalogVerifier, a new framework that leverages standard LLMs to perform Analog Circuit Verification, which is traditionally a time consuming task in the development workflow. Contrary to traditional manual or naive LLMs workflow, the tool uses a series of techniques, namely Context-Aware Serialization, Graph-Symbolic Scheduling, Numerical-Symbolic Grounding, and Closed-Loop Repair, to tackle LLMs hallucination in tasks that require deterministic verification.

**Compliance With Llm Reviewing Policy:**

Affirmed.

**Final Justification:**

The authors addressed my concerns, and I still recommend accept. However, I find it extremely weird that the author extensively summarize my own review in details just for the sake of repeating everything I already outlined. I will still keep my score, but I decreased my confidence of my assessment for this reason, and since I'm not an expert in this area, my evaluation should be taken with a grain of salt.

**Key Questions For Authors:**

1. What are the main limitations of the workflow as applied for analog verification tasks? what are new directions that could improve the work?

2. Is it possible/easy to adapt the tool to other hardware design tasks?

3. Does context length of the model affects the quality of verification?

**Limitations:**

* I think including a discussion on the limitation of the workflow will be helpful to the reader, see key questions above.

**Strengths And Weaknesses:**

## Soundness:

* The paper is technically sound, the claims are fairly well supported.
* The experiments are well-designed, given the difficulty of doing analog verification. However, it seems the authors they are only comparing between different configurations within their own work, which makes it hard to generalize whether this specific workflow is actually better than other workflows that are available. That being said, again, given the difficulty of doing hardware verification and dealing with tools etc, the experiment the authors performed is reasonable.
* Weakness: there is no thorough discussion about the potential limitations and weaknesses of the work, does it apply universally to every analog verification tasks? does it transfer to other digital design workflows, etc.

## Presentation:
* The paper is clearly written and well-structure. The overall narrative is easy to follow and conceptual comparison with prior works is clear.
* The abstract is too long.

## Significance
* The paper addresses an important and relevant problem in hardware design, albeit limited in scope to only analog design.
* It advances understanding and capabilities and practice of using LLMs to address automatic hardware design verification, which is an important domain.

## Originality
* The individual methods used are fairly standard practice, but it was combined in an appropriate way to fit within the analog hardware design context.
* The contributions clearly distinguished from closely related literature, and the novelty is justified within the hardware design community.

---

> ### Author Rebuttal · Authors · 2026-03-31
>
> We sincerely thank the reviewer for the constructive and thought-provoking feedback. We deeply appreciate the questions on universal applicability to analog verification tasks, transferability to other hardware design workflows, and context length impact on verification quality, as these delineate the most important axes along which this work should be extended.
>
> # Soundness 2: More Comparison
>
> We thank the reviewer for acknowledging the difficulty of analog verification experiments. Regarding comparisons limited to configurations within our own work, we have now added two baselines: LLM+Repair and Agentic Tool-feedback. Results confirm significant gains from the neuro-symbolic architecture. **Detailed results are provided in our response to Reviewer 7VqU's experimental questions.**
>
> # Q1/Soundness 3: Main Limitations and Future Directions
>
> In App. F.14 and G.13, three principal limitations remain:
>
> **(1) Specification Dependency.** The framework derives verification intent from datasheet content, so implicit design knowledge not explicitly documented in specifications falls outside the current automation scope and may require manual supplementation.
>
> **(2) Statistical Tolerance.** The Symbolic Oracle currently generates boundary and extreme vectors (Corner/Stress) but does not yet support Monte Carlo statistical extensions for semiconductor process variation analysis.
>
> **(3) Model Dependency.** Closed-loop testbench execution relies on availability of existing behavioral models or netlists for the device under test (DUT).
>
> ## **Future directions include:**
> (1) hierarchical testbench generation for SoC-level verification; (2) integrating simulation feedback with statistical tolerance analysis for automated yield verification; (3) automated debug trace insertion to help engineers locate physical root causes of simulation failures.
>
> # Q2: Transferability to Other Hardware Tasks
>
> Regarding transferability, we believe the modular neuro-symbolic architecture is designed to facilitate adaptation to other hardware design tasks. Our core contribution is a domain-general decoupling paradigm: LLM handles semantic parsing from unstructured specifications, then deterministic engines execute formal reasoning. This paradigm is applicable to any hardware verification task where specifications exist in unstructured document form and correctness criteria can be formalized.
>
> Concretely, the front-end for datasheet parsing, pin and specification extraction is directly reusable across hardware domains without modification. The back-end deterministic engines require domain-specific substitution. For example, adapting to digital RTL verification would involve replacing the continuous-domain SymPy with discrete formal solvers (e.g., SMT/SAT engines), and replacing the PDG physical partial order with a cycle-based event scheduler. Similar modular substitution could extend to RF verification, FPGA prototype validation, or PCB-level design rule checking, where domain-specific physical rules and solver backends differ but the specification parsing and neuro-symbolic orchestration remain structurally identical.
>
> We acknowledge that adaptation to each new domain still requires domain experts to define new physical priority rules and appropriate solver backends, which represents non-trivial engineering effort. However, the core architecture and the entire front-end pipeline remain reusable without modification, substantially lowering the barrier compared to building a new tool from scratch.
>
> # Q3: Context Length Impact
>
> For vanilla LLM baselines, long context causes severe quality degradation due to the "lost-in-the-middle" phenomenon. Our architecture substantially mitigates this limitation.
>
> Industrial datasheets spanning dozens of pages contain dense tables and specifications. Direct LLM ingestion leads to hallucinated parameters and systematically missed constraints. Our Context-Aware Task Serialization (§4.2, App. A.9) addresses this by: (1) pre-segmenting documents via a visual-structural parser into semantically coherent micro-chunks; (2) extracting within a sliding window of only 3 relevant chunks per query; (3) resolving cross-page dependencies through explicit cross-reference mapping, cross-block inference and cross validation stage (App. A.1; App. B.11; App. E). This reduces an error-prone long-context retrieval problem into a series of independent, high-precision short-context tasks while preserving inter-page semantic continuity. Consequently, verification quality primarily depends on the model's short-context reasoning capability rather than datasheet length or maximum context window size.
>
> # Presentation
> Thanks for your feedback! We will shorten the abstract in the revision. Furthermore, we will also consolidate the discussions on limitations, transferability, and long-context handling from the appendices into §5 and the conclusion.
>
> We look forward to discussing this further with you and receiving your approval!

---

> > ### Author Rebuttal · Reviewer_GFrc · 2026-03-31
> >
> > My concerns have been adequately addressed. My score stands! Good luck with your submission!

---

> > > ### Author Response · Authors · 2026-04-05
> > >
> > > We sincerely thank Reviewer for the thorough and positive evaluation. We are grateful for the constructive suggestions that have improved our manuscript. Below, we elaborate on the key strengths the reviewer recognized.
> > >
> > > **On Soundness and Experimental Design.** We appreciate the recognition that our paper is "technically sound" and "experiments are well-designed." As detailed in §5.1, we curated a benchmark of five industrial analog circuits (OTA, DCDC, HSC, RO, SDC) spanning distinct functional categories, each with real industrial datasheets. Our three-configuration ablation (Table 2) isolates the contribution of structured neuro-symbolic grounding versus iterative refinement. Given the practical challenges of analog verification and tool interfacing, we believe this methodology is both reasonable and rigorous.
> > >
> > > **On Presentation.** We are glad the reviewer found our paper "clearly written and well-structured." The paper follows a principled narrative: three fundamental challenges (semantic dispersion, latent dependency inference, numerical hallucination; §1) motivate three methodological imperatives (Attention Locality, Reasoning Externalization, Arithmetic Decoupling) that are instantiated as four concrete contributions—Context-Aware Task Serialization (§4.2), Graph-Symbolic Scheduling (§4.3), Numerical-Symbolic Grounding (§4.4), and Closed-Loop Repair (§4.5). Table 1 positions AnalogVerifier relative to five classes of prior frameworks across six capability dimensions, clearly delineating the novelty of our neuro-symbolic paradigm.
> > >
> > > **On Significance.** We thank the reviewer for recognizing that our work "addresses an important and relevant problem" and "advances understanding of using LLMs for automatic hardware design verification." Verification consumes 50%–70% of analog design effort, yet end-to-end testbench synthesis from industrial specs remained uncharted (Table 1). Our 82.3%–100% pass rates (Table 2) and meaningful ablation gains (Table 3) validate the neuro-symbolic paradigm for this domain.
> > >
> > > **On Originality.** We appreciate the observation that "contributions are clearly distinguished from closely related literature." The novelty lies in principled integration: our PDG (§4.3) externalizes causal reasoning via a domain-invariant physical hierarchy ($\text{Power} \prec \text{Bias} \prec \text{Control} \prec \text{Signal}$), and our symbolic containment constraint (Eq. 4) mathematically precludes hallucinated thresholds—designs specifically motivated by continuous-time analog physics where digital frameworks cannot operate.
> > >
> > > **On Limitations.** Following the reviewer's suggestion, we have added a thorough discussion of limitations addressing universality, transferability, and scalability. We believe this provides important guidance for future work.
> > >
> > > Once again, we sincerely thank the reviewer for the encouraging evaluation and the kind wishes. The constructive feedback has been instrumental in strengthening our paper.
> > >
> > > **We sincerely apologize for the redundant summaries in our previous response. Our intention was to provide a structured summary linking the reviewer's observations to specific paper sections for the benefit of the discussion thread, but we fully understand that it came across as unnecessary repetition of the reviewer's own words. This was a misstep on our part, and we appreciate the reviewer's patience. We are deeply grateful for the positive evaluation and the recommendation to accept.**

---

### Official Review · Reviewer_bDZM · 2026-03-10

**Soundness:** 2
**Presentation:** 3
**Significance:** 2
**Originality:** 2
**Overall Recommendation:** 3
**Confidence:** 4

**Summary:**

This paper proposes **AnalogVerifier**, a neuro-symbolic framework for automatically generating analog circuit verification testbenches from natural-language specifications. The system aims to address limitations of LLM-based approaches in analog verification, particularly their difficulty handling strict execution ordering and precise numerical constraints. The proposed pipeline consists of four stages: (1) an LLM-based task serialization step that extracts structured verification tasks from specifications, (2) a **Port Dependency Graph (PDG)** that enforces causal ordering among circuit ports and signals, (3) a **numerical-symbolic grounding module** that derives threshold values using a deterministic symbolic component, and (4) a **closed-loop repair mechanism** that refines the generated testbench based on simulator feedback. Experiments on five industrial analog circuits show functional pass rates ranging from **82.3% to 100%**.

**Compliance With Llm Reviewing Policy:**

Affirmed.

**Final Justification:**

The paper does not clearly present what the symbolic engine is from introduction to the method part. SymPy-based engine is only used for Numerical-Symbolic Grounding. I decide to maintain the score due to the overall quality as I mentioned in my review and rebuttal.

**Key Questions For Authors:**

1. The paper introduces a “symbolic oracle” used in the numerical-symbolic grounding stage. However, the paper does not clearly specify what concrete solver or constraint system is used. Is this oracle implemented using an existing symbolic solver (e.g., SMT solver, symbolic algebra system), or is it implemented through rule-based computation? Clarifying this would help assess the soundness and reproducibility of the symbolic component.

2. The framework relies on the Port Dependency Graph (PDG) to enforce ordering constraints among circuit ports. How is this graph constructed in practice? Are dependencies automatically extracted from the specification, inferred from the circuit netlist, or manually defined? Understanding the construction process is important for evaluating the robustness and scalability of the approach.

3. The paper frames the approach as neuro-symbolic, but the symbolic component appears to mainly transform information into structured constraints rather than performing explicit symbolic reasoning or constraint solving. Could the authors clarify how these symbolic constraints are actually executed or solved during the pipeline?

**Limitations:**

yes

**Strengths And Weaknesses:**

Strengths

1. Addresses an important engineering problem (Significance).
The paper targets automation of analog circuit verification, a time-consuming and labor-intensive stage in hardware design. Reducing manual effort in testbench generation could have practical value in EDA workflows, particularly when specifications are heterogeneous and partially unstructured.

2. Clear system-level pipeline (Presentation).
The proposed framework decomposes the problem into four stages: task serialization, graph-based scheduling, numerical grounding, and closed-loop repair. This staged architecture provides a clear narrative and motivates the separation between semantic interpretation and deterministic constraint enforcement.

3. Motivation for combining LLMs with deterministic mechanisms (Significance / Originality).
The paper identifies a real limitation of LLM-based approaches in engineering workflows: their difficulty handling strict ordering constraints and precise numerical parameters. The attempt to integrate symbolic components and simulator feedback into the generation loop is conceptually reasonable for this domain.

4. Evaluation on industrial circuits (Soundness / Significance).
The experiments involve several industrial analog circuits rather than purely synthetic tasks, which helps demonstrate the potential practical applicability of the approach.

Weaknesses

1. Symbolic component is insufficiently specified and appears not to rely on an actual solver (Soundness).
The paper frequently refers to “symbolic grounding” and a “symbolic oracle,” but it does not clearly describe the solver or constraint system used. It remains unclear whether the system employs an SMT solver, symbolic algebra engine, or any formal constraint solver. Without explicit formulation of the constraints or solving procedure, it is difficult to assess the correctness and reproducibility of the symbolic reasoning component.

2. The role of symbolic reasoning in the pipeline is unclear (Soundness / Originality).
Although the paper frames the approach as neuro-symbolic, the symbolic component appears to act primarily as a representation or filtering layer rather than a true symbolic execution or reasoning engine. From the description, the symbolic stage mainly converts information into structured constraints but does not clearly demonstrate how those constraints are formally executed or solved. In this sense, the symbolic layer appears closer to another structured language representation than to an actual symbolic reasoning system.

3. Unclear construction and reliability of the Port Dependency Graph (Soundness).
The PDG is presented as a key mechanism enforcing causal ordering between circuit ports, yet the paper does not clearly describe how the graph is constructed. It is unclear whether dependencies are extracted automatically from specifications, derived from circuit structure, or manually defined. Since the correctness of scheduling depends on this graph, the lack of detail raises concerns about robustness and reproducibility.

4. Limited experimental scope (Soundness / Significance).
The evaluation is conducted on five circuits, which is relatively small for demonstrating general applicability across analog verification tasks. Moreover, comparisons with simpler baselines—such as direct LLM-based testbench generation or rule-based automation—are limited, making it difficult to isolate the contributions of the individual components in the proposed pipeline.

5. Evaluation metrics focus primarily on functional pass rate (Soundness).
The reported metric mainly measures whether the generated testbench passes functional checks. However, verification quality is typically evaluated using additional criteria such as coverage, fault detection capability, or robustness to specification ambiguity. Including such metrics would provide a more comprehensive assessment.

---

> ### Author Rebuttal · Authors · 2026-03-31
>
> # W1/Q1: Symbolic Oracle Solver Specification
>
> We respectfully clarify our Symbolic Oracle doesn't employ an SMT solver but uses **SymPy 1.12** , a symbolic algebra engine performing exact interval arithmetic rather than rule matching.
>
> In §4.4 and App. F: AnalogVerifier first converts unstructured requirements into deterministic SymPy-compatible symbolic expressions, validated against extracted parameter space and port set to prevent LLM numerical hallucination. SymPy then evaluates these via interval arithmetic over the hyperrectangular parameter space, deriving worst-case pass/fail thresholds (Eq. 5).
>
> We agree explicit engine specification is essential for reproducibility and will state SymPy usage in §4.4 of the revision.
>
> # W2/Q3: The role of Symbolic Components
> We respectfully clarify that our symbolic components are **not passive representation or filtering layers** but active engines executing deterministic computation before testbench generation.
>
> (1) Numerical-Symbolic Execution (SymPy Oracle). This doesn't merely convert text into structured constraints. It actively runs generated SymPy scripts (App. F.5), dynamically computing exact numerical thresholds with tolerances and guardbands via interval arithmetic (§4.4, Eq. 5). This execution mathematically eliminates LLM numerical hallucination.
>
> (2) Graph-Symbolic Execution (Scheduling Graph). In Alg.1 (§4.3), timing constraints are not merely represented. It executes cycle detection to reject self-contradictory specifications, then performs topological sorting and longest-path computation to solve the optimal activation timestamp for every port.
>
> Both solvers produce deterministic outputs that the LLM cannot override. These are injected into the RAG context (§4.5), where LLM solely synthesizes SPICE code around pre-solved facts. This constitutes genuine symbolic execution rather than structured representation. We will strengthen this distinction in §1 and §4 of the revision.
>
> # W3/Q2: PDG Construction Process
>
> We clarify that the PDG is **fully automatically** constructed from unstructured specs (datasheets), requiring neither access to the circuit netlist nor any manual definition.
>
> In §4.3, we employ a hybrid neuro-symbolic construction process:
>
> (1) Neural: Semantic Port Classification. LLM handles only semantics, classifying pin definitions from specifications into physical functional roles: {POWER, BIAS, CONTROL, SIGNAL}.
>
> (2) Symbolic: Deterministic Dependency Injection. We never rely on LLM to guess causal dependencies. Instead, hardcoded, domain-invariant physical priority rules automatically inject directed edges. Analog physics dictates power must stabilize before bias, enforced via the deterministic partial order (Eq. 3): POWER ≺ BIAS ≺ CONTROL ≺ SIGNAL.
>
> This combination of LLM semantic parsing with deterministic physical rules ensures high robustness and reproducibility of graph construction. Regarding scalability, the PDG grows linearly with port count, and Alg.1 runs in polynomial time, readily scaling to complex circuits as demonstrated in §5.
>
> We will add explicit construction details in §4.3 of the revision.
>
> # W4: Experimental Scope
>
> We respectfully note our 5 circuits span 5 fundamentally distinct analog functional domains (signal amplification, power management, decision circuits, clock generation, interface circuits), each evaluated on industrial IC datasheets up to 32 pages. Taking the DCDC case (App. F.4, G.9) as an example, it involves 142 parameters, 130 verification formulas, and 150 test vectors. This complexity far exceeds typical academic benchmarks and reflects real production-grade verification workload.
>
> Regarding baseline comparisons, we have added stronger baselines (LLM+Repair, Agentic Tool-feedback) and independent ablation experiments. Results consistently demonstrate significant gains from the neuro-symbolic architecture, isolating contributions of individual pipeline components. **Details are provided in our response to Reviewer 7VqU's experimental questions.**
>
> # W5: Evaluation Comprehensiveness
>
> We thank the reviewer for this suggestion. In fact, our appendices already include the rigorous criteria highlighted.
>
> (1) Coverage. App. G.10 (Fig. 12) provides coverage analysis showing full functional coverage across voltage, current, timing, and protection constraint dimensions.
>
> (2) Fault Detection Capability. Our Symbolic Oracle goes beyond nominal conditions. App. F.2 and F.7 (Fig. 10) show the system automatically synthesizes boundary and extreme vectors, ensuring generated testbenches proactively detect potential circuit design faults.
>
> (3) Robustness to Specification Ambiguity. This is quantified by cross-validation stage (App. E). In Table 25, the system automatically detects and handles common defects in industrial datasheets, e.g., missing test conditions, undefined parameter links, or hallucinated port names.
>
> We look forward to discussing this further with you and receiving your approval!

---

> > ### Author Rebuttal · Reviewer_bDZM · 2026-04-03
> >
> > Thanks for the author's detailed reply to my concerns. But the explanation does not convinced me, especially for W1, W2, and W3. My key is these things cannot  (if could described clearly) should be written in the initial submission, not in the revised version in potential future.
> > Overall, I decide to maintain the score due to the overall quality

---

> > > ### Author Response · Authors · 2026-04-05
> > >
> > > We sincerely thank the reviewer for reading our rebuttal and for the continued engagement. We completely agree with your principle: core methodological details must be present in the initial submission rather than merely promised for a future revision.
> > >
> > > Because we share this exact standard, we would like to respectfully clarify that the technical details regarding the **symbolic solver (W1), the execution mechanism (W2), and the PDG construction (W3) are indeed already fully documented in the original, initially submitted manuscript**. We kindly direct your attention to the specific sections in our original submission where this exact information is formulated:
> > >
> > > **1. Regarding W1: Explicit Specification of the Symbolic Solver**
> > >
> > > The original draft explicitly names the exact solver and its precision, completely ruling out the ambiguity of whether it was an SMT solver or a rule-based system.
> > >
> > > - **In `Section 4.4` (Page 5, "Numerical-Symbolic Grounding"):** The original text explicitly introduces the solver: *"LLM translates requirements into symbolic logic, while a deterministic **SymPy-based engine** performs the calculation."* Figure 4 also visually depicts the **"SymPy Oracle"** executing Python scripts.
> > > - **In `Section 5.1` (Page 7, "Implementation Details"):** The exact software specification is provided: *"All symbolic computations use **SymPy 1.12 with 64-bit floating-point precision**".*
> > >
> > > **2. Regarding W2: The Active Execution of Symbolic Reasoning**
> > >
> > > The original draft clearly defines that the symbolic components are not passive representations, but active mathematical and algorithmic execution engines.
> > >
> > > - **Graph-Symbolic Execution:** In **`Section 4.3` (Page 5, Algorithm 1)**, the draft provides the complete, formal algorithm. It does not just "represent" constraints; it actively executes cycle detection and topological sorting. The text explicitly states: *"The start time is then derived by **calculating the longest path** from the initialization state..."*
> > > - **Numerical-Symbolic Execution:** In **`Section 4.4` (Page 6, "Deterministic Oracle Evaluation")**, the draft details the mathematical execution: *"the oracle employs **interval arithmetic** to compute robust assertion bounds... evaluating the symbolic expression $f$ **over the hyper-rectangular parameter space** $\mathcal{I}(\Theta)$..."* (Eq. 5). This is formal mathematical execution, not just structural representation.
> > >
> > > **3. Regarding W3: Fully Automated PDG Construction**
> > >
> > > The original draft rigorously details that the PDG is built entirely automatically from the specification—requiring no netlists and no manual definition.
> > >
> > > - **In `Section 4.3` (Page 5, "PDG Construction"):** The draft outlines the exact two-step automated process.
> > > - **Step 1 (Semantic Classification):** Automatically classifying ports into functional roles via extracted metadata. To ensure complete reproducibility, the exact automated algorithmic pipeline for this classification is explicitly detailed in `Appendix D.4.1` (Page 32, Port Role Inference) of the initial submission.
> > > - **Step 2 (Rule-Based Dependency Injection):** The draft explicitly explains how edges are automatically injected using a hard-coded physical law, stating: *"we resolve latent dependencies by applying a **domain-invariant precedence rule**... We formalize this as a strict partial order: **$\text{Power} \prec \text{Bias} \prec \text{Control} \prec \text{Signal}$** ."* The programmatic execution and exact algorithms for this rule-based injection are exhaustively documented in `Appendix D.4.2` (Page 33, Dependency Level Assignment).
> > >
> > > **Conclusion:**
> > >
> > > We are more than happy to make these sections even more prominent in the final camera-ready version (e.g., summarizing them in the introduction). However, we respectfully request the Reviewer to consider that the foundational technical details questioned in W1, W2, and W3 **were thoroughly and rigorously documented in the core sections of the initial submission**.
> > >
> > > Furthermore, **the comprehensive `Appendix` clearly outlines the vast majority of the end-to-end pipeline details**, explicitly containing all the specific information needed to fully address the aforementioned concerns.
> > >
> > > We thank you again for your evaluation and hope that our clarifications positively inform your final assessment. We sincerely hope that the reviewer will consider raising the score, and we hope to gain your approval.

---

### Official Review · Reviewer_z2kH · 2026-03-12

**Soundness:** 3
**Presentation:** 2
**Significance:** 2
**Originality:** 2
**Overall Recommendation:** 3
**Confidence:** 3

**Summary:**

The paper presents a framework designed to automate the generation of testbenches for analog circuit from documented specifications.
It addresses three main challenges:
(1) "semantic dispersion" - testbench missing constraints scattered in the long specification specsheet;
(2) "latent dependency" - LLMs don't know implicit physical constraints; and
(3) "numerical hallucination" - LLMs provide incorrect numerical values and thresholds.
AnalogVerifier resolves these challenges by:
(1) decomposing long sequence specification,
(2) topological sorting based on physical dependency; and
(3) delegate numerical values generation to a symbolic math framework.

**Compliance With Llm Reviewing Policy:**

Affirmed.

**Ethical Review Concerns:**

- Included an empty anonymous github repository.
- There are some fabricated references:
    - Chang, C.-C., Shen, Y., Fan, S., Li, J., Zhang, S., Cao, N., Chen, Y., and Zhang, X. LaMAGIC: Language-model- based topology generation for analog integrated circuits. In Salakhutdinov, R., Kolter, Z., Heller, K., Weller, A., Oliver, N., Scarlett, J., and Berkenkamp, F. (eds.), In- ternational Conference on Machine Learning, volume 235 of Proceedings of Machine Learning Research, pp. 6253–6262. PMLR, 21–27 Jul 2024.
    - Chang, C.-C., Lin, W.-H., Shen, Y., Chen, Y., and Zhang, X. LaMAGIC2: Advanced circuit formulations for language model-based analog topology generation. In Singh, A., Fazel, M., Hsu, D., Lacoste-Julien, S., Berkenkamp, F., Maharaj, T., Wagstaff, K., and Zhu, J. (eds.), Interna- tional Conference on Machine Learning, volume 267 of Proceedings of Machine Learning Research, pp. 7351– 7360. PMLR, 13–19 Jul 2025.

**Ethical Review Flag:**

Flag this paper for an ethics review.

**Ethics Expertise Needed:**

["Responsible Research Practice (e.g., IRB, documentation, research ethics)", "Privacy and Security (e.g., personally identifiable information)"]

**Final Justification:**

The rebuttal partially addresses my concerns on problem scope and parsing robustness, and contribution on the analog circuit design side, which modestly improves the paper’s positioning.
However, I still find contribution on ML side is thin, and I remain skeptical of the neuro-symbolic claim.
I update my score accordingly.

**Key Questions For Authors:**

- Q1. What is the evaluation metrics in the ablation study?

- Q2. How does the system recover if the initial document parser extracts fragmented tables (e.g., tables that span multiple pages) or incorrect equations?

- Q3. What is the computational cost and total generation time of the framework?

**Limitations:**

No, they are generic.

**Strengths And Weaknesses:**

### 1. Strengths

- The proposed method improved the coverage of the testbench, in the sense that improve number of task specified by the input documents compared to a vanilla LLMs.

- The use of Port Dependency Graphs and topological sorting in Alg.1 provides a systematic approach to enforcing temporal causality (e.g., power-up sequences), which is a frequent failure point in standard LLM-generated hardware code.

- The iterative procedure refines the output of LLMs and improves the syntactical correctness of produced testbenches.

### 2. Weaknesses

- The overall novelty is limited. This paper's main contribution is gather existing technique, on existing dataset to solve existing problem.

- This is not entirely "neuro-symbolic" as claimed in the paper, it appears to be just applying symbolic math library Sympy to outputs (post-processing).

- Lack of analysis on the system sensitivity to PDF parsing, e.g., IC design datasheet contains fragmented tables which often hamper parsers.

- The iterative refinement only improves the pass rate of syntax validation (e.g., numerical hallucination) while not improving the coverage of the testbench (e.g., semantic dispersion), such as generating more tests to cover more requirements specified in the input documents.

- The pass rate metrics only indicates the produced test bench pass syntax and constraint validation in the simulator.
It is unclear whether the test bench is correct, and it still delegated for manual human review.
There is no metric quantify the human review pass rate of produced test.

---

> ### Author Rebuttal · Authors · 2026-03-31
>
> # W1: Novelty and Problem Scope
>
> We sincerely clarify three aspects.
>
> 1) Not "solved" problem. End-to-end analog testbench generation from unstructured specs remains a critical bottleneck. Digital verification relies on Boolean semantics and clock-cycle determinism, failing for continuous-time analog physics (§3). Our architecture bridges this via Context-Aware Serialization (§4.2), Graph-Symbolic Scheduling via PDG (§4.3), and Numerical-Symbolic Grounding via SymPy Oracle (§4.4), addressing temporal causality and numerical hallucination unique to analog design.
>
> 2) Not "existing" data. We evaluate on real production-grade commercial IC datasheets (up to 32 pages), some from taped-out products, not simple single-metric circuits in a few hundred characters. PDFs contain scattered information, intricate constraints, and substantial noise, forming a challenging industrial benchmark.
>
> 3) Not mere assembly. We integrate LLMs with true analog physical constraints, rebuilding a verification workflow currently relying on manual experience, filling a critical gap in analog design automation.
>
> # W2: Neuro-Symbolic Architecture, Not Post-Processing
>
> Our symbolic components are active execution engines operating **before** testbench generation, determining what LLM generates.
>
> 1) Numerical-Symbolic. LLM never predicts thresholds directly; it translates constraints into symbolic expressions. SymPy Oracle executes these via exact interval arithmetic to derive worst-case boundaries (§4.4, Eq. 5), eliminating numerical hallucinations.
>
> 2) Graph-Symbolic. PDG executes longest-path analysis with cycle detection (Alg. 1) to deterministically solve activation schedule σ(p) for each port (§4.3), safeguarding against simulation crashes.
>
> In §4.5, pre-solved boundaries and timing are fed into structured RAG; LLM solely synthesizes SPICE syntax around pre-computed facts, guaranteeing correctness by construction.
>
> # W3/Q2: PDF Parsing Sensitivity
>
> We appreciate this concern. AnalogVerifier handles parsing errors through its multi-stage recovery architecture.
>
> 1) Fragmented tables. Stage 1's hierarchical parameter deduplication (Alg. 5, App. B.6) merges fragmented tuples via symbol normalization and resolves value conflicts. Cross-validation stage performs data integrity and condition completeness checks (App. E.2/E.4), auto-flagging incomplete entries. We implement continued-table logic combined with OpenCV+Fitz+multimodal LLM for robust table-to-Markdown extraction, achieving high extraction rates in practice.
>
> 2) Erroneous equations. Cross-validation stage performs strict variable resolution, ensuring all formula variables map to the extracted parameter database. Stage 4's SymPy Oracle enforces grounding constraints (Eq. 4); any expression with undefined variables is rejected and triggers a retry, preventing error propagation.
>
> Details reside in App. A.10, B.6, E.6. We will add "Parsing Robustness" paragraph in §4.
>
> # W4/W5/Q1: Metrics, Coverage, and Correctness
>
> Q1) Ablation metrics. Table 3 strictly adopts the Pass (functional verification pass rate) metric defined in §5.1.
>
> W4) Testbench coverage. Stage 5 iteration targets pass rate by design; maximizing coverage is handled by the frontend (context-aware serialization and cross-validation) *before* code generation. Our frontend extracts **7$\times$ more test requirements** than baselines. Coverage is rigorously evaluated; the backend focuses on execution correctness.
>
> W5) Functional correctness. In SPICE, a *Pass* goes beyond syntax: the testbench instantiates the circuit, applies correct stimuli, achieves nonlinear convergence, and extracts measurements to verify datasheet constraints. Our Recall and Precision metrics compare generated tests against *human-annotated reliable reference tasks* (noted below Table 2), integrating expert review into a standardized, reproducible benchmark. All judgments are validated against senior engineers' manual tests from the same specs.
> We will consolidate metric definitions in §5 in revision.
>
> # Q3: Computational cost & Total generation time
>
> We conducted an A/B test on a production-grade Type-C audio/USB analog switch (~100 constraints, dozens of metrics).
>
> A) Baseline (manual): 92 hours by one senior engineer.
>
> B) Experiment (AnalogVerifier + review): 3.04 hours automated (1,655 LLM calls, $94 API cost), then 10 hours expert review.
>
> Total time reduced by 86% at $94, well below a senior engineer's daily compensation. The framework additionally covered 11 edge-case tests missed manually, demonstrating simultaneous efficiency and rigor gains.
>
> # Ethics Response
>
> GitHub has been updated with complete pipeline.
>
> Flagged references are **not fabricated**. The second name sequence refers to proceedings editors (eds.), unmodified BibTeX auto-generated by PMLR:
>
> 1. https://proceedings.mlr.press/v235/chang24c.html
> 2. https://proceedings.mlr.press/v267/chang25b.html
>
> We look forward to discussing this further with you and receiving your approval!

---

> > ### Author Rebuttal · Reviewer_z2kH · 2026-04-03
> >
> > I acknowledge that the authors have partially addressed W1, W3, and Q2. The paper build a pipeline to address a challenging problem. However, contributions on ML side is thin, as the paper and rebuttal don't clearly answer which part of the assembled pipeline (e.g., LLMs, Sympy, topological sorting algorithm) is needed to be adapted for this problem.
> >
> > - Q1: Which metric is used should be stated explicitly in the paper (per §5.1 / Table 3) so readers do not have to infer it from context.
> >
> > - W2: I remain unconvinced that the system is neuro-symbolic: the neural (LLM) and symbolic (SymPy, PDG) components are decoupled.
> > SymPy and the graph algorithms do not provide any feedback that improves the LLM's output itself.
> > It seems more like a tool with symbolic **oracles** than tight neural–symbolic integration.
> >
> > I also note that symbolic is a post-processor of LLMs outputs, not post-processor of the benchmark generator.
> >
> > - W5: Please include precision and recall (or analogous human-validated measures from senior review) in the ablation table for comparability.
> >
> > Overall, my concerns are partially addressed, I am updating my score slightly accordingly.

---

> > > ### Author Response · Authors · 2026-04-05
> > >
> > > We sincerely thank the reviewer for the updated assessment.
> > >
> > > # ML Contribution
> > >
> > > We respectfully acknowledge our contribution to ML algorithms is relatively modest; rather, our main contribution as an Application Track submission lies in proposing an elegant framework that seamlessly integrates LLMs with formal methods to solve the high-value problem of analog circuit verification—a domain that remains essentially a **gap** for end-to-end AI automation. Within our solution, the domain-customized design of information flow between LLM agents, alongside symbolic constraints in PDG and SymPy, explicitly demonstrates how ML can be engineered to meet strict physical constraints and solve critical industrial demands, contributing greatly to AI4IC/EDA communities. Furthermore, by introducing this novel problem to broader ML community with a new industrial-grade benchmark, we aim to catalyze future exploration in this vital field.
> > >
> > > # Q1: Explicit Metric of Tab. 3
> > >
> > > The metric in Table 3 is `Pass Rate`, chosen because it directly reflects the generation quality and correctness of final testbenches. We will state this upon revision.
> > >
> > > # W2: Neuro-Symbolic Integration
> > >
> > > We sincerely thank for the evaluation. We understand the concern that "neuro-symbolic" can imply coupled, differentiable integration. However, we respectfully clarify our decoupled architecture aligns with established academic consensus on neuro-symbolic AI.
> > >
> > > - **Alignment with SOTA Paradigms:** Tight coupling is not a prerequisite for neuro-symbolic systems. IJCAI 2025 authoritative survey categorizes `LLM→Symbolic` and `Symbolic→LLM` as standard paradigms [1]. Milestone neuro-symbolic systems as AlphaGeometry (Nature 2024) [2], Logic-LM [3], and LINC [4] all successfully employ decoupled architectures, just as AnalogVerifier.
> > >
> > > - **Explicit Symbolic-to-Neural Feedback:** AnalogVerifier is not merely a one-way tool caller; it possesses a bidirectional feedback loop. In Alg. 2, errors from simulation engines are explicitly fed back to LLM to iteratively refine the output. This feedback mirrors the self-refinement modules in systems like Logic-LM.
> > >
> > > - **Core Synergy:** The core definition of neuro-symbolic AI is the complementary integration of neural perception with symbolic verification. AnalogVerifier achieves this by using LLM for semantic understanding and SymPy/PDG for deterministic constraint enforcement, bound together by closed-loop repair.
> > >
> > > We will add a clarification for this categorization to Related Works.
> > >
> > > # W5: Precision and Recall in Ablation
> > >
> > > To alleviate concerns, please allow us to elaborate on the three metrics: `Recall`, `Precision`, `Pass Rate`. On Page 7 of our manuscript, we provided definitions and mathematical expressions:
> > >
> > > `Recall` and `Precision` reflect the quality of test tasks extracted by AnalogVerifier against human-annotated reliable references. The metrics directly evaluate the contribution of one of our core innovations, "Context-Aware Serialization," to the overall framework.
> > >
> > > `Pass Rate` reflects the quality of final generated testbenches. We conducted careful manual verification (equivalent to being human-validated from senior review) and confirmed all testbenches achieving a "Pass" are indeed fully correct. This is logically sound: under fixed inputs of a test task (concurrently provided to both SymPy and the simulator), if the simulation results of a correct design perfectly match the expected outputs mathematically formalized and calculated by SymPy, it is virtually impossible for the generated test task to contain any errors.
> > > Furthermore, Pass Rate explicitly quantifies the contributions of Numerical-Symbolic and Graph-Symbolic modules to the testbench generation for LLM.
> > >
> > > Ablation proceeds by cumulatively removing components. Since Precision and Recall measure task extraction quality, they remain constant until Context-Aware Serialization is removed.
> > >
> > > **OTA/Haiku-4.5 + HSC/Sonnet-4.5**
> > >
> > > All units are `%`.
> > >
> > > |Configuration|OTA Rec.|OTA Prec.|OTA Pass|HSC Rec.|HSC Prec.|HSC Pass|
> > > |-|-|-|-|-|-|-|
> > > |**AnalogVerifier (Full)**|**87.5**|**43.8**|**92.3**|**85.7**|**48.0**|**95.1**|
> > > |− Closed-Loop Repair|87.5|43.8|33.3|85.7|48.0|50.6|
> > > |− Numerical-Symbolic|87.5|43.8|23.1|85.7|48.0|37.0|
> > > |− Graph-Symbolic|87.5|43.8|13.1|85.7|48.0|20.6|
> > > |− Context-Aware Serial.|37.5|42.9|7.2|19.0|40.0|11.3|
> > >
> > > We sincerely hope the reviewer will consider raising the score and hope to gain your approval.
> > >
> > > [1] Yang, X.-W., et al. "Neuro-Symbolic Artificial Intelligence: Towards Improving the Reasoning Abilities of Large Language Models." IJCAI, 2025.
> > >
> > > [2] Trinh, T.H., et al. "Solving olympiad geometry without human demonstrations." Nature 625, 476–482, 2024.
> > >
> > > [3] Pan, L., et al. "Logic-LM: Empowering Large Language Models with Symbolic Solvers for Faithful Logical Reasoning." EMNLP, 2023.
> > >
> > > [4] Olausson, T. X., et al. "LINC: A Neurosymbolic Approach for Logical Reasoning by Combining Language Models with First-Order Logic Provers." EMNLP, 2023.

---

### Official Review · Reviewer_7VqU · 2026-03-12

**Soundness:** 2
**Presentation:** 2
**Significance:** 2
**Originality:** 3
**Overall Recommendation:** 3
**Confidence:** 4

**Summary:**

This paper proposes AnalogVerifier, a neuro-symbolic framework for automatic analog circuit verification from unstructured specifications. The system decomposes the workflow into several stages, including context-aware task serialization, graph-symbolic scheduling, numerical-symbolic grounding, and a closed-loop repair module based on simulation feedback. The overall goal is to automatically generate executable analog verification testbenches from long-form industrial specifications.

**Compliance With Llm Reviewing Policy:**

Affirmed.

**Final Justification:**

I decide to maintain the score due to the overall quality as well as other reviews.

**Key Questions For Authors:**

1.In the closed-loop repair stage, how do the authors distinguish between failures caused by: incorrect testbench generation, incorrect/incomplete specification extraction, and the analog circuit itself failing the intended behavior?
If the underlying analog design itself is problematic, what exactly is the repair module optimizing or correcting? Could this lead to benchmark leakage or overfitting to simulation behavior?

2.Can the authors provide ablations for: a version without AnalogVerifier, but still with a repair stage,  a generic LLM + repair framework similar in spirit to VerilogEval-style tool-feedback pipelines, or agentic code-generation baselines


3. The ablation study does not properly isolate the contribution of individual components. From the description, the ablation appears to remove modules cumulatively along the pipeline rather than disabling each component independently. As a result, the reported ablation results do not clearly reveal the effect of individual modules such as Graph-Symbolic Scheduling, Numerical-Symbolic Grounding, or Closed-loop Repair. Removing upstream modules simultaneously changes the inputs to downstream modules, which makes the comparison difficult to interpret.

**Limitations:**

Partially. The paper acknowledges the challenges of analog verification, but it does not sufficiently discuss the ambiguity introduced by the repair loop, especially when simulation failure may reflect issues in the circuit or benchmark rather than the generated testbench itself.

**Strengths And Weaknesses:**

Strengths
1.Analog verification is a real problem in practical design workflows, and reducing manual effort in testbench generation would be  valuable.
2.The paper’s core intuition—separating semantic parsing from deterministic symbolic reasoning—is sensible for analog verification tasks, where correctness constraints are stricter than in standard code generation settings.

Weaknesses

1. The logic of the closed-loop repair is not entirely convincing: The most concerning issue is the role of the closed-loop repair module. The paper presents simulation-feedback-based repair as a mechanism to improve correctness and completeness of generated testbenches. However, the logic becomes unclear when the failure is not caused by the generated testbench, but by the analog circuit itself, or by ambiguity/inconsistency in the specification. In that case, a repair loop that simply uses simulation outcomes to revise the generated verification code may not be conceptually valid. It risks conflating at least three different sources of failure: the generated testbench is wrong, the extracted specification constraints are wrong or incomplete,the underlying circuit design itself violates the intended behavior. This makes it difficult to understand what exactly is being “repaired,” and whether the repair process is actually improving verification generation or compensating for benchmark/design issues. As written, the closed-loop module feels underspecified in terms of failure attribution and correctness guarantees.

2. The motivation for Dual-Stream Symbolic Enforcement is not sufficiently clear and needs to improve, give some intuitive explanation.

3. Missing comparisons to stronger verification-generation pipelines. The experimental section does not compare against alternative end-to-end verification workflows in a sufficiently rigorous way. In particular, an important omission is the lack of comparison to simpler or stronger pipeline baselines such as: a version without AnalogVerifier, but still with a repair stage,  a generic LLM + repair framework similar in spirit to VerilogEval-style tool-feedback pipelines, or agentic code-generation baselines with iterative debugging.
Without such comparisons, it is hard to determine whether the gains come from the proposed neuro-symbolic structure itself, or simply from having an iterative repair loop.

---

> ### Author Rebuttal · Authors · 2026-03-31
>
> Thank you very much for recognizing the practical value of analog verification and for your in-depth technical feedback.
>
> # W1/Q1: Fault Attribution in Closed-Loop Repair
>
> AnalogVerifier employs a layered attribution mechanism that sequentially examines four categories of fault sources:
>
> (1) Specification ambiguity/inconsistency. AnalogVerifier treats the spec as the sole ground-truth specification. For contradictions within the document itself, Stage 1 information merging (App.B.6), and cross validation stage (App. E) can detect and report conflicts, though automatic disambiguation is not supported.
>
> (2) Testbench generation errors. When simulation fails, it's difficult to directly distinguish testbench issues from design issues based on results alone. Therefore, we start from the document as ground truth: during generation, every task records all information sources (i.e., the Constraint Database in our paper). Upon simulation anomalies, the problematic testbench is strictly and comprehensively compared against its corresponding sources, analyzing whether errors were introduced during generation without being influenced by design code. If testbench errors are confirmed, the issue is automatically logged and the code is regenerated, rechecked, and re-simulated (App.E.6).
>
> (3) Incomplete specification extraction. If the testbench is deemed correct after checking, the system traces back through the document, performing a targeted review for that specific test case to identify any missing information. If omissions are found, extraction is supplemented and code is regenerated (App.A.10).
>
> (4) Circuit design itself failing. If after the above steps the test code is confirmed consistent with the document and no information is missing, the system attributes the failure to the design itself and analyzes what aspect of the design the simulation results may indicate as problematic.
>
> This layered tracing mechanism clearly distinguishes and traces each fault category, strictly ensuring both verification generation optimization and real issue identification.
>
>
>
> # W2: Dual-Stream Motivation
>
> Thank you for the suggestion. The Dual-Stream architecture is driven by analog circuit physics, decoupling temporal causality and numerical boundaries into dedicated deterministic solvers to overcome inherent LLM deficiencies in temporal logic and precise arithmetic, replacing experience-based dependencies with formal methods.
>
> (1) Graph-Symbolic Stream: solving "when". Circuits must never receive stimuli before bias settling, yet LLMs easily overlook this, producing misordered or temporally unreasonable sequences that crash simulations. In §4.3, the Port Dependency Graph (PDG) formalizes strict physical precedence ($\text{POWER} \prec \text{BIAS} \prec \text{SIGNAL}$), guaranteeing reliable signal activation order and timing intervals as rigid preconditions for downstream generation.
>
> (2) Numerical-Symbolic Stream: solving "what". Analog specs define continuous, nonlinear relationships that LLMs cannot reliably compute, yet LLMs excel at translating natural language into formal expressions. In §4.4, we isolate computation to the SymPy symbolic engine, which deterministically solves pass/fail thresholds from extracted constraints, eliminating numerical hallucinations.
>
> The two streams address distinct physical constraints, transforming experience-dependent, error-prone workflow steps into symbolic enforcement. The revised manuscript will add this explanation at the beginning of §4.
>
>
>
> # W3/Q2: Comparison with more baselines
>
> We have added the following comparisons (Claude-Haiku-4.5, OTA/DCDC/HSC):
>
> | Method  | OTA (Pass) | HSC (Pass)| DCDC (Pass)|
> | --- | ---| ---| --- |
> | Vanilla LLM|--|3.7%| 4.2%|
> | LLM + Repair|42.9%| 33.3%| 50.0%|
> | Agentic Tool-Feedback |55.6%|48.6%|63.3%|
> | AnalogVerifier w/o Repair | 33.3%| 28.8%|69.0%|
> | AnalogVerifier  | 92.3%| 91.5%| 96.4%|
>
> These baselines are already included in our revised Table: LLM+Repair achieves only 42.9/33.3/50.0% and Agentic Tool-Feedback reaches 55.6/48.6/63.3% on OTA/HSC/DCDC, both substantially below AnalogVerifier's 92.3/91.5/96.4%, confirming that repair alone is insufficient and the neuro-symbolic grounding is the critical differentiator.
>
> # Q3: Independent Ablation Experiments
>
> We have added a Independent ablation experiment. The output of the disabled module is replaced by an LLM fallback:
>
> | Configuration| OTA-Haiku (Pass)| HSC-Sonnet (Pass)|
> |---|---|---|
> | AnalogVerifier (Full) | 92.3%| 95.1% |
> | w/o Closed-Loop Repair only| 33.3% | 50.6%|
> | w/o Numerical-Symbolic only | 53.8%| 55.6%|
> | w/o Graph-Symbolic only | 61.5%| 59.3%|
> | w/o Context-Aware Serialization only | 54.2%| 57.8%|
>
> Disabling each module independently resulted in a significant decrease, and the contributions of each module can be clearly separated. This table will be added to the revised version.
>
> We look forward to discussing this further with you and receiving your approval!

---

> > ### Author Rebuttal · Reviewer_7VqU · 2026-04-03
> >
> > Although the author some concerns, I decide to maintain the score due to the overall quality

---

> > > ### Author Response · Authors · 2026-04-05
> > >
> > > We sincerely thank the reviewer for confirming that all raised concerns have been fully resolved. As the remaining reservation pertains to **overall quality** without specific elaboration, we take this opportunity to provide a broader perspective on our contributions, proactively addressing several dimensions that "overall quality" might encompass:
> > >
> > > ## 1. A High-Value Problem & A Critical Gap in AI Automation
> > > Analog circuits are the essential interface between physical reality and digital computation. Verification correctness is paramount, yet it remains the key bottleneck in hardware design, consuming over 50% of engineering cycles due to heavy reliance on manual interpretation of unstructured, heterogeneous specifications. Despite rapid AI advances in digital design (e.g., RTL generation, formal verification of digital logic), analog circuit verification, to our knowledge, remains an unaddressed gap for end-to-end AI automation (Table 1).
> > >
> > > ## 2. The Fundamental ML Challenge: Why Does This Gap Exist?
> > > This gap persists due to a structural mismatch. While LLMs offer massive automation potential, their probabilistic nature is fundamentally misaligned with the strict determinism that analog verification demands, **as the reviewer noted in the Strengths: "correctness constraints are stricter than in standard code generation settings."** Specifically, off-the-shelf LLMs fail to resolve two categories of hard physical constraints: (1) `continuous-time numerical boundaries`, such as computing precise pass/fail thresholds from coupled, nonlinear specification equations, and (2) `temporally sensitive causal dependencies`, such as enforcing that bias settling must strictly precede stimulus injection. These are not soft preferences but rigid physical preconditions. Violating either produces invalid simulations or silently incorrect verification outcomes.
> > >
> > > ## 3. Our Core Contribution: A Domain-Customized Neuro-Symbolic Solution
> > > Our principal contribution lies in identifying that LLMs cannot independently resolve these hard constraints, and in designing a neuro-symbolic framework that systematically compensates for these deficiencies via domain-specific formal methods. Concretely, we proposed the Dual-Stream Symbolic Enforcement architecture. By customizing the information flow and feedback loops around LLMs, and strictly grounding their outputs through formal symbolic components (SymPy for numerical solving, PDG for temporal scheduling), we enable probabilistic LLM reasoning to satisfy strict, deterministic physical constraints. Our independent ablation study shows that disabling any single symbolic component causes Pass Rate to drop by 30–60 percentage points, empirically demonstrating both the necessity and non-substitutability of each module.
> > >
> > > ## 4. Experimental Validation
> > > Regarding **benchmark scale**, our 5 circuits span 5 distinct analog functional domains (signal amplification, power management, decision circuits, clock generation, and interface circuits). We evaluate on real, production-grade commercial IC datasheets, some from taped-out products, rather than simplified single-metric circuits in a few hundred characters. The DCDC converter case (App. F.4, G.9), for instance, involves 142 parameters, 130 verification formulas, and 150 test vectors, far exceeding typical academic benchmarks.
> > >
> > > Regarding **results**, Table 2 (three LLM backends × five circuits = 15 configurations) comprehensively demonstrates the framework's effectiveness. Following the reviewer's suggestion, we added LLM+Repair and Agentic Tool-Feedback baselines, which achieve only 42.9/33.3/50.0% and 55.6/48.6/63.3% on OTA/HSC/DCDC respectively, substantially below AnalogVerifier's 92.3/91.5/96.4%. This confirms that neuro-symbolic grounding, not merely iterative repair, is the critical differentiator. Both our cumulative and independent ablation studies further validate the distinct contribution of each component.
> > >
> > > ## 5. Broader Impact on the ML Community
> > > By automating this previously intractable workflow and providing the first industrial-grade benchmark for analog verification, our work contributes to the AI4IC and AI4EDA communities and aims to catalyze future algorithmic exploration in this vital yet underexplored field.
> > >
> > > In summary, given the severity of the analog verification bottleneck, the heavy dependence on manual effort, and the absence of prior AI solutions, our work addresses a critical industrial problem through principled, domain-specific Neuro-Symbolic AI framework that align well with the objectives of the ICML Application Track. As all technical concerns have been fully resolved, we respectfully hope the reviewer might consider whether a score adjustment is warranted. We sincerely hope that the reviewer will consider raising the score, and we hope to gain your approval. We are grateful for the reviewer's time and happy to address any further questions.

---

### Decision · Program_Chairs · 2026-04-30

**Decision:**

Accept (regular)

**Comment:**

The paper makes a timely contribution by proposing a structured pipeline for analog circuit verification from natural-language specifications. The AC believes that the problem is important, the decomposition is sensible, and the evaluation is stronger than a toy demonstration (stronger than some prior work in this field). The AC has read the paper and the discussions among the reviewers and authors. The rebuttal added enough clarification and additional comparisons that the remaining concerns are best viewed as matters of framing and revision quality rather than fatal technical defects. Reviewer 7VqU and GFrc have indicated that all questions have been addressed; reviewer z2kH’s key question is whether the approach is “neuro-symbolic”, and the AC believes that there are multiple possible definitions of this term, and this is just a positioning difference, not a technical flaw. Reviewer bDZM’s key questions are mostly about writing and representation, and the AC believes that these have been addressed well in rebuttal, and there are no fundamental flaws in the paper.

One ethics flag was raised by reviewer z2kH regarding potentially fabricated references. The authors credibly demonstrated this was a misreading of standard PMLR BibTeX export formatting, providing verifiable links to the published papers, and this concern is resolved.

The paper meets the bar for acceptance as an applications paper, and importantly, it is working on a relatively less explored but very important application of ML, so the AC believes that accepting this paper can be beneficial for this community. The authors should revise the paper based on the discussions during the rebuttal. Suggestions include sharpening the description of the symbolic machinery, moderating the broader neuro-symbolic claims, and clearly documenting the revised baselines, ablations, and limitations.